# SPARSE ALIGNMENT ENHANCED LATENT DIFFUSION TRANSFORMER FOR ZERO-SHOT SPEECH SYNTHESIS

## ABSTRACT

While recent zero-shot text-to-speech (TTS) models have significantly improved speech quality and expressiveness, mainstream systems still suffer from issues related to speech-text alignment modeling: 1) autoregressive large language models are inefficient and not robust in long-sentence inference; 2) non-autoregressive diffusion models without explicit speech-text alignment require substantial model capacity for alignment learning; 3) predefined alignment-based diffusion models suffer from naturalness constraints of forced alignments and a complicated inference pipeline. This paper introduces *S-DiT*, a TTS system featuring an innovative sparse alignment algorithm that guides the latent diffusion transformer (DiT). Specifically, 1) we provide sparse alignment boundaries to S-DiT to reduce the difficulty of alignment learning without limiting the search space; 2) to simplify the overall pipeline, we propose a unified frontend language model (F-LM) training framework to cover various speech processing tasks required by TTS models. Additionally, we adopt the piecewise rectified flow technique to accelerate the generation process and employ a multi-condition classifier-free guidance strategy for accent intensity adjustment. Experiments demonstrate that S-DiT matches state-of-the-art zero-shot TTS speech quality while maintaining a more efficient pipeline. Moreover, our system can generate high-quality one-minute speech with only 8 sampling steps. Audio samples are available at `https://sditdemo.github.io/sditdemo/`.

## 1 INTRODUCTION

In recent years, neural codec language models (Wang et al., 2023; Zhang et al., 2023; Song et al., 2024; Xin et al., 2024) and large-scale diffusion models (Shen et al., 2023; Matthew et al., 2023; Lee et al., 2024a; Eskimez et al., 2024; Ju et al., 2024; Yang et al., 2024d;b) have brought considerable advancements to the field of speech synthesis. Unlike traditional text-to-speech (TTS) systems (Shen et al., 2018; Jia et al., 2018; Li et al., 2019; Kim et al., 2020; Ren et al., 2019; Kim et al., 2021; 2022), these models are trained on large-scale, multi-domain speech corpora, which contributes to notable improvements in the naturalness and expressiveness of synthesized audio. Given only seconds of speech prompt, these models can synthesize identity-preserving speech in a zero-shot manner.

To generate high-quality speech with clear and expressive pronunciation, a TTS model must establish an alignment mapping from text to speech signals (Kim et al., 2020; Tan et al., 2021). However, from the perspective of speech-text alignment, current solutions suffer from the following issues:

- **Autoregressive codec language models (AR LM)** are inefficient and lack robustness. These models (Wang et al., 2023; Chen et al., 2024a) achieve the alignment paths through attention mechanisms in their time-autoregressive generation processes. However, the lengthy discrete speech codes, which typically require a minimum bit rate of 1.5 kbps (Kumar et al., 2024; Wu et al., 2024), impose a significant burden on these autoregressive language models.

- **Diffusion models without predefined alignments (Diffusion w/o PA)** require substantial parameters. Recent diffusion-based TTS works (Lee et al., 2024a; Eskimez et al., 2024; Lovelace et al., 2023; Gao et al., 2023; Cámbara et al., 2024; Yang et al., 2024d;b) demonstrate that non-autoregressive diffusion models can effectively perform text-to-speech synthesis without the need for explicit duration modeling, which significantly speeds up the speech generation process. However, these algorithms require a significant portion of

Table 1: Intrinsic characteristics of zero-shot TTS systems. "$-$" denotes the moderate performance.

| Characteristics | AR LM | Diffusion w/o PA | Diffusion w/ PA | Ours |
|---|---|---|---|---|
| Representative Works | VALL-E 1/2 | DiTTo-TTS | NaturalSpeech 2/3 | / |
| w/o Prosodic Constraints from Alignments | ✓ | ✓ | × | ✓ |
| Robust | × | $-$ | ✓ | ✓ |
| Controllable Duration | × | $-$ | ✓ | ✓ |
| Parameter Efficient | × | × | ✓ | ✓ |
| Simple Training Data Preparation | ✓ | ✓ | × | × |
| Simple Inference Pipeline | ✓ | ✓ | × | ✓ |
| Fast Inference | × | ✓ | $-$ | ✓ |

parameters to establish the text-to-speech alignment. ARDiT (Liu et al., 2024b) proves that when compared under an identical number of parameters, methods without explicit duration modeling exhibit some decline in speech intelligibility. Besides, these methods cannot provide fine-grained control over the duration of specific pronunciations and can only adjust the overall speech rate.

- **Predefined alignment-based diffusion models (Diffusion w/ PA)** have prosodic naturalness constraints of forced alignments and a complex inference process. During training, alignment paths (Ren et al., 2020; Kim et al., 2020) are directly introduced into their models (Matthew et al., 2023; Shen et al., 2023; Ju et al., 2024) to reduce the complexity of text-to-speech generation, which achieves higher intelligibility and similarity. Nevertheless, they suffer from the following two limitations: 1) predefined alignments constrain the model's search space to produce natural-sounding speech (Anastassiou et al., 2024; Chen et al., 2024a); 2) an external alignment tool is required in inference to obtain the duration prompt, which is time-consuming and complicates the overall pipeline.

Intuitively, we can integrate the two aforementioned diffusion-based methods to pursue optimal performance. To be specific, 1) we propose a novel sparse speech-text alignment strategy to enhance the latent diffusion transformer (DiT), termed S-DiT. In our approach, phoneme tokens are sparsely distributed within the corresponding forced alignment regions to provide coarse pronunciation information that is then refined by the latent DiT model; 2) we propose a joint training framework for the frontend language model that facilitates TTS models. In previous zero-shot TTS pipelines, training and inference often rely on various complex frontend systems, such as automatic speech recognition (ASR) (Radford et al., 2023), grapheme-to-phoneme (G2P) conversion (Park & Kim, 2019; Park & Lee, 2020; Bernard & Titeux, 2021), external alignment tools (McAuliffe et al., 2024), and duration prediction (Kim et al., 2020; Ren et al., 2020; Ju et al., 2024; Yang et al., 2024b). In this work, however, we find that these systems can be merged into a unified language model to efficiently handle all four frontend tasks within a single autoregressive process.

Experimental results demonstrate that S-DiT achieves nearly state-of-the-art speaker similarity on the LibriSpeech test-clean set (Panayotov et al., 2015) with only 8 sampling steps, while also exhibiting high speaker similarity. The main contributions of this work are summarized as follows:

- We design a sparse alignment enhanced latent diffusion transformer model (S-DiT) that combines the naturalness of "diffusion w/o PA" with the robustness of "diffusion w/ PA". The advantages of our model are listed in Table 1. Moreover, sparse alignment is more robust against duration prediction errors than forced alignment. We also visualize the attention score matrices of different layers in S-DiT and obtain interesting conclusions in Appendix G.

- To achieve higher generation quality and more flexible control, we propose a multi-condition CFG strategy to adjust the guidance scales for speaker timbre and text content separately. Furthermore, we discover that the text guidance scale can also be used to modulate the intensity of personal accents, offering a new direction for enhancing speech expressiveness.

- We successfully reduce S-DiT's inference steps from 25 to 8 with the piecewise rectified flow (PeRFLow) technique, achieving highly efficient zero-shot TTS with minimal quality degradation. Moreover, when we scale S-DiT from 0.5B to 7B parameters, it exhibits exceptional performance while maintaining a low inference latency.

- Our proposed F-LM not only simplifies the inference process of zero-shot TTS models but also can be directly used for processing training data during model fine-tuning. The unified training framework enhances F-LM's speech understanding capabilities, allowing it to surpass the independent modules for each subtask.

## 2 BACKGROUND

**Zero-shot TTS.** Zero-shot TTS (Casanova et al., 2022; Wang et al., 2023; Zhang et al., 2023; Shen et al., 2023; Matthew et al., 2023; Jiang et al., 2024; Liu et al., 2024b; Lee et al., 2024a; Li et al., 2024; Lee et al., 2023; Ju et al., 2024; Meng et al., 2024; Chen et al., 2024b) aims to synthesize unseen voices with speech prompts. Among them, neural codec language models (Chen et al., 2024a) are the first that can autoregressively synthesize speech that rivals human recordings in naturalness and expressiveness. However, they still face several challenges, such as the lossy compression in discrete audio tokenization and the time-consuming nature of autoregressive generation. To address these issues, some subsequent works explore solutions based on continuous vectors and non-autoregressive diffusion models (Shen et al., 2023; Matthew et al., 2023; Lee et al., 2024a; Eskimez et al., 2024; Yang et al., 2024d;b; Chen et al., 2024b). These works can be categorized into two main types: 1) the first type directly models speech-text alignments using attention mechanisms without explicit duration modeling (Lee et al., 2024a; Eskimez et al., 2024). Although these models perform well in terms of generation speed and quality, they typically require a large number of parameters to learn speech-text alignments. The second category (Shen et al., 2023; Matthew et al., 2023) utilizes predefined alignments to simplify alignment learning. However, the search space of the generated speech of these models is limited by predefined alignments and the inference pipeline is quite complex. To address these limitations, 1) we propose a sparse alignment mechanism to reduce the constraints of predefined alignment-based methods while also reducing the difficulty of speech-text alignment learning; 2) we introduce a frontend language model to simplify the inference and fine-tuning pipeline. Additionally, we describe the CFG mechanism used in zero-shot TTS systems in Appendix B.

**Accented TTS.** While accented TTS is not yet mainstream in the field of speech synthesis, it offers valuable potential for customized TTS services, by enhancing the expressiveness of speech synthesis systems and improving listeners' comprehension of speech content (Tan et al., 2021; Melechovsky et al., 2022; Badlani et al., 2023; Zhou et al., 2024; Shah et al., 2024; Ma et al., 2023; Inoue et al., 2024; Zhong et al., 2024). With the emergence of conversational AI systems, accented TTS technology has even broader application scenarios. In this paper, we focus on a specific task of accented TTS: adjusting the accent intensity of speakers to make them sound like native English speakers or accented speakers who use English as a second language (Liu et al., 2024a). Unlike previous work, our approach does not require paired data or accurate accent labels; instead, it allows for flexible control over the accent intensity using the proposed multi-condition CFG mechanism.

**TTS Frontend Systems.** In traditional TTS systems, the frontend typically refers to text analysis modules (Tan et al., 2021), such as text normalization (Sproat & Jaitly, 2016; Zhang et al., 2020) and grapheme-to-phoneme conversion (Yao & Zweig, 2015; Park & Lee, 2020; Bernard & Titeux, 2021; Chen et al., 2022). With the emergence of zero-shot TTS, the frontend has taken on additional responsibilities, including processing the prompt speech during the inference stage, which should at least support automatic speech recognition (ASR). Moreover, some advanced non-autoregressive models (Ju et al., 2024; Li et al., 2024; Lee et al., 2023; Matthew et al., 2023) require additional speech-text aligners and duration predictors. These complex frontend modules impose significant limitations on the efficiency of zero-shot TTS models. In this work, we unify these frontend components into a single language model, thereby simplifying the overall pipeline.

## 3 METHOD

This section introduces S-DiT. To begin with, we describe the architecture design of S-DiT. Then, we provide detailed explanations of the sparse alignment mechanism, the piecewise rectified flow acceleration technique, and the multi-condition classifier-free guidance strategy. Finally, we outline the unified frontend language model training framework and the overall system's inference pipeline.

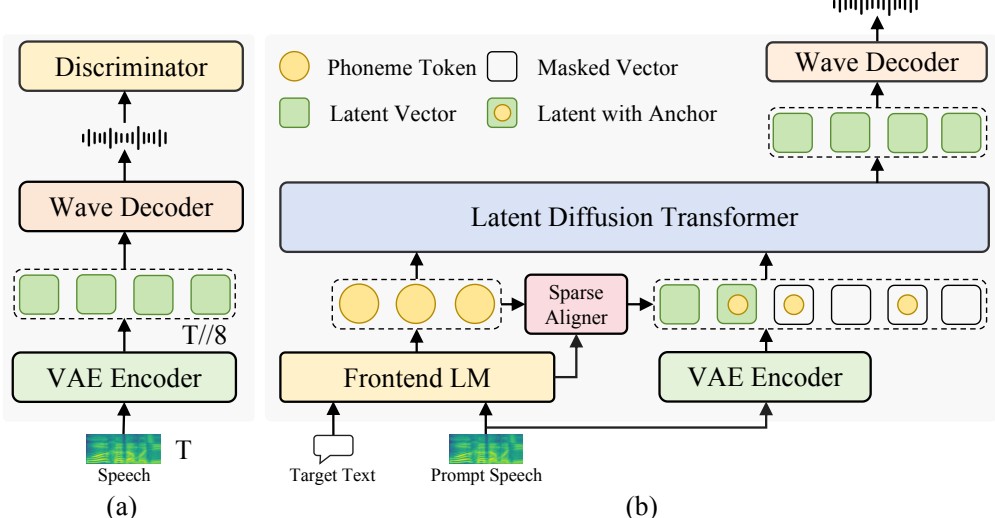

Figure 1: (a) The speech compression model. (b) Overview of S-DiT. We insert the sparse alignment anchors into the latent vector sequence to provide coarse alignment information. The transformer blocks in S-DiT will automatically build fine-grained alignment paths.

## 3.1 ARCHITECTURE

**Speech Compression.** As shown in Figure 1 (a), given a speech spectrogram $s \in \mathcal{R}^{T \times C}$, the VAE encoder $E$ encodes $s$ into a latent vector $z$, and the wave decoder $D$ reconstructs the waveform $x = D(z) = D(E(s))$, where $T$ is the time dimension and $C$ is the frequency dimension. To reduce the computational burden of the model and simplify speech-text alignment learning, the encoder $E$ downsamples the spectrogram by a factor of $d = 8$ in length. The encoder $E$ is similar to the one used in Rombach et al. (2022), and the decoder $D$ is based on Kong et al. (2020). We also adopt the multi-period discriminator (MPD), multi-scale discriminator (MSD), and multi-resolution discriminator (MRD) (Kong et al., 2020; Jang et al., 2021) to model the high-frequency details in waveforms, which ensure perceptually high-quality reconstructions. The training loss of the speech compression model can be formulated as $\mathcal{L} = \mathcal{L}_{\text{rec}} + \mathcal{L}_{\text{KL}} + \mathcal{L}_{\text{Adv}}$, where $\mathcal{L}_{\text{rec}} = \|s - \hat{s}\|^2$ is the spectrogram reconstruction loss, $\mathcal{L}_{\text{KL}}$ is the slight KL-penalty loss (Rombach et al., 2022), and $\mathcal{L}_{\text{Adv}}$ is the LSGAN-styled adversarial loss (Mao et al., 2017). After training, a one-second speech clip can be encoded into 12.5 vector frames. For more details, please refer to Appendix A.1 and J.

**Latent Diffusion Transformer with Masked Speech Modeling.** The latent diffusion transformer is used to predict speech that matches the style of the given speaker and the content of the provided text. Given the random variables $Z_0$ sampled from a standard Gaussian distribution $\pi_0$ and $Z_1$ sampled from the latent space given by the speech compression model with data density $\pi_1$, we adopt the rectified flow Liu et al. (2022) to implicitly learn the transport map $T$, which yields $Z_1 := T(Z_0)$. The rectified flow learns $T$ by constructing the following ordinary differential equation (ODE):

$$\mathrm{d}Z_t = v(Z_t, t)\,\mathrm{d}t, \tag{1}$$

where $t \in [0, 1]$ denotes time and $v$ is the drift force. Equation 1 converts $Z_0$ from $\pi_0$ to $Z_1$ from $\pi_1$. The drift force $v$ drives the flow to follow the direction $(Z_1 - Z_0)$. The latent diffusion transformer, parameterized by $\theta$, can be trained by estimating $v(Z_t, t)$ with $v_\theta(Z_t, t)$ through minimizing the least squares loss with respect to the line directions $(Z_1 - Z_0)$:

$$\min_v \int_0^1 \mathbb{E}\left[\|(Z_1 - Z_0) - v(Z_t, t)\|^2\right]\,\mathrm{d}t. \tag{2}$$

We use the standard transformer block from LLAMA (Dubey et al., 2024) as the basic structure for S-DiT and adopt the Rotary Position Embedding (RoPE) (Su et al., 2024) as the positional embedding. During training, we randomly divide the latent vector sequence into a prompt region $z_{prompt}$ and a masked target region $z_{target}$, with the proportion of $z_{prompt}$ being $\gamma \sim U(0.1, 0.9)$. We use $v_\theta$

to predict the masked target vector $\hat{z}_{target}$ conditioned on $z_{prompt}$ and the phoneme embedding $p$, denoted as $v_\theta(\hat{z}_{target}|z_{prompt}, p)$. The loss is calculated using only the masked region $z_{target}$. S-DiT learns the average pronunciation from $p$ and the specific characteristics such as timbre, accent, and prosody of the corresponding speaker from $z_{prompt}$.

## 3.2 SPARSE ALIGNMENT ENHANCED LATENT DIFFUSION TRANSFORMER (S-DiT)

In this subsection, we describe the sparse alignment strategy as the foundation of S-DiT, followed by the piecewise rectified flow and multi-condition CFG strategies to further enhance S-DiT's capacity.

**Sparse Alignment Strategy.** Let's first analyze the reasons behind the characteristics of different speech-text alignment modeling methods in depth. "Diffusion w/o PA" requires more parameters for speech intelligibility due to the difficulty in end-to-end modeling of speech-text alignment non-autoregressively. On the other hand, the use of predefined hard alignment paths limits the model's search space and increases the complexity of the pipeline. The characteristics of these systems motivate us to design an approach that combines the advantages of both: we first provide a rough alignment to S-DiT and then use attention mechanisms in Transformer blocks to construct the fine-grained implicit alignment path. The visualizations of the implicit alignment paths are included in Appendix G. In specific, denote the latent speech vector sequence as $z = [z_1, z_2, \cdots, z_n]$, the phoneme sequence as $p = [p_1, p_2, \cdots, p_m]$, and the phoneme duration sequence as $d = [d_1, d_2, \cdots, d_m]$, where $n$, $m$ is the length of the sequence. The length of the speech vector that corresponds to a phoneme $p_i$ is the duration $d_i$. Given $d = [2, 2, 3]$, the hard speech-text alignment path used by "Diffusion w/ PA" can be denoted as $a = [p_1, p_1, p_2, p_2, p_3, p_3, p_3]$. To construct the rough alignment $\tilde{a}$, we randomly retain only one anchor for each phoneme: $\tilde{a} = [\underline{M}, p_1, p_2, \underline{M}, \underline{M}, \underline{M}, P_3]$, where $\underline{M}$ represents the mask token. $\tilde{a}$ is downsampled to match the length of the latent sequence $z$. Then, we directly concatenate the downsampled $\tilde{a}$ and $z$ along the channel dimension. We also concatenate the phoneme embedding $p$ with $z$ along the time dimension as the prefix information. The anchors in $\tilde{a}$ provide S-DiT with approximate positional information for each phoneme, simplifying the model's learning of speech-text alignment. At the same time, the rough alignment information does not limit S-DiT's search space and also enables fine-grained control over each phoneme's duration.

**Piecewise Rectified Flow Acceleration.** We adopt Piecewise Rectified Flow (PeRFlow) (Yan et al., 2024) to distill the pretrained S-DiT model into a more efficient generator. Although our S-DiT is non-autoregressive in terms of the time dimension, it requires multiple iterations to solve the Flow ODE. The number of iterations (i.e., number of function evaluations, NFE) has a great impact on inference efficiency, especially when the model scales up further. Therefore, we adopt the PeRFlow technique to further reduce NFE by segmenting the flow trajectories into multiple time windows. Applying reflow operations within these shortened time intervals, PeRFlow eliminates the need to simulate the full ODE trajectory for training data preparation, allowing it to be trained in real-time alongside large-scale real data during the training process. Given number of windows $K$, we divide the time $t \in [0, 1]$ into $K$ time windows $\{(t_{k-1}, t_k]\}_{k=1}^K$. Then, we randomly sample $k \in \{1, \cdots, K\}$ uniformly. We use the startpoint of the sampled time window $z_{t_{k-1}} = \sqrt{1 - \sigma^2(t_{k-1})}z_1 + \sigma(t_{k-1})\epsilon$ to solve the endpoint of the time window $\hat{z}_{t_k} = \phi_\theta(z_{t_{k-1}}, t_{k-1}, t_k)$, where $\epsilon \sim \mathcal{N}(0, I)$ is the random noise, $\sigma(t)$ is the noise schedule, and $\phi_\theta$ is the ODE solver of the teacher model. Since $z_{t_{k-1}}$ and $\hat{z}_{t_k}$ is available, the student model $\hat{\theta}$ can be trained via the following objectives:

$$\ell = \left\| v_{\hat{\theta}}(z_t, t) - \frac{\hat{z}_{t_k} - z_{t_{k-1}}}{t_k - t_{k-1}} \right\|^2, \tag{3}$$

where $v_{\hat{\theta}}$ is the estimated drift force with parameter $\hat{\theta}$ and $t$ is uniformly sampled from $(t_{k-1}, t_k]$. We provide details of PeRFlow training for S-DiT in Appendix C.

**Multi-condition Classifier-Free Guidance (CFG).** We employ classifier-free guidance approach (Ho & Salimans, 2022) to steer the model $g_\theta$'s output towards the conditional generation $g_\theta(z_t, c)$ and away from the unconditional generation $g_\theta(z_t, \varnothing)$:

$$\hat{g}_\theta(z_t, c) = g_\theta(z_t, \varnothing) + \alpha \cdot [g_\theta(z_t, c) - g_\theta(z_t, \varnothing)], \tag{4}$$

where $c$ denotes the conditional state, $\varnothing$ denotes the unconditional state, and $\alpha$ is the guidance scale selected based on experimental results. Unlike standard classifier-free guidance, S-DiT's conditional

states $c$ consist of two components: phoneme embeddings $p$ and the speaker prompt $z_{prompt}$. In the experiments, as the text guidance scale increases, we observe that the pronunciation changes according to the following pattern: 1) starting with improper pronunciation; 2) then shifting to pronouncing with the current speaker's accent; 3) and finally approaching the standard pronunciation of the target language. The detailed experimental setup are described in Appendix M. This allows us to use the text guidance scale $\alpha_{txt}$ to control the accent intensity. At the same time, the speaker guidance scale $\alpha_{spk}$ should be a relatively high value to ensure a high speaker similarity. Therefore, we adopt the multi-condition classifier-free guidance technique to separately control $\alpha_{txt}$ and $\alpha_{spk}$:

$$\hat{g}_\theta(z_t, p, z_{prompt}) = \alpha_{spk} \left[ g_\theta(z_t, p, z_{prompt}) - g_\theta(z_t, p, \varnothing) \right] + \alpha_{txt} \left[ g_\theta(z_t, p, \varnothing) - g_\theta(z_t, \varnothing, \varnothing) \right] \\ + g(z_t, \varnothing, \varnothing) \tag{5}$$

In training, we randomly drop condition $z_{prompt}$ with a probability of $p_{spk} = 0.10$. Only when $z_{prompt}$ is dropped, we randomly drop condition $p$ with a probability of 50%. Therefore, our model is able to handle all three types of conditional inputs described in Equation 5. We select the guidance scale $\alpha_{spk}$ and $\alpha_{txt}$ based on experimental results.

### 3.3 FRONTEND LANGUAGE MODEL (F-LM)

**Training Strategy.** Our frontend language model transforms the ASR, speech-text alignment, G2P, and duration prediction processes required in the TTS pipeline into a unified sequence modeling task. Denote the phoneme embedding sequence as $p = [p_1, p_2, \cdots, p_m]$, the duration embedding sequence as $d = [d_1, d_2, \cdots, d_m]$, the speech vector sequence as $a = [a_1, a_2, \cdots, a_l]$, and the byte-pair encoding (BPE) sequence of the transcription as $t = [t_1, t_2, \cdots, t_{\hat{m}}]$. For duration representation $d$, to inform the model of how long it has been speaking during inference, we use the absolute timestamp of each phoneme on the time axis to construct the "phoneme/timestamp tokens" sequence in Figure 2, which can be represented as $\hat{p}_t = [p_1, d_1, p_2, d_1 + d_2, \cdots, p_m, \sum_{i=1}^{m} d_i]$.

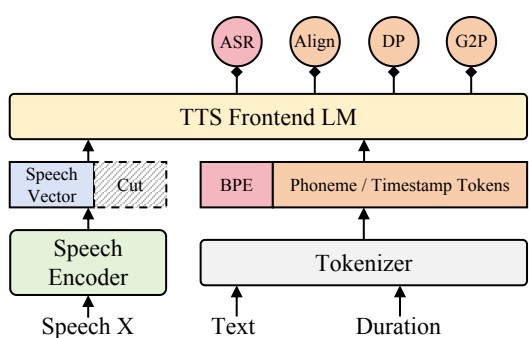

Figure 2: The frontend language model, which first solves the ASR task, followed by addressing the aligning, DP, and G2P tasks simultaneously.

In training, we first concatenate the speech vector sequence $a$ and the BPE sequence $t$ and the phoneme/timestamp sequence $\hat{p}_t$ as the input $h$ to the decoder-only LM, which can be represented as $h = [a_1, \cdots, a_l, \ t_1, \cdots, t_{\hat{m}}, \ p_1, d_1, \cdots, p_m, \sum_{i=1}^{m} d_i]$. Then, we added special tokens to indicate the start and end of sequences $t$ and $\hat{p}_t$. Notably, as shown in Figure 2, we randomly discard the latter part of the speech vector sequence. This allows the phoneme/timestamp sequence corresponding to the discarded region to be used in training F-LM for duration prediction (DP) and G2P. Meanwhile, the BPE sequence and the phoneme/timestamp sequence from the non-discarded region can be used to train F-LM for ASR and speech-text aligning, respectively. Details about F-LM's training procedure are included in Appendix A.1 and Appendix E. Our experiments in Section 4.4 demonstrate that large-scale unified training can improve the robustness and generalization of frontend models.

**Inference Pipeline.** During inference, we can enjoy a highly simplified pipeline with F-LM. As shown in Figure 1, starting with a speech prompt, we first extract its text through ASR. We then append the target text to the ASR result and finally obtain the predicted phonemes and durations for the target text. The entire pipeline can be completed in a single autoregressive process, making it highly efficient. Moreover, in Section 4.4, F-LM achieves superior and generalizable performance than that of individual models, demonstrating the effectiveness of the proposed unified training.

## 4 EXPERIMENTS

In this subsection, we describe the datasets, training, inference, and evaluation metrics. We provide the model configuration and detailed hyper-parameter setting in Appendix A.1.

### 4.1 EXPERIMENTAL SETUP

**Datasets.**  We train S-DiT and F-LM on the LibriLight (Kahn et al., 2020) dataset, which contains 60k hours of unlabeled speech derived from LibriVox audiobooks. All speech data are sampled at 16KHz. We transcribe the speeches using an internal ASR system and extract the predefined speech-text alignment using the external alignment tool (McAuliffe et al., 2017). We utilize two benchmark datasets: 1) the librispeech (Panayotov et al., 2015) test-clean set following (Shen et al., 2023; Ju et al., 2024) for zero-shot TTS and F-LM's evaluation; 2) the L2-arctic dataset (Zhao et al., 2018) following (Melechovsky et al., 2022; Liu et al., 2024a) for accented TTS evaluation.

**Training and Inference.**  We train the speech compression model, S-DiT, and F-LM on 8 NVIDIA A100 GPUs. The batch sizes, optimizer settings, and learning rate schedules are described in Appendix A.1. It takes 2M steps for the speech compression model's training and 1M steps for S-DiT and F-LM's training until convergence. The pre-training of S-DiT requires 800k steps and PeRFlow distillation requires 200k steps. During the inference stage, given the prompt speech and target text, F-LM will process all the information required by S-DiT. Then, S-DiT synthesizes the target latent vector, which is converted into the target waveform by the wav decoder. The entire inference pipeline is simple and efficient.

**Objective Metrics.**  1) For zero-shot TTS, we evaluate speech intelligibility using the word error rate (WER) and speaker similarity using SIM-O (Ju et al., 2024). To measure SIM-O, we utilize the WavLM-TDCNN speaker embedding model[1] to calculate the cosine similarity score between the generated samples and the prompt. As SIM-R (Matthew et al., 2023) is not comparable across baselines using different acoustic tokenizers, we recommend focusing on SIM-O in our experiments. The similarity score is in the range of $[-1, 1]$, where a higher value indicates greater similarity. In terms of WER, we use the publicly available HuBERT-Large model (Hsu et al., 2021), fine-tuned on the 960-hour LibriSpeech training set, to transcribe the generated speech. The WER is calculated by comparing the transcribed text to the original target text. All samples from the test set are used for the objective evaluation; 2) For accented TTS, we evaluate the Mel Cepstral Distortion (MCD) in dB level and the moments (standard deviation ($\sigma$), skewness ($\gamma$) and kurtosis ($\kappa$)) (Andreeva et al., 2014; Niebuhr & Skarnitzl, 2019) of the pitch distribution to evaluate whether the model accurately captures accent variance; 3) For F-LM, we evaluate the WER for ASR models, the alignment boundary error (AE) for speech-text aligners, and the duration error (DE) for duration predictors.

**Subjective Metrics.**  We conduct the MOS (mean opinion score) evaluation on the test set to measure the audio naturalness via Amazon Mechanical Turk. We keep the text content and prompt speech consistent among different models to exclude other interference factors. We randomly choose 40 samples from the test set of each dataset for the subjective evaluation, and each audio is listened to by at least 10 testers. We analyze the MOS in three aspects: CMOS (quality, clarity, naturalness, and high-frequency details), SMOS (speaker similarity in terms of timbre reconstruction and prosodic pattern), and ASMOS (accent similarity). We tell the testers to focus on one corresponding aspect and ignore the other aspect when scoring.

### 4.2 RESULTS OF ZERO-SHOT SPEECH SYNTHESIS

**Evaluation Baselines.**  We compare the zero-shot speech synthesis performance of S-DiT with 11 strong baselines, including: 1) VALL-E (Wang et al., 2023); 2) VALL-E 2 (Chen et al., 2024a); 3) VoiceBox (Matthew et al., 2023); 4) StyleTTS 2 (Li et al., 2024); 5) HierSpeech++ (Lee et al., 2023); 6) UniAudio (Yang et al., 2023b); 7) Mega-TTS 2 (Jiang et al., 2024); 8) ARDiT (Liu et al., 2024b); 9) DiTTo-TTS (Lee et al., 2024a); 10) NaturalSpeech 3 (Ju et al., 2024); 11) CosyVoice (Du et al., 2024); Explanation and details of the selected baseline systems are provided in Appendix A.4.

**Analysis**  As shown in Table 2, we can see that 1) S-DiT achieves state-of-the-art SIM-O, SMOS, and WER scores, comparable to NaturalSpeech 3 (the "Diffusion w/ PA" counterpart), and significantly surpasses other "Diffusion w/o PA" models. The improved SIM-O and SMOS suggest that the proposed sparse alignment effectively simplifies the text-to-speech mapping challenge like predefined

---

[1]https://github.com/microsoft/UniSpeech/tree/main/downstreams/speaker_verification

Table 2: Zero-shot TTS results on LibriSpeech test-clean set. * means the results are obtained from the paper. † means the results are obtained from the authors. #Params denotes the number of parameters. RTF denotes the real-time factor.

| Model | #Params | Training Data | SIM-O↑ | SIM-R↑ | WER↓ | CMOS↑ | SMOS↑ | RTF↓ |
|---|---|---|---|---|---|---|---|---|
| GT | - | - | 0.68 | - | 1.94% | +0.12 | 3.92 | - |
| VALL-E* | 0.4B | LibriLight | - | 0.58 | 5.90% | - | - | 4.520 |
| VALL-E 2* | 0.4B | LibriHeavy | 0.64 | 0.68 | 2.44% | - | - | - |
| VoiceBox† | 0.4B | Collected (60kh) | 0.64 | 0.67 | 2.03% | -0.20 | 3.81 | 0.340 |
| StyleTTS 2 | 0.2B | Collected (0.6kh) | 0.38 | - | 2.49% | -0.26 | 3.31 | **0.045** |
| HierSpeech++ | 0.1B | Collected (2.8kh) | 0.51 | - | 6.33% | -0.37 | 3.58 | 0.047 |
| UniAudio | 1.0B | Mixed (165kh) | 0.57 | 0.68 | 2.49% | -0.24 | 3.85 | 3.586 |
| Mega-TTS 2† | 0.4B | LibriLight | 0.53 | 0.59 | 2.32% | -0.21 | 3.72 | 0.368 |
| ARDiT† | 0.4B | LibriTTS | 0.56 | - | 2.38% | -0.22 | 3.70 | 1.061 |
| DiTTo-TTS* | 0.7B | Collected (55kh) | 0.62 | 0.65 | 2.56% | - | - | - |
| NaturalSpeech 3† | 0.5B | LibriLight | **0.67** | **0.76** | **1.81%** | -0.10 | 3.95 | 0.296 |
| CosyVoice | 0.4B | Collected (172kh) | 0.62 | - | 2.24% | -0.18 | 3.93 | 1.375 |
| S-DiT | 0.5B | LibriLight | **0.67** | 0.70 | 1.84% | **0.00** | **3.94** | 0.208 |
| S-DiT-accelerated | 0.5B | LibriLight | 0.65 | 0.69 | 1.92% | -0.04 | 3.91 | 0.160 |

Table 3: The objective and subjective experimental results for accented TTS. MCD (dB) denotes the Mel Cepstral Distortion at the dB level. $\sigma$, $\gamma$, and $\kappa$ are the standard deviation, skewness, and kurtosis of the pitch distribution.

| Model | MCD (dB)↓ | $\sigma$↑ | $\gamma$↓ | $\kappa$↓ | ASMOS↑ | CMOS↑ | SMOS↑ |
|---|---|---|---|---|---|---|---|
| GT | - | 45.1 | 0.591 | 0.783 | 4.03 | +0.09 | 3.95 |
| CTA-TTS | 5.98 | 41.1 | 0.602 | 0.799 | 3.72 | -0.60 | 3.64 |
| S-DiT | **5.69** | **42.3** | **0.601** | **0.790** | **3.84** | **+0.00** | **3.89** |

forced duration information, allowing the model to focus more on learning timbre information. And the improved WER indicates that S-DiT also enjoys strong robustness; 2) S-DiT significantly surpasses all baselines in terms of CMOS, demonstrating the effectiveness of the proposed sparse alignment strategy; 3) After the PeRFlow acceleration, the student model of S-DiT shows on par quality with the teacher model and enjoys extremely fast inference speed. For a fair comparison, we ignore the time taken by the frontend processing for each model when calculating the RTF in Table 2. Even when taking the frontend processing time into account, the RTF of our pipeline is only 0.432, which is highly efficient. Detailed average frontend processing time comparisons are included in Appendix K. The duration controllability of S-DiT is verified in Appendix F. We also validate whether the prosodic naturalness is enhanced by sparse alignments in Appendix N.

## 4.3 RESULTS OF ACCENTED TTS

In this subsection, we evaluate the accented TTS performance of our model on the L2-ARCTIC dataset (Zhao et al., 2018). This corpus includes recordings from non-native speakers of English whose first languages are Hindi, Korean, etc. In this experiment, we focus on verifying whether our model and baseline can synthesize natural speech with different accent types (standard English or English with specific accents) while maintaining consistent vocal timbre. We compare our S-DiT model with CTA-TTS (Liu et al., 2024a). More details of the baseline model are provided in Appendix A.5. 1) First, we evaluate whether the models can synthesize high-quality speeches with ac-

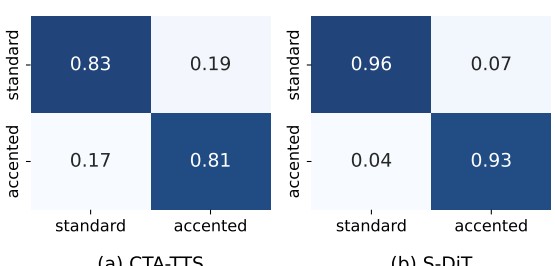

Figure 3: The confusion matrices between the perceived and intended accent categories of synthesized speech. The X-axis and Y-axis represent the intended and perceived categories, respectively.

Table 4: ASR accuracy comparison. We report the WER (%) metric on the LibriSpeech test-clean and test-other set.

| ASR Model | test-clean | test-other |
|---|---|---|
| Mini-Omni | 4.5 | 9.7 |
| Whisper-small | **3.4** | **7.6** |
| F-LM | 4.2 | 8.3 |

Table 5: Duration accuracy comparison. $\Delta_p$ and $\Delta_s$ denote the absolute boundary difference of phonemes and sentences, respectively.

| Duration Model | $\Delta_p$ (ms) | $\Delta_s$ (s) |
|---|---|---|
| NAR-based | $28.52 \pm 0.75$ | $2.25 \pm 0.68$ |
| AR-based | $21.47 \pm 0.91$ | $1.81 \pm 0.77$ |
| F-LM | $\mathbf{18.80 \pm 0.94}$ | $\mathbf{1.59 \pm 0.74}$ |

cents. As shown in Table 3, our S-DiT model significantly outperforms the CTA-TTS baseline in terms of the subjective accent similarity MOS core, the MCD (dB) values, and the statistical moments ($\sigma$, $\gamma$, and $\kappa$) of pitch distributions. These results demonstrate the superior accent learning capability of S-DiT compared to the baseline system. Besides, the S-DiT model achieves higher CMOS and SMOS scores compared to CTA-TTS, indicating a significant improvement in speech quality and speaker similarity; 2) Secondly, we evaluate whether the models can accurately control the accent types of the generated speeches. We follow CTA-TTS to conduct the intensity classification experiment (Liu et al., 2024a). At run-time, we generate speeches with two accent types, and the listeners are instructed to classify the perceived accent categories, including "standard" and "accented". Figure 3 shows that our S-DiT significantly surpasses CTA-TTS in terms of accent controllability.

## 4.4 Results of F-LM

In this subsection, we evaluate the performance of our frontend language model (F-LM) on the LibriSpeech test-clean set. In this experiment, we evaluate the performance of F-LM on three important front-end tasks during the TTS inference process: ASR, speech-text aligning, and duration prediction. 1) For ASR, we compared our model with Mini-Omni (Xie & Wu, 2024), an end-to-end speech understanding and synthesis system based on the language model, and Whisper-small (Radford et al., 2023), an advanced expert ASR system that has the similar model size as F-LM. From Table 4, it can

Table 6: Results for speech-text aligning. $\Delta_p$ means the absolute alignment boundary difference of phonemes.

| Aligner Model | $\Delta_p$ (ms) |
|---|---|
| MFA | $13.42 \pm 0.73$ |
| F-LM | $\mathbf{8.79 \pm 0.59}$ |

be seen that F-LM has comparable WER scores with the strong baseline systems, demonstrating its speech understanding capacity; 2) For speech-text aligning, we train a Montreal Forced Aligner (MFA) (McAuliffe et al., 2017) on the LibriLight dataset as the baseline. Based on Table 6, the speech-text alignment accuracy of F-LM is significantly higher than that of MFA; 3) For duration prediction, we train a non-autoregressive (NAR) duration predictor following Ren et al. (2020) and an auto-regressive (AR) duration predictor following Jiang et al. (2024) as the baselines. In the experiments, we keep the parameter size of the baselines consistent with that of F-LM to ensure a fair comparison. Table 5 demonstrates that F-LM is superior to NAR-based and AR-based methods in terms of duration prediction accuracy, due to F-LM's large-scale unified training pipeline; For additional experimental results, please refer to Appendix E.

## 4.5 Ablation Studies

**Alignments and CFG**   We test the following four settings: 1) *w/ Forced Alignment*, which replaces the sparse alignment in S-DiT with forced alignment used in (Matthew et al., 2023; Shen et al., 2023); 2) *w/o Alignment*, we do not use the predefined alignments and modeling the duration information implicitly; 3) *w/ Standard CFG*, we use the standard CFG following the common practice in Diffusion-

Table 7: Ablation studies of alignment strategies and CFG mechanisms on the LibriSpeech test-clean set.

| Setting | SIM-O↑ | WER↓ | CMOS↑ | SMOS↑ |
|---|---|---|---|---|
| Ours | 0.67 | 1.84% | 0.00 | 3.94 |
| *w/ Forced Alignment* | 0.67 | 1.82% | -0.17 | 3.94 |
| *w/o Alignment* | 0.61 | 2.55% | -0.12 | 3.88 |
| *w/ Standard CFG* | 0.65 | 1.80% | -0.02 | 3.89 |
| *w/o CFG* | 0.45 | 6.93% | -0.56 | 3.35 |

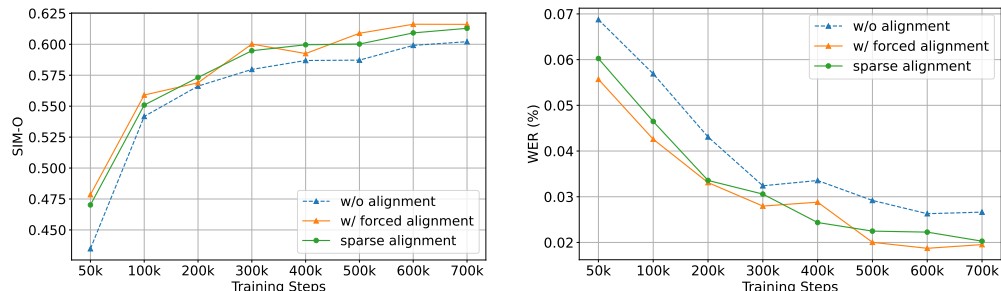

Figure 4: The visualization for effects of different speech-text alignment strategies on S-DiT training.

based TTS; 4) *w/o CFG*, we do not use the CFG mechanism. All tests follow the experimental setup described in Section 4.2. The results are shown in Table 7. For settings 1) and 2), it can be observed that both forced alignment and sparse alignment can enhance the performance of speech synthesis models. However, compared to forced alignment, sparse alignment does not constrain the model's search space, leading to a higher CMOS score. We also evaluate the effects of sparse alignment on training efficiency by visualizing the WER and SIM curve in S-DiT's training process in Figure 4. It can be seen that the training efficiency of "sparse alignment" is similar to "w/ forced alignment" and both of them surpass "w/o alignment", indicating that both sparse alignment and forced alignment can reduce the training difficulty. Moreover, we visualize the attention score matrices from different transformer layers in S-DiT in Appendix G, leading to some interesting observations. For setting 3), compared with the standard CFG, our multi-condition CFG performs slightly better as it allows for flexible control over the weights between the text prompt and the speaker prompt. Setting 4) proves that the CFG mechanism is crucial for S-DiT.

**Data and Model Scaling** We evaluate the effectiveness of data and model scaling on the proposed S-DiT model. In this experiment, we train models with 0.5B parameters on multilingual internal datasets with data sizes of 2kh, 40kh, 200kh, and 600kh, respectively. We also train models with 0.5B, 1.5B, and 7.0B parameters on the 600kh dataset. We evaluate the zero-shot TTS performance in terms of speaker similarity (Sim-O) and speech intelligibility (WER) on an internal test set consisting of 400 speech samples from various sources. Based on Table 8, we conclude that: 1) as the data size increases from 2kh to 600kh, both the model's speaker similarity and speech intelligibility improve consistently, demonstrating strong data scalability of our model; 2) as the model size scales from 0.5B to 7.0B parameters, SIM-O improves by 12.1% and WER decreases by 9.52%, validating the model scalability of S-DiT. Additionally,

Table 8: Results of data and model scaling experiments.

| Setting | SIM-O↑ | WER↓ |
|---|---|---|
| 2kh | 0.52 | 4.27% |
| 40kh | 0.63 | 2.98% |
| 200kh | 0.65 | 2.34% |
| 600kh | **0.66** | **2.10%** |
| 0.5B | 0.66 | 2.10% |
| 1.5B | 0.72 | 1.98% |
| 7.0B | **0.74** | **1.90%** |

we find that increasing the model parameters enhances its para-linguistic capabilities, with specific audio examples available on the demo page. The detailed descriptions of the training corpus, test set, and visualizations are included in Appendix D.

## 5 CONCLUSIONS

In this paper, we introduce S-DiT, a zero-shot TTS framework that 1) leverages novel sparse alignment boundaries to ease the difficulty of alignment learning while retaining the naturalness of the generated speeches, and 2) incorporates a unified front-end language model (F-LM) to streamline the overall pipeline. These strategies allow our approach to combine the strengths of both "Diffusion w/o PA" and "Diffusion w/ PA" methods. Additionally, we employ the PeRFlow technique to further accelerate the generation process and design a multi-condition classifier-free guidance strategy to offer more flexible control over accents. Experimental results show that S-DiT achieves state-of-the-art zero-shot TTS speech quality while maintaining a more efficient pipeline. Due to space constraints, further discussions are provided in the appendix.

## 6 ETHICS STATEMENT

The proposed model, S-DiT, is designed to advance zero-shot TTS technologies, making it easier for users to generate personalized speech. When used responsibly and legally, this technique can enhance applications such as movies, games, podcasts, and various other services, contributing to increasing convenience in everyday life. However, we acknowledge the potential risks of misuse, such as voice cloning for malicious purposes. To mitigate this risk, solutions like building a corresponding deepfake detection model will be considered. Additionally, we plan to incorporate watermarks and verification methods for synthetic audio to ensure ethical use in real-world applications. Restrictions will also be included in the licensing of our project to further prevent misuse. By addressing these ethical concerns, we aim to contribute to the development of responsible and beneficial AI technologies, while remaining conscious of the potential risks and societal impact.

## 7 REPRODUCIBILITY STATEMENT

We have taken several steps to ensure the reproducibility of the experiments and results presented in this paper: 1) the architecture and algorithm of the S-DiT model are described in Section 3 and and relevant hyperparameters are fully described in Appendix A.1; 2) The evaluation metrics, including WER, SIM-O, MCD (dB), the moments of the pitch distribution, alignment error, CMOS, SMOS, and ASMOS, are described in detail in Section 4.1; 3) For most of the key experiments, we utilize publicly available datasets such as LibriLight, LibriSpeech, and L2Arctic. The selection of the test sets is identical to that used in previous zero-shot TTS research. However, as the publicly available datasets are insufficient for our data scaling experiments, we construct a larger dataset, which is described in detail in Appendix D; 4) To ensure reproducibility of the results, we have carefully set random seeds in our experiments and the random seeds are provided in Appendix A.2. All objective results reported are based on the average performance across multiple runs.

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

# A    DETAILED EXPERIMENTAL SETTINGS

## A.1    MODEL CONFIGURATION

Our model comprises a speech compression VAE, an S-DiT, and an F-LM.

- **The speech compression VAE** consists of a VAE encoder, a wave decoder, and discriminators; The VAE encoder follows the architecture used in Stable Diffusion (Rombach et al., 2022) but we replace the 2D convolution layers with 1D convolution layers and remove the attention layers to accommodate data of arbitrary lengths and to improve efficiency. The channel size is 256 with channel multipliers [1, 2, 4, 8]. The wave decoder consists of a stable diffusion decoder and a Hifi-GAN decoder (Kong et al., 2020). The stable diffusion decoder shares the same hyperparameter settings as the encoder, which is used for upsampling the latent vectors. The latent channel size is set to 16. The weight of the KL loss is set to $1 \times 10^{-2}$, which only imposes a slight KL penalty on the learned latent. In training, we use batches of fixed length, consisting of 800 mel-spectrogram frames, with a batch size set to 50 for each GPU. We use the Adam optimizer with a learning rate of $1 \times 10^{-4}$, $\beta_1 = 0.9$, $\beta_2 = 0.999$, and 10K warmup steps.

- **The S-DiT model** use the standard transformer block from LLAMA (Dubey et al., 2024) as the basic structure, which comprises a 24-layer Transformer with 16 attention heads and 1024 embedding dimensions. It contains 339M parameters in total. We adopt the Rotary Position Embedding (RoPE) (Su et al., 2024) as the positional embedding following the common practice in LLAMA implementations. For simplicity, we do not use the phoneme encoder and style encoder like previous works. We only use a linear projection layer to transform these features to the same dimension. During training, we use 8 A100 80GB GPUs with a batch size of 12K latent frames per GPU for 1M steps. We use the Adam optimizer with a learning rate of $5 \times 10^{-5}$, $\beta_1 = 0.9$, $\beta_2 = 0.999$, and 10K warmup steps. In zero-shot TTS experiments, we set the text guidance scale $\alpha_{txt}$ and the speaker guidance scale $\alpha_{spk}$ to 2.5 and 3.5, respectively. In accented TTS experiments, we set $\alpha_{spk} = 6.5$, $\alpha_{txt} = 1.5$ to generate the accented speech and set $\alpha_{spk} = 2.0$, $\alpha_{txt} = 5.0$ to generate the speech with standard English.

- **The F-LM** use the same architecture as S-DiT. F-LM use an 8-layer Transformer with 16 attention heads and 1024 embedding dimensions, which contains 124M parameters in total. The audio encoder of F-LM follows the architecture of Whisper-small encoder (Radford et al., 2023). We use the tokenizers from Yi-1.5[2] to obtain the BPE tokens from texts. To improve robustness, we add SpecAugment (Park et al., 2019) in the training process. We use the Adam optimizer with a learning rate of $1 \times 10^{-4}$, $\beta_1 = 0.9$, $\beta_2 = 0.999$, and 10K warmup steps.

## A.2    RANDOM SEEDS

We ran objective experiments 10 times with 10 different random seeds and obtained the averaged results. The chosen random seeds are [4475, 5949, 6828, 6744, 3954, 3962, 6837, 1237, 3824, 3163].

## A.3    SAMPLING STRATEGY

For S-DiT, we applied the Euler sampler with a fixed step size following the common practice in flow ODE sampling. We use 25 and 8 sampling steps for *S-DiT* and *S-DiT-accelerated*, respectively. For F-LM, when transcribing the prompt speech, we use beam search with 5 beams using the log probability as the score function to reduce repetition looping following Radford et al. (2023). For G2P conversion and speech-text aligning, we use greedy decoding with top-1 sampling. For duration prediction, we use top-50 sampling to enhance the output diversity.

## A.4    DETAILS ABOUT ZERO-SHOT TTS BASELINES

In this subsection, we provide the details about the baselines in our zero-shot TTS experiments:

---

[2]https://github.com/01-ai/Yi

- **VALL-E** (Wang et al., 2023) regard TTS as a conditional language modeling task and use an autoregressive and an additional non-autoregressive model for discrete token generation.

- **VALL-E 2** (Chen et al., 2024a), based on VALL-E, introduces Repetition Aware Sampling to stabilize the decoding process and proposes the Grouped Code Modeling to effectively address the challenges of long sequence modeling.

- **VoiceBox** (Matthew et al., 2023) is a non-autoregressive flow-matching model designed to infill mel-spectrograms based on provided speech context and text. We obtained the samples by contacting the authors.

- **StyleTTS 2** (Li et al., 2024) models styles as a latent random variable through diffusion models to generate the most suitable style for the text and employ large pre-trained speech language models as discriminators with novel differentiable duration modeling for end-to-end training. We use the official code and pretrained weights[3].

- **HierSpeech++** (Lee et al., 2023) designs a hierarchical speech synthesis frameworks that significantly improve the robustness and expressiveness of the synthetic speech. We use the official code and pretrained weights[4]. We do not use its speech super-resolution model for fair comparison.

- **UniAudio** (Yang et al., 2023b) utilizes a multi-scale Transformer model to handle the overly long sequences caused by the residual vector quantization-based neural codec in tokenization. We obtained the samples by contacting the authors.

- **Mega-TTS 2** (Jiang et al., 2024) designs an acoustic autoencoder that separately encodes the prosody and timbre information into the compressed latent space and proposes a multi-reference timbre encoder and a prosody latent language model to extract useful information from multi-sentence prompts. We obtained the samples by contacting the authors.

- **ARDiT** (Liu et al., 2024b) proposes to encode audio as vector sequences in continuous space and autoregressively generate these sequences using a decoder-only diffusion transformer (DiT). We obtained the samples by contacting the authors.

- **DiTTo-TTS** (Lee et al., 2024a) addresses the challenge of text-speech alignment via cross-attention mechanisms with the prediction of the total length of speech representations. We directly obtain the results of objective evaluations from their paper.

- **NaturalSpeech 3** (Ju et al., 2024) designs a neural codec with factorized vector quantization (FVQ) to disentangle speech waveform into subspaces of content, prosody, timbre, and acoustic details and propose a factorized diffusion model to generate attributes in each subspace following its corresponding prompt. We obtained the samples by contacting the authors.

- **CosyVoice** (Du et al., 2024) utilizes an LLM for text-to-token generation and a conditional flow matching model for token-to-speech synthesis. We use the official code and the model snapshot named "CosyVoice-300M" in our experiments[5].

The evaluation is conducted on a server with 1 NVIDIA V100 GPU and batch size 1. RTF denotes the real-time factor, i.e., the seconds required for the system (together with the vocoder) to synthesize one-second audio.

### A.5 Details about the Accented TTS Baseline

CTA-TTS (Liu et al., 2024a) is a TTS framework that uses a phoneme recognition model to quantify the accent intensity in phoneme level for accent intensity control. CTA-TTS first trains the phoneme recognition model on the standard pronunciation LibriSpeech dataset, and then uses the output probability distribution of the model to assess the accent intensity and create accent labels on the accented L2Arctic dataset. These labels were input into the TTS model to enable control over accent intensity.

---

[3]https://github.com/yl4579/StyleTTS2
[4]https://github.com/sh-lee-prml/HierSpeechpp
[5]https://github.com/FunAudioLLM/CosyVoice

Systems like CTA-TTS require precise accent annotations during training, so we trained them on the L2-ARCTIC dataset. However, our model does not require accent annotations and learns different accent patterns from large-scale data, using only the multi-condition CFG mechanism to achieve accent intensity control. Therefore, we directly compare the zero-shot results of our model with the baselines, which is a more challenging task.

## A.6 DETAILS IN SUBJECTIVE EVALUATIONS

We conduct evaluations of audio quality, speaker similarity, and accent similarity on Amazon Mechanical Turk (MTurk). We inform the participants that the data will be utilized for scientific research purposes. For each dataset, 40 samples are randomly selected from the test set, and the TTS systems are then used to generate corresponding audio samples. Each audio sample is listened to by a minimum of 10 listeners. For CMOS, following the approach of Loizou (2011), listeners are asked to compare pairs of audio generated by systems A and B and indicate their preference between the two. They are then asked to choose one of the following scores: 0 indicating no difference, 1 indicating a slight difference, 2 indicating a significant difference and 3 indicating a very large difference. We instruct listeners to "*Please focus on speech quality, particularly in terms of clarity, naturalness, and high-frequency details, while disregarding other factors*". For SMOS and ASMOS, each participant is instructed to rate the sentence on a 1-5 Likert scale based on their subjective judgment. For speaker similarity evaluations (SMOS), we instruct listeners to "*Please focus solely on the timbre and prosodic similarity between the reference speech and the generated speech, while disregarding differences in content, grammar, audio quality, and other factors*". For accent similarity evaluations (ASMOS), we instruct listeners to "*Please focus solely on the accent similarity between the ground-truth speech and the generated speech, while disregarding other factors*". The screenshots of instructions for testers are shown in Figure 5. Additionally, we insert audio samples with known quality levels (e.g., reference recordings with no artifacts or intentionally corrupted audio with noticeable distortions) into the evaluation set to verify whether evaluators are attentive and professional. We also randomly repeat some audio clips in the evaluation set to check whether evaluators provide consistent ratings for the same sample. If large deviations in scores (larger than 1.0) for repeated clips occurs, we will select a new rater to evaluate this audio clip. We paid $8 to participants hourly and totally spent about $500 on participant compensation.

## A.7 DETAILS IN OBJECTIVE EVALUATIONS

In zero-shot TTS experiments, we carefully follow the experimental setup of NaturalSpeech 3 (Ju et al., 2024) to ensure fair comparisons. The LibriSpeech test-clean set contains 40 distinct speakers and 5.4 hours of speech. We randomly select one sentence for each speaker for LibriSpeech test-clean benchmark. To construct the prompt-target pairs, we randomly extract 3-second clips as prompts from the same speaker's speech.

However, 40 samples may not be sufficient enough to determine the actual SIM-O and WER of the model. Therefore, we also conduct experiments on the LibriSpeech test-clean 2.2-hour subset (following the setting in VALL-E 2 and Voicebox), the results are shown in the following Table.

Table 9: Comparisons on the LibriSpeech test-clean 2.2-hour subset.

| Model - with Longer Samples | WER↓ | SIM-O↑ |
| --- | --- | --- |
| VALL-E 2 | 2.44% | 0.643 |
| MELLE | 2.10% | 0.625 |
| DiTTo-TTS | 2.56% | 0.627 |
| Voicebox | 1.9% | 0.662 |
| S-DiT | **1.87%** | **0.697** |

## B CLASSIFIER-FREE GUIDANCE USED IN ZERO-SHOT TTS

Classifier-Free Guidance (CFG) (Ho & Salimans, 2022) is a technique that balances sample fidelity and mode coverage in diffusion models by combining the score estimates from both a conditional

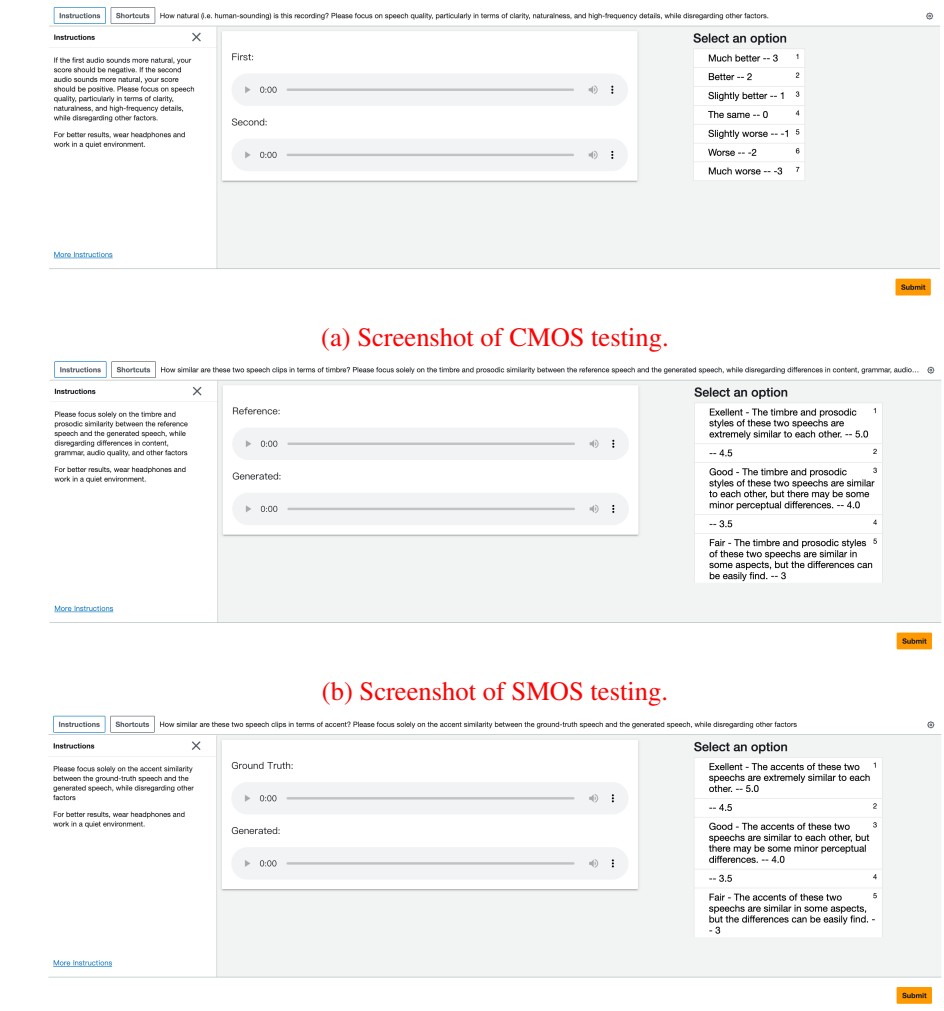

(a) Screenshot of CMOS testing.

(b) Screenshot of SMOS testing.

(c) Screenshot of ASMOS testing.

Figure 5: Screenshots of subjective evaluations.

and an unconditional model. The unconditional model is trained alongside the conditional model by randomly omitting the conditioning variable $c$ with a certain probability, allowing the same model to provide score estimates for both $p(x)$ and $p(x|c)$. In large-scale zero-shot TTS, VoiceBox (Matthew et al., 2023) and NaturalSpeech 2 (Shen et al., 2023) achieve CFG mechanism by dropping the text and prompt speech features. However, these works overlook that text and timbre should be controlled separately. Inspired by VoiceLDM (Lee et al., 2024b) that introduces separate control of environmental conditions and speech contents, a concurrent work (Yang et al., 2024e) proposes separately controlling the speaker fidelity and text intelligibility. However, this work is limited to improving the audio quality of TTS and does not explore the impact of CFG on accent.

## C  DETAILS OF PERFLOW TRAINING PROCEDURE

Once the pretrained ODE solver of the teacher model $\phi_\theta$ is available, we perform the PeRFlow technique to train an accelerated solver in real time. When training, we only consider the shortened segments of the ODE trajectories, reducing the computational load of inference for the teacher model at each training step, and accelerating the training process.

At each training step, given a data sample $z_1$ and a sample $z_0$ drawn from the source distribution (in this case, $z_0 \sim \mathcal{N}(0, I)$, i.e., Gaussian distribution), we randomly select a time window $(t_{k-1}, t_k]$ and compute the standpoint of the segmented probability path $z_{t_{k-1}} = \sqrt{1 - \sigma^2(t_{k-1})}z_1 + \sigma(t_{k-1})z_0$, where $K$ is a hyperparameter indicating the total number of segments, $k \in \{1, \cdots, K\}$, $t_k = k/K$, and $\sigma(t)$ is the noise schedule. The teacher solver only needs to infer the endpoint of this segmented path, $\hat{z}_{t_k} = \phi_\theta(z_{t_{k-1}}, t_{k-1}, t_k)$, with a remarkably smaller number of iterations $\widehat{T}$, comparing to that of a full trajectory, $T$. Finally, the student model is optimized on the segmented trajectory from $z_{t_{k-1}}$ to $\hat{z}_{t_k}$. We set $T$ to 25 and $\widehat{T}$ to 8, achieving a non-negligible acceleration of the training process.

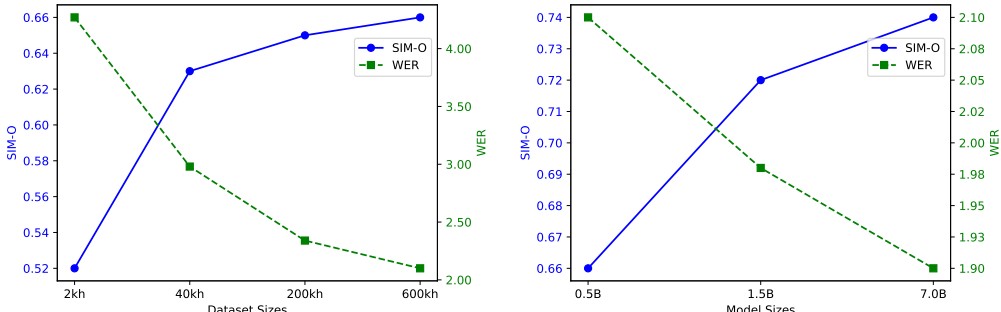

Figure 6: Data scaling results.     Figure 7: Model scaling results.

## D DETAILS ABOUT DATA AND MODEL SCALING EXPERIMENTS

We visualize the experimental results of data and model scaling in Figure 6 and Figure 7. The details are as follows:

**Training Corpus.** The data/model scalability is crucial for practical TTS systems. To evaluate the scalability of S-DiT in Section 4.5, we construct a 600kh internal multilingual training corpus comprising both English and Chinese speech. Most of the audiobook recordings are crawled from YouTube and online podcasts like novelfm[6]. We also include the academic datasets like LibriLight (Kahn et al., 2020), WenetSpeech (Zhang et al., 2022), and GigaSpeech (Chen et al., 2021). Since the crawled corpus may contain unlabelled speeches. We transcribe them using an internal ASR model.

**Test Set.** Most prior studies of zero-shot TTS evaluate performances using the reading-style LibriSpeech test set, which may be different from real-world speech generation scenarios. In section 4.5, we evaluate our model using the test sets collected from various sources, including: 1) CommonVoice (Ardila et al., 2019), a large voice corpus containing noisy speeches from various scenarios; 2) RAVDESS (Livingstone & Russo, 2018), an emotional TTS dataset featuring 8 emotions and 2 emotional intensity. We follow Ju et al. (2024) and use strong-intensity samples to validate the model's ability to handle emotional variance; 3) LibriTTS (Zen et al., 2019), a high-quality speech corpus; 4) we collect samples from videos, movies, and animations to test whether our model can simulate timbres with distinctly strong individual characteristics. The test set consists of 40 audio samples extracted from each source.

**Model Scaling.** In Section 4.5, we scale up S-DiT from 0.5B to 7.0B following the hyper-parameter settings in Qwen 2 (Yang et al., 2024a). In this experiment, we only increase the parameters of the S-DiT model to verify its scalability. The parameters of the speech compression VAE remained unchanged. In theory, expanding the parameters of both models could yield the optimal results, which we leave for future work.

**Speech-Text Alignment Labels for Large-Scale Data.** Training an MFA model directly on a 600k-hour dataset is impractical. Therefore, we randomly sampled a 10k-hour subset from the dataset

---

[6] https://novelfm.changdunovel.com/

to train a robust MFA model, which is then used to align the full dataset. Since data processing inherently requires some alignment model (such as an ASR model) for speech segmentation, using a pretrained MFA model for alignment extraction does not limit the system's data scalability.

## E  DETAILS ABOUT F-LM

**Special Tokens**   We add special tokens <Begin of BPE> and <End of BPE> at the beginning and end of the BPE sequence to indicate the start and end of the BPE sequence. We also add <EOS> token to the phoneme/timestamp sequence to indicate the end of the sentence. In training, we add special tokens <Full> or <Partial> to the input sequence depending on whether we discard parts of the speech encoder output, respectively. Through this strategy, the model given the <Full> token is constrained to generate only up to the text corresponding to the speech prompt, which is used by the ASR process.

**Training Loss**   We use the cross-entropy loss computed solely for the BPE and phoneme/timestamp sequences as the training loss for F-LM. Initially, we train for 500k steps on the ASR task to ensure F-LM's speech understanding capability. After that, we conduct multi-task training for an additional 500k steps.

**Speech-Text Alignment Labels**   Since MFA requires a significant amount of CPU power during the alignment process, we are unable to obtain all the alignment labels for the entire LibriLight dataset at once for training F-LM. We divided the LibriLight dataset into several 5k-hour subsets and used MFA on each subset separately to obtain the alignment labels. As shown in Section 4.4, the alignment accuracy of F-LM surpasses the teacher MFA model, demonstrating that the large-scale training and unified multi-task training significantly improve the robustness and generalization of models.

**Additional Experiment**   In this section, we evaluated the impact of the unified frontend language model (F-LM) compared to the cascaded frontend model on the synthesized speeches. We introduce a baseline frontend system composed of the Whisper-small, a grapheme-to-phoneme conversion module, and an AR-based duration predictor. For this experiment, we use the 7.0B version of S-DiT trained on the 600k-hour dataset. The results, shown in Table 10, indicate that the WER of F-LM is lower than that of the baseline system, demonstrating that the unified system can effectively reduce cascaded errors.

Table 10: Ablation studies of the unified frontend and the cascaded frontend model.

| Frontend Systems | SIM-O↑ | WER↓ | CMOS↑ | SMOS↑ |
|---|---|---|---|---|
| Cascaded | 0.73 | 2.02% | -0.06 | 4.14 |
| F-LM | **0.74** | **1.90%** | **0.00** | **4.15** |

## F  DURATION CONTROLLABILITY OF S-DIT

In this section, we aim to verify S-DiT's duration control capabilities through case studies. We randomly selected a speech prompt from the test set and used the sentence "Notably, raising questions about both the size of the perimeter and efforts to sweep and secure." as the target sentence to generate speeches. In the generation process, we first control the sentence-level duration by multiplying the time coordinates of the phoneme anchors described in Section 3.2 by a fixed value. As shown in Figure 8, our S-DiT demonstrates good sentence-level duration control. Moreover, our S-DiT is also capable of fine-grained phoneme-level duration control. As illustrated in Figure 9, we multiplied the anchor coordinates of the phoneme within the red box by a fixed value while keeping the relative positions of other phoneme anchors unchanged. The figure shows that our S-DiT also exhibits good fine-grained phoneme-level duration controllability.

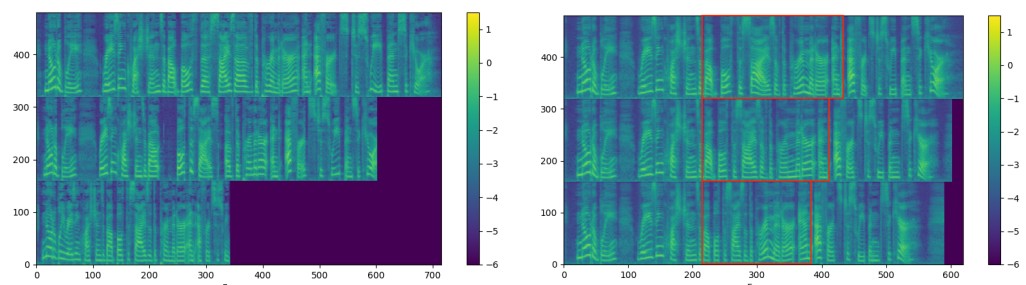

Figure 8: Sentence-level duration control.    Figure 9: Phoneme-level duration control.

## G    VISUALIZATION OF ATTENTION MATRICES

We visualize the attention matrices from all layers in the 1.4B S-DiT model, using 8 sampling steps. From Figure 10, we observe: 1) within the same layer, despite different timesteps, the attention matrices remain identical. In other words, the function of each layer stays consistent across timesteps; 2) the functions of the transformer layers can be categorized into three types. As shown in Figure 10 (a), the bottom layers handle text and audio feature extraction; in Figure 10 (b), the middle layers focus on speech-text alignment; and in Figure 10 (c), the top layers refine the target latent features.

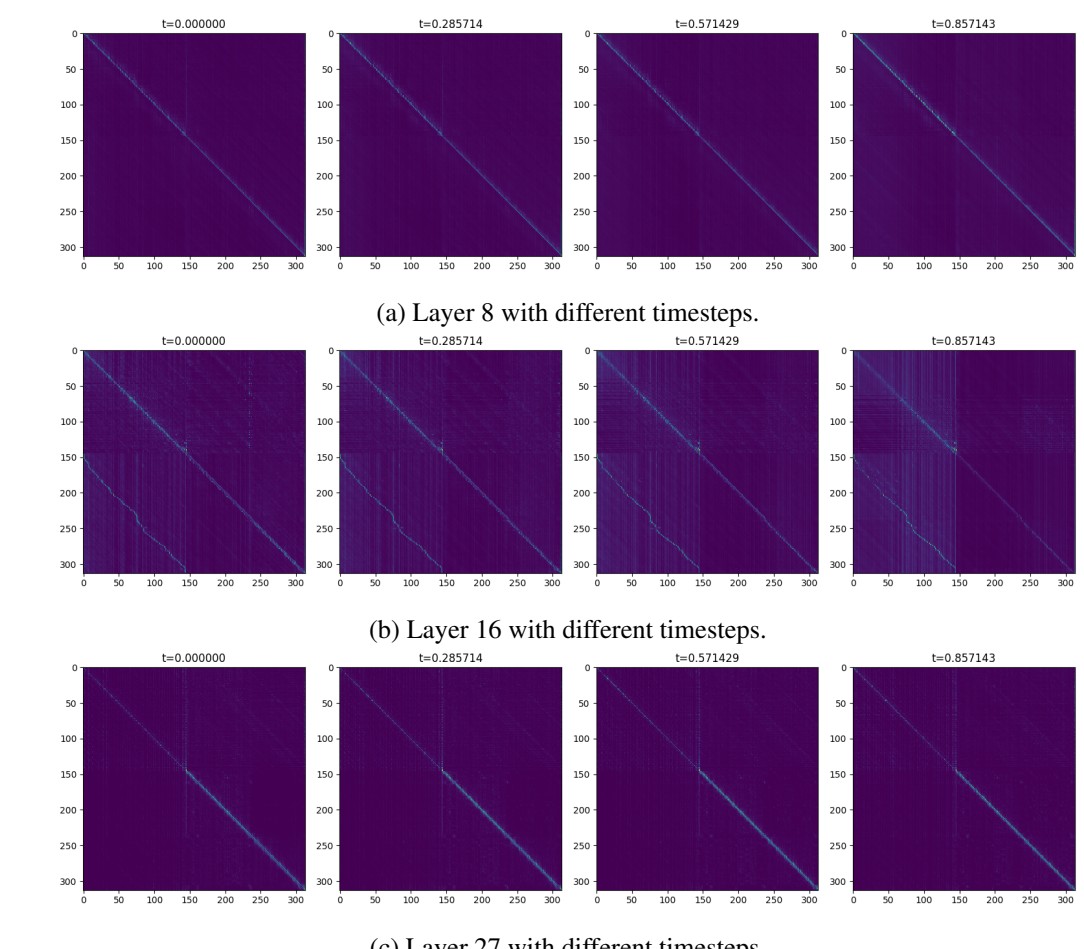

(a) Layer 8 with different timesteps.

(b) Layer 16 with different timesteps.

(c) Layer 27 with different timesteps.

Figure 10: Visualization of Attention Matrices from different layers in S-DiT.

## H    ABOUT DIFFERENT LENGTHS OF CONTEXT

An imbalanced distribution of prompt and target lengths during training can lead to unstable generation performance during inference. For example, if the majority of the sampled data during training consists of 20-second targets, the generation performance for audio with a 40-second target will be worse than that of 20-second targets in inference. To solve the imbalanced distribution issue, we recommend using the following multi-sentence data sampling strategy: we concatenate all audio recordings of the same speaker in the dataset in time order, and then randomly extract audio segments of length $t \sim U(t_{min}, t_{max})$ from the concatenated audio, where $t_{min}$ is the minimum sampling time and $t_{max}$ is the maximum sampling time. Then, following Section 3.1, we randomly divide the sampled sequence into a prompt region and a target region. Although we do not use this strategy in our experiments in order to make a fair comparison with other methods, this strategy is effective in practical scenarios.

## I    LIMITATIONS AND FUTURE WORKS

In this section, we discuss the limitations of the proposed method and outline potential strategies for addressing them in future research.

- **Language Coverage.** Although our model currently supports both English and Chinese, there are far more languages in the world. In particular, for some low-resource languages, the performance of our model requires further validation. To address this, we plan to incorporate additional training data from a wider range of languages and apply adaptation-based techniques, such as LoRA tuning (Hu et al., 2021), to enhance speech quality for low-resource languages.
- **Function Coverage.** We can make S-DiT more user-friendly by enabling it to generate speech in various styles according to text descriptions through instruction-based fine-tuning. We can further fine-tune S-DiT on the paralinguistic corpus, allowing it to generate speech that is closer to a natural human style.
- **Frontend Coverage.** While our current F-LM supports four key tasks (ASR, MFA, duration prediction, and G2P), there are additional tasks in the TTS data preprocessing pipeline, such as speech enhancement, speaker diarization, and emotion classification, that remain to be included. In the future, we aim to design a truly universal frontend language model capable of efficiently handling all speech data processing tasks for TTS, thereby simplifying the overall workflow.

## J    EVALUATION OF THE SPEECH COMPRESSION MODEL

In this section, we conduct evaluations of the speech compression model's impact on the overall system. First, we evaluate the reconstruction quality of the speech compression model, with results presented in Table 11. We report the objective metrics, including Perceptual Evaluation of Speech Quality (PESQ), Virtual Speech Quality Objective Listener (ViSQOL), and Mel-Cepstral Distortion (MCD). We select the following codec models as baselines: EnCodec (Défossez et al., 2022), HiFi-Codec (Yang et al., 2023a), Descript-Audio-Codec (DAC) (Kumar et al., 2024), and SoundStream (Zeghidour et al., 2021). To ensure fair comparisons under the 16kHz setting, we reproduce the 5 kbps EnCodec model following the hyperparameter configuration of 5 kbps Encodec reproduced in NaturalSpeech 3 (Ju et al., 2024). The results demonstrates that, despite applying an additional 8x compression in the temporal dimension, our speech compression model's performance on various reconstruction metrics, such as PESQ and ViSQOL, remains close to that of the Encodec model, due to the use of continuous representations and a slight KL-penalty loss during training. Moreover, it even significantly outperforms all baseline models in the MCD metric.

Second, in terms of the zero-shot TTS performance resulting from each speech compression method, we report the experimental results below. It can be seen that although the reconstruction quality of DAC is better than our speech compression model, S-DiT outperforms "w/ DAC", due to the fact that the latent space of our speech compression model is more compact (only 1 layer with 8x time-axis compression). This conclusion is also verified by a previous work, DiTTo-TTS (Lee et al., 2024a), which shows compact target latents facilitate learning in diffusion models.

Table 11: Comparison of the reconstruction quality. $\star$ denotes the reproduced results. Underline means that results are infered from offical checkpoints. The sampling rate are set to 16 kHz.

| Models | Hop Size | Latent Layer | Type | Bandwidth | PESQ↑ | ViSQOL↑ | MCD↓ |
|---|---|---|---|---|---|---|---|
| EnCodec$^\star$ | 320 | 10 | Discrete | 5.0 kbps | 3.10 | 4.27 | 3.10 |
| HiFi-Codec | 320 | 4 | Discrete | 2.0 kbps | 3.17 | 4.19 | 3.05 |
| DAC | 320 | 9 | Discrete | 4.5 kbps | **3.52** | **4.54** | 2.65 |
| SoundStream$^\star$ | 200 | 6 | Discrete | 4.8 kbps | 3.01 | 4.16 | 3.36 |
| Ours | 200 (x8) | 1 | Continuous | - | 3.06 | 4.31 | **2.47** |

Table 12: Comparison of zero-shot TTS performance with different speech compression models.

| Setting | SIM-O↑ | WER↓ |
|---|---|---|
| Ours | **0.67** | **1.84%** |
| w/ Encodec | 0.56 | 2.24% |
| w/ DAC | 0.64 | 1.93% |

## K  AVERAGE FRONTEND PROCESSING TIME COMPARISONS

To evaluate the efficiency gains achieved by our F-LM, we compare its processing time with that of a traditional frontend pipeline, which is required by *Diffusion w/ PA* models like NaturalSpeech 3. The traditional pipeline consists of an ASR model (SenseVoice small (An et al., 2024)), a phonemizer (Bernard & Titeux, 2021), a speech-text aligner (MFA), and an auto-regressive duration predictor (Yang et al., 2024b; Jiang et al., 2024). Since F-LM decodes phoneme and duration tokens simultaneously, we divide the decoding time equally into two parts to represent the time required for each. We report the average processing time per speech clip based on the experiments in Section 4.2. The results, shown in Table 13, indicate that our model achieves a 5.1x speed-up by significantly reducing the computational time required by speech-text aligning. It is noteworthy that no additional acceleration techniques are applied to F-LM in this experiment. In practical applications, since the entire frontend pipeline is unified within a single language model, further acceleration can be achieved through techniques like automatic mixed precision or leveraging the parallel capabilities of GPUs.

Notably, alternatives like training a GPU-compatible aligner (e.g., MAS from Glow-TTS (Kim et al., 2020)) or using a duration predictor to add alignments to ASR outputs (e.g., WhisperX (Bain et al., 2023)) could be faster in speech-text aligning than F-LM. However, as demonstrated by Rousso et al. (2024), MFA significantly outperforms WhisperX in terms of alignment accuracy. Since our F-LM also outperforms MFA, the alignment accuracy of F-LM is a significant advantage, despite being slightly slower.

Table 13: Comparison of processing time for each frontend module in seconds.

| Frontend | ASR↓ | Speech-Text Aligning↓ | Phonemization↓ | Duration Prediction↓ | Total↓ |
|---|---|---|---|---|---|
| Traditional Pipeline | 0.69 | 24.10 | **0.08** | 1.86 | 26.73 |
| F-LM | **0.62** | **2.29** | 1.16 | **1.16** | **5.23** |

## L  LOSS WEIGHTS FOR BPE OF F-LM

The loss for $t$ in Section 3.3 that is not from the speech prompt can be regarded as the text-modality language modeling task. We have conducted experiments with three loss weights for the parts of $t$ that are not from the speech prompt: $\{0, 0.01, 1.0\}$. The results are shown in Table 14 and Table 15. When the weight is set to 0.01, the performance of duration prediction shows improvement, suggesting that learning textual information can guide the prediction of prosodic information. When the weight is set to 1.0, however, the increased difficulty of training a text-only LM might affect the duration prediction task. Nevertheless, the difference in weights does not significantly impact the alignment accuracy, possibly because the alignment is already precise enough, leaving limited room for improvement.

These observations are aligned with the perspectives in BASE-TTS (Łajszczak et al., 2024), which adopts the text-only loss with a small weight for SpeechGPT to retain textual information and guide prosody learning.

Table 14: Duration accuracy comparison with different $\lambda_w$. $\Delta_p$ denotes the absolute boundary difference of phonemes. $\lambda_w$ denotes the loss weight for the parts of $t$ that is not from the speech prompt.

| $\lambda_w$ | $\Delta_p$ (ms) |
|---|---|
| 0.00 | $18.72 \pm 0.91$ |
| 0.01 | $\mathbf{18.52 \pm 0.86}$ |
| 0.10 | $18.65 \pm 0.90$ |
| 1.00 | $18.80 \pm 0.94$ |

Table 15: Results for speech-text aligning with different $\lambda_w$. $\Delta_p$ means the absolute alignment boundary difference of phonemes. $\lambda_w$ denotes the loss weight for the parts of $t$ that is not from the speech prompt.

| $\lambda_w$ | $\Delta_p$ (ms) |
|---|---|
| 0.00 | $8.81 \pm 0.57$ |
| 0.01 | $\mathbf{8.76 \pm 0.60}$ |
| 0.10 | $8.81 \pm 0.58$ |
| 1.00 | $8.79 \pm 0.59$ |

## M  ADDITIONAL DETAILS FOR MULTI-CONDITION CFG

In Section 3.2, regarding the multi-condition CFG technique, the experimental setup for the preliminary experiment for accent control is: fixing $\alpha_{spk}$ at 2.5 and varying $\alpha_{txt}$ from 1.0 to 6.0. Specifically, as $\alpha_{txt}$ increases from 1.0 to 1.5, the generated speeches contains improper pronunciations and distortions. When $\alpha_{txt}$ ranges from 1.5 to 2.5, the pronunciations align with the speaker's accent. Finally, once $\alpha_{txt}$ exceeds 4.0, the generated speech converges toward the standard pronunciation of the target language.

## N  EXPERIMENTS OF PROSODIC NATURALNESS FOR ZERO-SHOT TTS

To validate whether sparse alignment enhances prosodic naturalness, in this section, we evaluate the moments (standard deviation ($\sigma$), skewness ($\gamma$), and kurtosis ($\kappa$)) of pitch and duration distributions. The results are presented in the Table 16 and Table 17. Compared to NaturalSpeech 3, the results of "Ours w/ Sparse Alignment" are closer to the reference speeches. Besides, although both "Ours w/ Sparse Alignment" and "Ours w/ Forced Alignment" use the same durations predicted by F-LM, the performance of "Ours w/ Sparse Alignment" surpasses that of "Ours w/ Forced Alignment". This demonstrates that the proposed sparse alignment strategy offers superior prosodic naturalness than forced alignment based methods.

Table 16: Comparisons about the moments of pitch distribution. $\sigma$, $\gamma$, and $\kappa$ are the standard deviation, skewness, and kurtosis of the pitch distribution.

| Model | $\sigma$ | $\gamma$ | $\kappa$ |
|---|---|---|---|
| Reference | 80.75 | 0.36 | -0.81 |
| NaturalSpeech 3 | 87.38 | 0.49 | -0.66 |
| Ours w/ Forced Alignment | 88.17 | 0.44 | -0.96 |
| Ours w/ Sparse Alignment | **81.90** | **0.39** | **-0.91** |

We also measure the objective metrics MCD, SSIM, STOI, GPE, VDE, and FFE following InstructTTS (Yang et al., 2024c) to evaluate the expressiveness of our method. The test set uses the

Table 17: Comparisons about the moments of duration distribution. $\sigma$, $\gamma$, and $\kappa$ are the standard deviation, skewness, and kurtosis of the duration distribution.

| Model | $\sigma$ | $\gamma$ | $\kappa$ |
|---|---|---|---|
| Reference | 7.74 | 3.40 | 16.39 |
| NaturalSpeech 3 | 7.52 | 5.96 | 62.98 |
| Ours w/ Forced Alignment | 7.48 | 6.30 | 54.01 |
| Ours w/ Sparse Alignment | **7.83** | **4.84** | **31.23** |

same objective evaluation set provided by the authors of NaturalSpeech 3, consisting of 40 samples. The results in Table 18 demonstrate that our method achieves superior performance than the two baselines based on forced alignment.

However, 40 samples may not be sufficient to convincingly verify the effectiveness of our method. To further evaluate the actual performance of the model, we conduct experiments on the LibriSpeech test-clean 2.2-hour subset (following the setup in VALL-E 2 and Voicebox). The results are shown in the Table below. We compare S-DiT with the following baselines: 1) "Ours w/ Forced Alignment", we replace the sparse alignment with the forced alignment; 2) "Ours w/ Standard CFG", we replace the multi-condition CFG with standard CFG; 3) "Ours w/ Standard AR Duration", we replace the duration from F-LM with the duration from standard AR duration predictor following SimpleSpeech 2 (Yang et al., 2024b). The results in Table 19 show that sparse alignment brings significant improvements, and both multi-condition CFG and F-LM duration contribute positively to the performance.

Table 18: Comparisons about "expressiveness" metrics for 40 samples.

| Method | MCD↓ | SSIM↑ | STOI↑ | GPE↓ | VDE↓ | FFE↓ |
|---|---|---|---|---|---|---|
| GT | - | - | - | - | - | - |
| NaturalSpeech 3 | 4.45 | 0.46 | 0.62 | 0.44 | 0.33 | 0.37 |
| Ours w/ Forced Alignment | 4.48 | 0.44 | **0.63** | 0.44 | 0.35 | 0.40 |
| Ours w/ Sparse Alignment | **4.42** | **0.50** | **0.63** | **0.31** | **0.29** | **0.34** |

Table 19: Comparisons about "expressiveness" metrics on the LibriSpeech test-clean set.

| Method | MCD↓ | SSIM↑ | STOI↑ | GPE↓ | VDE↓ | FFE↓ |
|---|---|---|---|---|---|---|
| GT | - | - | - | - | - | - |
| Ours w/ Sparse Alignment | **4.56** | **0.52** | **0.62** | **0.34** | **0.30** | **0.35** |
| Ours w/ Forced Alignment | 4.62 | 0.45 | **0.62** | 0.42 | 0.34 | 0.40 |
| Ours w/ Standard CFG | 4.59 | 0.51 | 0.61 | 0.36 | 0.32 | 0.37 |
| Ours w/ Standard AR Duration | 4.58 | 0.50 | **0.62** | 0.36 | 0.31 | 0.36 |

## O    EXPERIMENTS WITH LONGER SAMPLES

To directly compare S-DiT's robustness to long sequences against other AR models, we have conducted experiemnts for a test set with longer samples. Specifically, we randomly select 10 sentences, each containing more than 50 words. For each speaker in the LibriSpeech test-clean set, we randomly chose a 3-second clip as a prompt, resulting in 400 target samples in total. To make our results more convincing, we include strong-performing TTS models, VoiceCraft (Peng et al., 2024) and CosyVoice (AR+NAR) (Du et al., 2024), as our baselines. The results for longer samples are presented in Table 20. As shown, compared to the baseline systems, S-DiT does not exhibit a significant decline in speech intelligibility when generating longer sentences, illustrating the effectiveness of the combination of F-LM and S-DiT.

Table 20: Comparisons with longer samples.

| Model - with Longer Samples | WER↓ | SIM-O↑ |
|---|---|---|
| VoiceCraft | 12.81% | 0.62 |
| CosyVoice | 5.52% | 0.68 |
| S-DiT | **2.39%** | **0.70** |

| Model - with Single-Sentence Samples | WER↓ | SIM-O↑ |
|---|---|---|
| CosyVoice | 4.07% | 0.58 |
| VoiceCraft | 2.24% | 0.62 |
| S-DiT | **1.84%** | **0.67** |

## P  EXPERIMENTS WITH HARD SENTENCES

The transcriptions on the LibriSpeech test-clean set are relatively simple since they come from audiobooks. To further indicate the speech intelligibility of different methods, we evaluate our model on the challenging set containing 100 difficult textual patterns from ELLA-V (Song et al., 2024). Since the speech prompts used by ELLA-V are not publicly available, we randomly sample 3-second-long speeches in the LibriSpeech test-clean set as speech prompts. For this evaluation, we used the official checkpoint of F5-TTS (Chen et al., 2024b) and the E2-TTS (Eskimez et al., 2024) inference API provided on F5-TTS's Hugging Face page. We employ Whisper-large-v3 for WER calculation. Based on the results presented in Table 21, our model shows stronger robustness against hard transcriptions.

Table 21: Comparisons with hard sentences.

| Model | WER↓ | Substitution↓ | Deletion↓ | Insertion↓ |
|---|---|---|---|---|
| E2-TTS | 8.49% | 3.65% | 4.75% | 0.09% |
| F5-TTS | 4.28% | **1.78%** | 2.28% | 0.22% |
| S-DiT | **3.95%** | 1.80% | **2.07%** | **0.08%** |

## Q  END PREDICTION OR BINARY APPROACH

As described in Appendix E, we use the <Full> token to constrain the model to generate only up to the text corresponding to the speech prompt, which is used by the ASR process. This approach simplifies the task to a binary decision of whether to generate up to the end or not. However, the end prediction is also a possible way to solve this issue. We finetune the pretrained F-LM for 100k steps to incorporate the end-prediction mode. The ASR performance are shown in Table 22. It can be seen that the WER of "F-LM w/ End Prediction" is slightly higher. When analyzing specific error cases, we found that in the end-prediction mode, inaccurate prediction of the end token can also impact the model's performance.

Table 22: Ablation study for F-LM's ASR performance.

| Setting | test-clean (WER)↓ | test-other (WER)↓ |
|---|---|---|
| F-LM w/ Binary Approach | **4.2%** | **8.3%** |
| F-LM w/ End Prediction | 4.9% | 11.8% |

