# OpenReview forum: "Sparse Alignment Enhanced Latent Diffusion Transformer for Zero-Shot Speech Synthesis"
_ICLR.cc/2025/Conference — ICLR 2025 Conference Withdrawn Submission_

### Official Review · Reviewer_syxD · 2024-11-02

**Soundness:** 2
**Presentation:** 2
**Contribution:** 2
**Rating:** 6
**Confidence:** 4

**Summary:**

The paper presents S-DiT, a zero-shot text-to-speech system that addresses speech-text alignment issues by using a sparse alignment algorithm to guide a latent diffusion transformer. By providing sparse alignment boundaries and simplifying the pipeline with a unified frontend language model, S-DiT reduces alignment learning difficulty without limiting expressiveness.

**Strengths:**

The paper addresses a clear problem in zero-shot text-to-speech systems by introducing S-DiT, a method that employs a sparse alignment algorithm to guide a latent diffusion transformer, reducing the difficulty of alignment learning without limiting expressiveness. Multi-condition CFG looks promising for cross-lingual settings. The authors have conducted most of the necessary experiments to demonstrate their claims.

**Weaknesses:**

Quality:
- The paper presents sparse alignment as a major contribution, but its impact is not clearly demonstrated. While experiments show improvements over the "Diffusion w/o PA" model, this is expected since models like ns3 already outperform it. This raises questions about the specific benefits of sparse alignment, especially since it requires similar preprocessing efforts but appears to omit some information to achieve sparsity. Providing detailed comparisons with ns3 and illustrating how sparse alignment enhances aspects like expressiveness would clarify its advantages. Currently, multi-condition CFG and multi-task learning F-LM seem to offer more substantial advancements independently of sparse alignment, which may not align with the paper's intended focus.
- Since F-LM is also an autoregressive model, it would strengthen the paper to directly compare its robustness to long sequences against other AR models. For instance, does the combination of F-LM and S-DiT exhibit less performance degradation as sequence length increases compared to traditional AR approaches?
- In line 155, the paper claims improvements in efficiency. However, the required data preprocessing steps (phonemization, MFA, duration prediction, etc.) remain the same and have been unified, yet inference still relies on an inefficient AR model. To convincingly demonstrate efficiency gains, it would be helpful to measure and compare the average frontend processing time of the "Duration w/ PA" model, similar to how S-DiT's frontend time is measured in line 412. This comparison would offer concrete evidence of any efficiency improvements and validate the claim of enhanced efficiency.
- Regarding the Speech Compression module in Section 3.1, the reconstruction performance has not been clearly verified. Given the existence of various established neural codecs (e.g., DAC [1], EnCodec [2]) and recent advancements in neural vocoding (e.g., BigVGAN [3]), it would be pertinent to explain why the paper adopts a structure based on image processing techniques. Since the performance of the speech compressor sets an upper bound on the final TTS quality, evaluating its reconstruction performance is crucial for properly assessing S-DiT. Comparing the performance of different speech compression methods, including DAC and EnCodec, using objective metrics like PESQ or ViSQOL, and evaluating the TTS performance resulting from each method would provide a comprehensive understanding of the speech compression's impact on the overall system.

Clarity:

- The boldface in each table does not have a consistent meaning. Please standardize it according to a consistent criterion (e.g., highlighting the highest-performing values).
- In line 91, please specify exactly what is meant by "certain commonalities."
- In line 101, please clarify what the "interesting conclusions" are.
- In line 249, for "Piecewise Rectified Flow Acceleration," please indicate at the beginning of the paragraph that this represents distillation, meaning that a pretrained model is required.
- In line 335, regarding "1M steps for S-DiT," does this 1 million include both pre-training and distillation? What is the proportion between the two?

[1] Kumar, R., Seetharaman, P., Luebs, A., Kumar, I., & Kumar, K. (2024). High-fidelity audio compression with improved rvqgan. *Advances in Neural Information Processing Systems*, *36*.

[2] Défossez, A., Copet, J., Synnaeve, G., & Adi, Y. (2022). High fidelity neural audio compression. *arXiv preprint arXiv:2210.13438*.

[3] Lee, S. G., Ping, W., Ginsburg, B., Catanzaro, B., & Yoon, S. (2022). Bigvgan: A universal neural vocoder with large-scale training. *arXiv preprint arXiv:2206.04658*.

**Questions:**

- In the demo, are the results for "Zero-Shot TTS" and "Accent Intensity Control" from the 7B model? Why are there no 7B samples in "Incredible Improvement Brought by Scaling"? In "Robustness" and "Code-Switched Generation," is the comparison made with the 0.5B model to match the baseline in terms of model parameters?
- What issues arise if Duration Control is made more extreme? For example, if it's too long, does the voice change or words repeat? If it's too short, does the speech become flat or omit words? I'm curious about the extent of robustness problems.
- It would be helpful to dedicate a separate section in the demo to add samples that clearly showcase the advantages of S-DiT in terms of expressiveness compared to duration w/ PA models like ns3. This would help intuitively understand the main claims of the paper.
- Multi-condition CFG seems particularly useful in code-switching. In the code-switching section of the demo, it would be nice to have a comparison of the results from the three models used in Table 7—Ours, w/ Standard CFG, and w/o CFG.
- Section 3.2
    - In the Sparse Alignment Strategy, the experiments in the paper appear to assume a setting with only one anchor for rough alignment. Is there a trade-off between expressiveness and robustness depending on the number of anchors? Is having one anchor the optimal choice in this context?
    - In the Multi-condition CFG, regarding the experiments mentioned in line 273 that examined changes in points 1), 2), and 3), what exactly does this experimental setup entail? Does it refer to fixing \( a_{\text{spk}} \) at a specific value (e.g., 1) in Equation (5) and varying \( a_{\text{txt}} \)?
- Section 3.3
    - When examining the input \( h \), it seems that \( t \) addresses the ASR task, while \( p \) simultaneously tackles alignment, DP, and G2P. In Figure 2, grouping these latter three components together instead of distinguishing them separately might aid in understanding.
    - Is there a particular reason why the speech encoder and language model were not initialized with pretrained weights? Utilizing existing speech compression modules and small LMs for initialization could potentially enhance performance.
    - In line 305, regarding \( t \) (BPE) in the input \( h \), is it necessary to apply a loss to match parts that are not from the speech prompt? Was including this in the multitask learning beneficial for improving performance in alignment, DP, and G2P, prompting its inclusion in training?
    - It is unclear how, during inference, the model is constrained to generate only up to the text corresponding to the speech prompt when the speech prompt is provided. For instance, when randomly discarding the speech encoder output, one might expect the model to align and truncate the text accordingly, insert a speech prompt EOS between the prompt text and target text during training, and then generate only up to that point during inference to concatenate with the given target text. However, the paper does not detail this process.
- Appendix
    - Could the paper provide a more detailed explanation of "the simple scaling of weakly supervised pre-training" mentioned in line 1052? Why is this referred to as weakly supervised pre-training? Additionally, how were the MFAs in Section 4.4 trained?

Typos

- Line 100: In the phrase "expressiveness of 'diffusion w/ PA' with the robustness of 'diffusion w/o PA'," it appears that "w/ PA" and "w/o PA" might be mistakenly swapped.
- Line 112: The phrase "but can also" might be more appropriately written as "but also can."

---

> ### Author Response · Authors · 2024-11-18
> **Response to Reviewer syxD (Part 1/4)**
>
> We are really grateful for your constructive review and valuable feedback, and we hope our response fully resolves your concerns.
>
> **[About Weakness 1]**
> Thank you for your suggestions! We agree that a detailed comparison with NS3 in terms of expressiveness is important for validating the advantages of our sparse alignment. In this evaluation, we examine the moments (standard deviation (σ), skewness (γ), and kurtosis (κ)) of pitch and duration distributions to assess whether our model enhances expressiveness. The results of pitch and duration distributions are presented in the following table, respectively. Compared to NaturalSpeech 3, the results of "Ours w/ Sparse Alignment" are closer to the reference speeches. Besides, although using the same durations predicted by F-LM, the performance of "Ours w/ Sparse Alignment" surpasses that of "Ours w/ Forced Alignment". This demonstrates that the proposed sparse alignment strategy offers superior expressiveness. We have included these results in Appendix N and marked them in red.
>
> |  Model | σ | γ | κ |
> | :------------: | :------------: |:------------: |:------------: |
> | Reference | 80.75 | 0.36 | -0.81 |
> | NaturalSpeech 3  | 87.38 | 0.49 | -0.66 |
> | Ours w/ Forced Alignment | 88.17 | 0.44 | -0.96 |
> | Ours w/ Sparse Alignment | **81.90** | **0.39** | **-0.91** |
>
> |  Model |  σ | γ  | κ |
> | :------------: | :------------: |:------------: |:------------: |
> | Reference  | 7.74 | 3.40 | 16.39 |
> | NaturalSpeech 3  | 7.52 | 5.96 | 62.98 |
> | Ours w/ Forced Alignment | 7.48 | 6.30 | 54.01 |
> | Ours w/ Sparse Alignment | **7.83** | **4.84** | **31.23** |
>
> The introduction of multi-condition CFG aims to improve the overall performance of S-DiT while also introducing accent controllability. Meanwhile, the multi-task learning framework of F-LM not only enhances duration prediction capabilities but also facilitate the inference pipeline of S-DiT, making it more practical. Both of these models are designed with the goal of enhancing the overall effectiveness of S-DiT and they complement the sparse alignment approach outlined in the paper.
>
>
> **[About Weakness 2]**
> We have compared S-DiT's robustness to long sequences against VoiceCraft (an AR model) and included the results in the ``Rebuttal: Robustness to Long Sequences against Other AR Models`` section on the demo page. The results demonstrate that the combination of F-LM and S-DiT indeed exhibits less performance degradation as sequence length increases. Thanks for your advice!
>
> **[About Weakness 3]**
> We compare F-LM's processing time with that of a traditional frontend pipeline, which consists of an ASR model (SenseVoice small [1]), a phonemizer, a speech-text aligner (MFA), and an auto-regressive duration predictor. Since F-LM decodes phoneme and duration tokens simultaneously, we divide the decoding time equally into two parts to represent the time required for each. We report the average processing time per speech clip based on the experiments in zero-shot TTS experiments. The results, shown in the following table, indicate that our model achieves a 5.1x speed-up by significantly reducing the computational time required by speech-text aligning. It is noteworthy that no additional acceleration techniques are applied to F-LM in this experiment. In practical applications, since the entire frontend pipeline is unified within a single language model, further acceleration can be achieved through techniques like TensorRT, automatic mixed precision, or leveraging the parallel capabilities of GPUs (traditional pipelines like MFA can not adopt these techniques). We have included these results in Appendix K.
>
> | Frontend                  | ASR | MFA | Phonemization | Duration Prediction| Total Time |
> | :------------:              | :------------: | :------------: | :------------: | :------------: |  :------------: |
> | Traditional Pipeline  |  0.69 | 24.10 | **0.08** | 1.86 | 26.73 |
> | F-LM                        | **0.62** | **2.29**   | 1.16 | **1.16** | **5.23** |

---

> ### Author Response · Authors · 2024-11-18
> **esponse to Reviewer syxD (Part 2/4)**
>
> **[About Weakness 4]**
> The reason why S-DiT's encoder adopts a structure based on image processing techniques: to ensure training stability, the speech compression model uses mel spectrograms as the input for the encoder. However, models like DAC [2] and Encodec [3] use raw waveforms as input for their encoders, so we do not adopt their encoder architectures. We also experiment with structures like ConvNext and WavNet, but there is no performance gains compared with the structure used by S-DiT.
> Thanks for your helpful suggestions. We have evaluated the reconstruction quality of the speech compression model, with results presented in the following table. Despite applying an additional 8x compression in the temporal dimension, our speech compression model's performance on various reconstruction metrics, such as PESQ and ViSQOL, remains close to that of the Encodec-16kHz model, due to the use of continuous representations and a slight KL-penalty loss during training. Moreover, it even significantly outperforms all baseline models in the MCD metric.
>
> | Models|  Sampling rate | Hop Size | Latent Layer | Type | Bandwidth | PESQ | ViSQOL | MCD |
> | :------------:               | :------------: | :------------: | :------------: | :------------: |  :------------: |  :------------: |  :------------: |  :------------: |
> | EnCodec                   |  16kHz   | 320|  10 | Discrete      | 5.0 kbps |3.10 | 4.27 | 3.10
> | HiFi-Codec               |  16kHz    | 320 | 4 | Discrete      | 2.0 kbps | 3.17 | 4.19 | 3.05
> | DAC                          |  16kHz   | 320 | 9 | Discrete      | 4.5 kbps | **3.52** | **4.54** |  2.65
> | SoundStream            |  16kHz   | 200 | 6 | Discrete      | 4.8 kbps | 3.01 | 4.16  |  3.36
> | Ours                         |  16kHz   | **200 (x8)** | **1** | Continuous | - | 3.06 | 4.31 | **2.47**
>
> In terms of the zero-shot TTS performance resulting from each speech compression method, we report the experimental results below. It can be seen that although the reconstruction quality of DAC is better than ours, "S-DiT" outperforms "S-DiT w/ DAC", due to the fact that the latent space of our speech compression model is more compact (only 1 layer with 8x time-axis compression). We have included these results in Appendix J in the revised version of the paper.
>
> | Models| SIM-O | WER |
> | :------------:                 | :------------: | :------------: |
> | S-DiT                          |  **0.67**   | **1.84%** |
> | S-DiT w/ Encodec        |  0.56  |  2.24% |
> | S-DiT w/ DAC              |  0.64    |  1.93% |
>
>
>
> **[About Clarity]**
> We are sorry for the clarity issues and make the following modifications:
> 1) We have standardized the use of boldface in each table to indicate the highest-performing values;
> 2) "we find that these systems share certain commonalities and can be merged into a unified sequence modeling task" -> "we find that these systems can be merged into a unified language model";
> 3) The "interesting conclusions" are described in Appendix G in the original version of the paper. We have added a link towards Appendix G in line 101;
> 4) We have indicate that PeRFlow represents distillation at the beginning of the corresponding paragraph;
> 5) Yes, 1M steps include both pre-training and distillation. The pre-training requires 800k steps and distillation requires 200k steps.
>
> All of the modifications are marked in red in the revised version of the paper.
>
>
> **[About Question 1]**
> 1) For fair comparisons, the results for "Zero-Shot TTS" and "Accent Intensity Control" are from the 0.5B model;
> 2) We are sorry for the missing 7B samples in ``Incredible Improvement Brought by Scaling`` section and we have added them on the demo page;
> 3) Yes, in ``Robustness`` and ``Code-Switched Generation`` sections, the comparison is made with the 0.5B model.
>
> We have added these details on the demo page. Thanks for your comments!
>
> **[About Question 2]**
> We have generated speech examples with extremely long or short durations and included them into the ``Rebuttal: Examples of Potential Failure Cases`` section on the demo page. The results demonstrate that our S-DiT is as robust as "Diffusion w/ PA" against extreme durations.
>
> **[About Question 3]**
> We have added relevant speech examples in the ``Rebuttal: Advantages of Sparse Alignment in Terms of Expressiveness`` section on the demo page.
>
> **[About Question 4]**
> We have added a comparison of the results for these models in ``Rebuttal: Code-Switched Generation with Different CFG`` section of the demo page.

---

> ### Author Response · Authors · 2024-11-18
> **Response to Reviewer syxD (Part 3/4)**
>
> **[About Question 5]**
> (1) Yes, having one anchor is the optimal choice. To examine the trade-off between expressiveness and robustness depending on the number of anchors, we conduct experiments with three different settings: 1) Ours w/ Forced Alignment, where we use full alignment; 2) Ours w/ Half Alignment, where we randomly chose 50% span from the original alignment region; 3) Ours w/ Sparse Alignment, where we use only one anchor. We report the the moments (standard deviation (σ), skewness (γ) and kurtosis (κ)) of pitch distribution and WER in the following table. As shown, setting 3) is sufficiently robust, while setting 2) is limited in terms of prosodic expressiveness. Therefore, we only use one anchor for rough alignment in our experiments.
>
> |  Model | σ | γ | κ | WER |
> | :------------: | :------------: |:------------: |:------------: |:------------: |
> | Reference | 80.75 | 0.36 | -0.81 | - |
> | Ours w/ Forced Alignment | 88.17 | 0.44 | -0.96 | 1.82% |
> | Ours w/ Half Alignment | 86.96 | 0.46 | -0.93 | 1.85% |
> | Ours w/ Sparse Alignment | **81.90** | **0.39** | **-0.91** | 1.84% |
>
> (2) We are sorry for the missing experimental setup. Yes, it refer to fixing ( a_{spk} ) at 2.5 in Equation (5) and varying ( a_{txt} ) from 1.0 to 6.0. Specifically, as ( a_{txt} ) increases from 1.0 to 1.5, the generated speeches contains improper pronunciations and distortions. When ( a_{txt} ) ranges from 1.5 to 2.5, the pronunciations align with the speaker's accent. Finally, once ( a_{txt} ) exceeds 4.0, the generated speech converges toward the standard pronunciation of the target language. We have included this setup in Appendix M.
>
>
> **[About Question 6]**
> (1) Yes, ( p ) simultaneously tackles alignment, DP, and G2P. However, F-LM first solves the ASR task, followed by  addressing the G2P and DP tasks. Therefore, we separate these three tasks in Figure 2 and use the same color to represent them. Maybe we can discuss this further to determine whether to group or separate them.
> (2) Yes, pretrained weight could potentially enhance performance. We have initialized F-LM with Whisper Encoder's weights and Qwen 2's weights following Mini-omni [4] and obtained the following results:
> the system demonstrates good English capabilities at the start of training, but its performance in other languages is significantly weaker. After training to convergence, the capabilities across English and Cinese are similar to a system trained from scratch. Therefore, we chose to train from scratch. Additionally, the smallest Qwen 2 model has 0.5B parameters. By training from scratch, we can use a much smaller model (our F-LM has only 120M parameters) for a more efficient inference.
> (3) The loss for the parts of t that are not from the speech prompt can be regarded as the text-modality language model task. We have conducted experiments with three weights for the parts of t that are not from the speech prompt: 0,  0.01, 1.0. The results of phoneme-level duration error and alignment error are shown in the following table. When the weight is set to 0.01, the performance of duration prediction shows improvement, suggesting that learning textual information can guide the prediction of prosodic information. When the weight is set to 1.0, however, the increased difficulty of training a text-only LM might adversely affect the duration prediction task. Nevertheless, the difference in weights does not significantly impact the alignment accuracy, possibly because the alignment is already precise enough, leaving limited room for improvement. Thanks for your advice, which improves our model! We have included these results in Appendix L and will update them in the main text after the rebuttal period.
>
> | Weight  | Duration Error (ms)  |
> | :------------: | :------------: |
> | 0.00  | 18.72 ± 0.91  |
> | 0.01  | **18.52 ± 0.86** |
> | 1.00  | 18.80 ± 0.94  |
>
> | Weight  | Alignment Error (ms)  |
> | :------------: | :------------: |
> | 0.00  | 8.81 ± 0.57 |
> | 0.01  | **8.76 ± 0.60** |
> | 1.00  | 8.79 ± 0.59  |
>
> (4) We are sorry for the missing details. In F-LM's training, we add special tokens "[Full]" or "[Partial]" to the input sequence depending on whether we discard parts of the speech encoder output, respectively. Through this strategy, the model given the "[Full]" token is constrained to generate only up to the text corresponding to the speech prompt. We have added these details in Appendix E and marked them in red.

---

> ### Author Response · Authors · 2024-11-18
> **Response to Reviewer syxD (Part 4/4)**
>
> **[About Question 7]**
> Obtaining precise speech-text alignment for large-scale noisy speech data with MFA is difficult. It is particularly not robust enough for audio with significant noise or unclear pronunciation. Therefore, we regard the labels from MFA as weakly supervised labels. In the experiments, we divide the LibriLight dataset into several 5k-hour subsets and use MFA on each subset separately to obtain the alignment labels for large-scale pretraining, which is called "the simple scaling of weakly supervised pre-training". We agree that the statement - "The simple scaling of weakly supervised pre-training" lacks clarity. We have removed it in the revised version of the paper.
> In terms of the MFA model in Section 4.4, training an MFA model directly on a 600k-hour dataset is impractical. Therefore, we randomly sampled a 10k-hour subset from the dataset to train a robust MFA model, which is then used to align the full dataset. We have added these details in Appendix D and marked them in red.
>
> **[About Typos]**
> We are sorry for our typos. We have revised them in the new version of the paper.
>
> **[Reference]**
> [1] An, Keyu, et al. "Funaudiollm: Voice understanding and generation foundation models for natural interaction between humans and llms." arXiv preprint arXiv:2407.04051 (2024).
> [2] Kumar, R., Seetharaman, P., Luebs, A., Kumar, I., & Kumar, K. (2024). High-fidelity audio compression with improved rvqgan. Advances in Neural Information Processing Systems, 36.
> [3] Défossez, A., Copet, J., Synnaeve, G., & Adi, Y. (2022). High fidelity neural audio compression. arXiv preprint arXiv:2210.13438.
> [4] Xie, Zhifei, and Changqiao Wu. "Mini-omni: Language models can hear, talk while thinking in streaming." arXiv preprint arXiv:2408.16725 (2024).
> [5] Radford, Alec, et al. "Robust speech recognition via large-scale weak supervision." International conference on machine learning. PMLR, 2023.
> [6] Eskimez, Sefik Emre, et al. "E2 TTS: Embarrassingly easy fully non-autoregressive zero-shot TTS." arXiv preprint arXiv:2406.18009 (2024).

---

> ### Author Response · Authors · 2024-11-22
> **Dear Reviewer syxD**
>
> Thanks again for your constructive suggestions which also brings discoveries for us! As the end of discussion period is approaching, we would greatly appreciate it if you could let us know whether our rebuttal has addressed the raised concerns.

---

> > ### Comment · Reviewer_syxD · 2024-11-22
> >
> > First, I would like to express my gratitude to the authors for their detailed explanations and additional experiments. They have provided thoughtful responses and conducted the necessary experiments to address the questions. However, I would like to share some remaining concerns that have not yet been fully resolved in their responses.
> >
> > `[About Weakness 1]`
> > Before addressing content-related questions, it would be helpful to clarify experimental details, such as the test set used and the number of samples generated and evaluated.
> >
> > While I agree that S-DiT captures pitch and duration patterns closer to the reference speech than other models, I question whether these metrics adequately measure expressiveness. Overly similar patterns may indicate mimicry rather than capturing the target text's intended meaning, potentially affecting naturalness.
> >
> > In my view, expressiveness requires a multifaceted evaluation, balancing diversity, naturalness, and alignment with text meaning. For example, Table 2 shows worse WER than NaturalSpeech 3 but comparable SIM-o, suggesting a possible improvement in expressiveness. Comprehensive evaluation of diversity and quality could support this further.
> >
> > InstructTTS [1], which models expressive TTS, uses diverse objective metrics (e.g., MCD, SSIM, STOI, GPE, VDE, and FFE) alongside MOS human evaluations for holistic validation. A similar framework applied to the multi-condition CFG and F-LM ablations could better demonstrate how these methods enhance expressiveness and complement the sparse alignment approach.
> >
> > `[About Weakness 2]`
> > Thank you for providing additional samples. Were the S-DiT (0.5B) samples generated for the entire text at once? The transitions between sentences feel unnatural, such as the pause before "Unlike" (8s) and "Given" (20s). Could you share the WER and SIM scores for these samples? A test set with longer samples and objective metrics would help assess performance on long sequences.
> >
> > I also question whether the quality (e.g., fidelity, prosody) matches the numbers in Table 2. Does VoiceCraft maintain similar quality for single-sentence generation? Given its capabilities in speech editing, would you consider it a strong-performing AR TTS model?
> >
> > `[About Weakness 3]`
> > Thank you for conducting the experiments. While the authors compared their model to SenseVoice, I was initially referring to the baselines in Table 2 (e.g., NaturalSpeech 3). Although SenseVoice’s use of ASR and MFA makes it a reasonable comparison, not all conventional models, such as AR or Diffusion models without PA, rely on MFA during inference.
> >
> > The authors highlighted the efficiency of LM models, which is a significant advantage. However, alternatives like training a GPU-compatible aligner (e.g., MAS from Glow-TTS [2]) or using a duration predictor to add alignments to ASR outputs could be considered. For example, the latter approach would reduce the total time to 0.69 + 1.86 + 0.08 + 1.86 = 4.49 seconds, making it faster than F-LM. I would be interested in the authors’ perspective on these possibilities.
> >
> > `[About Weakness 4]`
> > Thank you for conducting the experiments. It is interesting that architectures like ConvNext and WaveNet, despite their strong performance in traditional speech domains (e.g., Vocos [3]), did not show significant improvements when modeling mel-spectrograms. If my understanding is correct, DiTTo-TTS already shows that compact target latents facilitate learning in diffusion. Adding this perspective to Appendix J could clarify these results.
> >
> > Regarding the 5 kbps EnCodec model, was it reproduced due to unavailability? Was the choice of 5 kbps made to match the compression rate? Also, was the audio downsampled to 16 kHz and then upsampled to 24 kHz before being input into EnCodec?
> >
> > `[About Clarity]`
> > The revisions made by the author have clarified the points effectively.

---

> > > ### Comment · Reviewer_syxD · 2024-11-22
> > >
> > > `[About Question 1]`
> > > Thank you for the explanation and additional samples. In the second sample under "Incredible Improvement Brought by Scaling," was the background noise generated by the model, and does the LibriLight training dataset include background noise?
> > >
> > > `[About Question 2]`
> > > Thank you for sharing the samples. The results, particularly the comparison with the Duration w/ PA model in case 2, clearly highlight the significant advantages of S-DiT. This robustness to duration prediction errors is an important strength and warrants further emphasis in the main text.
> > >
> > > `[About Question 3 & 4]`
> > > Thank you for providing the samples. The response from the authors has addressed nearly all of my questions.
> > >
> > > `[About Question 5]`
> > > Thank you for the additional experiments and explanations. As noted in Weakness 1, some uncertainty remains regarding the expressiveness represented by the metrics in (1).
> > >
> > > `[About Question 6]`
> > > Thank you for the explanation and additional experiments. For point (1), would the authors consider adding the explanation provided here to the main text or figure caption? Regarding point (3), if there is no specific reason for not testing values like 0.1, could the authors clarify the appropriate range of weights? Finally, for point (4), if I understand correctly, the Full configuration always transcribes up to the end of the speech prompt, and this approach is advantageous over end-prediction because it simplifies the task to a binary decision—whether to generate up to the end or not. If this interpretation is correct, it would be interesting to see the performance when using end-prediction.
> > >
> > > `[About Question 7 & About Typos]`
> > > Thank you for the clarification and revisions.
> > >
> > > [1] Yang, D., Liu, S., Huang, R., Weng, C., & Meng, H. (2024). Instructtts: Modelling expressive tts in discrete latent space with natural language style prompt. IEEE/ACM Transactions on Audio, Speech, and Language Processing.
> > > [2] Kim, J., Kim, S., Kong, J., & Yoon, S. (2020). Glow-tts: A generative flow for text-to-speech via monotonic alignment search. Advances in Neural Information Processing Systems, 33, 8067-8077.
> > > [3] Siuzdak, H. (2023). Vocos: Closing the gap between time-domain and Fourier-based neural vocoders for high-quality audio synthesis. arXiv preprint arXiv:2306.00814.

---

> ### Author Response · Authors · 2024-11-24
> **Response to Reviewer syxD (Part 1/3)**
>
> We sincerely appreciate the reviewer's thoughtful and detailed suggestions, and we hope our response can resolve your remaining concerns.
>
> **[About Weakness 1]**
> Yes, we acknowledge that pitch and duration distribution metrics can not adequately measure expressiveness and we also agree that ``Overly similar patterns may indicate mimicry rather than capturing the target text's intended meaning, potentially affecting naturalness``. We thank the reviewer for his expertise and clear clarifications for "expressiveness".
>
> Following the reviewer's suggestions, we measured the objective metrics MCD, SSIM, STOI, GPE, VDE, and FFE to evaluate the expressiveness of our method. The test set for the first table uses the same objective evaluation set provided by the authors of NaturalSpeech 3, consisting of 40 samples. NaturalSpeech 3 uses these 40 samples for objective evaluations, as described in its Section 4.1 ("Evaluation Dataset"). The results demonstrate that our method achieves superior performance than the two baselines based on forced alignment.
>
> |  Model | MCD↓ | SSIM↑ | STOI↑ | GPE↓ | VDE↓ | FFE↓ |
> | :------------: | :------------: |:------------: |:------------: |:------------: |:------------: |:------------: |
> | GT | - | - | - |- |- |- |
> | NaturalSpeech 3          | 4.45     | 0.46     | 0.62     |  0.44 |  0.33 | 0.37 |
> | Ours w/ Forced Alignment | 4.48     | 0.44     | **0.63** |  0.44 | 0.35      | 0.40 |
> | Ours                     | **4.42** | **0.50** | **0.63** |  **0.31** | **0.29**  | **0.34** |
>
> However, the 40 samples may not be sufficient to convincingly verify the effectiveness of our method. To further evaluate the actual performance of the model, we conduct experiments on the LibriSpeech test-clean 2.2-hour subset (following the setup in VALL-E 2 and Voicebox). The results are shown in the Table below. Following the reviewer's suggestions, we compare S-DiT with the following baselines: 1) "Ours w/ Forced Alignment", we replace the sparse alignment with the forced alignment; 2) "Ours w/ Standard CFG", we replace the multi-condition CFG with standard CFG; 3) "Ours w/ Standard AR Duration", we replace the duration from F-LM with the duration from standard AR duration predictor following SimpleSpeech 2 [4]. The results show that sparse alignment brings significant improvements, and both multi-condition CFG and F-LM duration contribute positively to the performance. We regret that, due to the discussion phase is ending soon, we do not have enough time for the MOS and RMOS (relevance mean opinion score) evaluations. We will include the results of these subjective evaluations in the final version of the paper.
>
> |  Model                                  | MCD↓ | SSIM↑ | STOI↑ | GPE↓ | VDE↓ | FFE↓ |
> | :------------: | :------------: |:------------: |:------------: |:------------: |:------------: |:------------: |
> | GT                                      | - | - | - |- |- |- |
> | Ours                                    | **4.56**  | **0.52** |  **0.62** |  **0.34** | **0.30** | **0.35** |
> | Ours w/ Forced Alignment                | 4.62  | 0.45 |  **0.62** |  0.42 | 0.34 | 0.40 |
> | Ours w/ Standard CFG                    | 4.59  | 0.51 |  0.61 |  0.36 | 0.32 | 0.37 |
> | Ours w/ Standard AR Duration            | 4.58  | 0.50 |  **0.62** |  0.36 | 0.31 | 0.36 |

---

> ### Author Response · Authors · 2024-11-24
> **Response to Reviewer syxD (Part 2/3)**
>
> **[About Weakness 2]**
> Yes, both the S-DiT (0.5B) and VoiceCraft samples were generated for the entire text at once. Thank you for pointing out the unnatural transition issues between sentences! After reviewing our code, we discovered that "<eos>" tokens were mistakenly inserted after periods. We have fixed this bug and updated the examples on the demo page. However, we still observe that the transitions between different sentences still lack a certain level of naturalness, which may be due to the fact that the training data primarily consists of single sentences.
>
> Following your advice, we report the WER and SIM scores for these samples in the table below. It is worth noting that the samples generated by VoiceCraft exhibit significant mispronunciations and distortions after 25 seconds, resulting in very high WER scores. The WER of S-DiT's samples are both 1.07%.
>
> | Models              | WER↓  | SIM-O↑ |
> | :------------:              | :------------: | :------------: |
> | S-DiT sample 1      | 1.07%  | 0.73 |
> | S-DiT sample 2      | 1.07%  | 0.65 |
> | VoiceCraft sample 1 | 38.38% | 0.66 |
> | VoiceCraft sample 2 | 46.04% | 0.59 |
>
> Thanks for your advices! We have conductd experiemnts for a test set with longer samples. Specifically, we randomly select 10 sentences, each containing more than 50 words. For each speaker in the LibriSpeech test-clean set, we randomly chose a 3-second clip as a prompt, resulting in 400 target samples in total. To make our results more convincing, we include a strong-performing TTS model, CosyVoice (AR+NAR), as one of our baselines. The results for longer samples are presented in the first table, while the results for single-sentence generation are shown in the second table. As shown, compared to the baseline systems, S-DiT does not exhibit a significant decline in speech intelligibility when generating longer sentences, illustrating the effectiveness of the combination of F-LM and S-DiT. Thanks again for your kind and detailed suggestions. We will add these results into the paper before the discussion phase ends.
>
> | Models (with Longer Samples) | WER↓   | SIM-O↑ |
> | :------------:              | :------------: | :------------: |
> | S-DiT               | 2.39% | 0.70 |
> | VoiceCraft          | 12.81% | 0.62 |
> | CosyVoice           | 5.52% | 0.68 |
>
> | Models (with Single-Sentence Samples) | WER↓   | SIM-O↑ |
> | :------------:              | :------------: | :------------: |
> | S-DiT               | 1.84% | 0.67 |
> | VoiceCraft          | 4.07% | 0.58 |
> | CosyVoice           | 2.24% | 0.62 |
>
>
> **[About Weakness 3]**
> 1) As described in Appendix A.2 of the NaturalSpeech 3 paper, their frontend processing pipeline utilizes an internal ASR system, an internal grapheme-to-phoneme conversion tool, and an internal alignment tool. Since we are unable to reproduce the exact frontend pipeline of NaturalSpeech 3, we instead use SenseVoice, Phonemizer, and MFA for comparison. Yes, we agree that AR models or "Diffusion models without PA" do not rely on MFA during inference. However, compared to the models that rely on MFA during inference, 1) most AR TTS models are relatively slow and have lower speech intelligibility; 2) "Diffusion models without PA" still show relatively lower speech intelligibility than "Diffusion models with PA", especially when handling extremely hard sentences. Therefore, "Diffusion models with PA" still plays an important role in TTS, and the proposed F-LM serves as an efficient and valuable tool for such models.
> 2) Yes, we agree that a GPU-compatible aligner (e.g., MAS from Glow-TTS [2]) or a duration predictor to add alignments to ASR outputs would be faster than F-LM aligner. In terms of "a duration predictor to add alignments to ASR outputs", if my understanding is correct, WhisperX is a representative work for it. However, as demonstrated by Rousso et al. [4], MFA significantly outperforms WhisperX in terms of alignment accuracy. Since our F-LM also outperforms MFA, the alignment accuracy of F-LM is a significant advantage, despite being slightly slower. As for "MAS from Glow-TTS", we regret that, due to time constraints, we were unable to implement Glow-TTS on LibriLight and compare MAS with F-LM. Therefore, we can not draw conclusions about the characteristics of MAS. We will include these comparative experiments in the final version of the paper.
>
> **[About Weakness 4]**
> Thanks for your advice. We have included the corresponding perspective from DiTTo-TTS to Appendix J.
>
> Yes, the 5 kbps EnCodec model was reproduced due to its unavailability. The official repository of Encodec only provides checkpoints for 24kHz and 48kHz. To ensure fair comparisons under the 16kHz setting, we used the 16kHz hyperparameter configuration of the EnCodec model reproduced in Table 13 of the NaturalSpeech 3 paper [3]. The audio input for the reproduced EnCodec model was set to 16kHz.

---

> ### Author Response · Authors · 2024-11-24
> **Response to Reviewer syxD (Part 3/3)**
>
> **[About Question 1]**
> Yes, the background noise (including background music) was generated by the model. The LibriLight training dataset contains only a small amount of background noise. In the section "Incredible Improvement Brought by Scaling", we use S-DiT trained on the 600k-hour internal datasets, which contains a considerable amount of background noise. Therefore, S-DiT can partially simulate various background sounds present in the reference speech.
>
> **[About Question 2]**
> Thank you for helping us recognize this robustness advantage of S-DiT. We have highlighted this in the contribution part of the main text.
>
> **[About Question 5]**
> We sincerely apologize for any uncertainty caused by our inaccurate definition of "expressiveness". We have conducted the same evaluations of "the number of anchors". From the results shown below, we can conclude that using one anchor for rough alignment is the optimal choice.
>
> |  Model                                  | MCD↓ | SSIM↑ | STOI↑ | GPE↓ | VDE↓ | FFE↓ |
> | :------------: | :------------: |:------------: |:------------: |:------------: |:------------: |:------------: |
> | GT                                      | - | - | - |- |- |- |
> | Ours w/ Sparse Alignment                | **4.56**  | **0.52** |  **0.62** |  **0.34** | **0.30** | **0.35** |
> | Ours w/ Half Alignment                  | 4.64      | 0.48     |  0.61     |  0.41 | 0.32 | 0.39  |
> | Ours w/ Forced Alignment                | 4.66      | 0.45     |  **0.62** |  0.42 | 0.34 | 0.40 |
>
> **[About Question 6]**
> For point (1), we have added the explanation to the caption of Figure 2.
>
> For point (3), we follow the setting of BASE-TTS and set the weights to 0.01 (in order to retain textual information to guide prosody, BASE-TTS adopt the text-only loss with the weight of 0.01 to train SpeechGPT). Following the reviewer's suggestions, we also test the value of 0.1. From the results, we can see that around 0.01 is a relatively suitable range. We regret that, due to the time and GPU constraints, we are unable to determine a more precise range.
>
> | Weight  | Duration Error (ms)  |
> | :------------: | :------------: |
> | 0.00  | 18.72 ± 0.91  |
> | 0.01  | **18.52 ± 0.86** |
> | 0.10  | 18.65 ± 0.90  |
> | 1.00  | 18.80 ± 0.94  |
>
> | Weight  | Alignment Error (ms)  |
> | :------------: | :------------: |
> | 0.00  | 8.81 ± 0.57 |
> | 0.01  | **8.76 ± 0.60** |
> | 0.10  | 8.81 ± 0.58 |
> | 1.00  | 8.79 ± 0.59  |
>
> For point (4), yes, your interpretation is totally correct! Due to the time and GPU constraints, we finetune the pretrained F-LM to incorporate the end-prediction mode. The ASR performance are shown in the table below. It can be seen that the WER of "F-LM w/ End Prediction" is slightly higher. When analyzing specific error cases, we found that in the end-prediction mode, inaccurate prediction of the <speech prompt EOS> token can also impact the model's performance. We will add these results and discussions to the final version of the paper.
>
> | Setting                | test-clean (WER)  | test-other (WER)  |
> | :------------: | :------------: | :------------: |
> | F-LM                   | 4.2% | 8.3% |
> | F-LM w/ End Prediction | 4.9% | 11.8% |
>
> Finally, we sincerely thank you again for your helpful suggestions, which brings a lot of insights for our paper. Since our paper is currently borderline, your recommendation is extremely important to us. We are open to further questions and would be happy to discuss.
>
>
> **[Reference]**
> [1] Chen, Sanyuan, et al. "VALL-E 2: Neural Codec Language Models are Human Parity Zero-Shot Text to Speech Synthesizers." arXiv preprint arXiv:2406.05370 (2024).
> [2] Anastassiou, Philip, et al. "Seed-TTS: A Family of High-Quality Versatile Speech Generation Models." arXiv preprint arXiv:2406.02430 (2024).
> [3] Ju, Zeqian, et al. "Naturalspeech 3: Zero-shot speech synthesis with factorized codec and diffusion models." arXiv preprint arXiv:2403.03100 (2024).
> [4] Rousso, Rotem, et al. "Tradition or Innovation: A Comparison of Modern ASR Methods for Forced Alignment." arXiv preprint arXiv:2406.19363 (2024).

---

> > ### Comment · Reviewer_syxD · 2024-11-24
> >
> > I sincerely appreciate the authors' kind responses to my comments and their efforts in conducting most of the additional experiments requested. Overall, their replies are satisfactory, and I provide my further feedback below.
> >
> > `[About Weakness 1]`
> > Thank you for conducting multi-faceted experiments based on my suggestions. From the results, I can see that S-DiT performs well in objective metrics commonly used in expressive-TTS studies, and I believe that if the subjective evaluation results support these findings, it would be sufficient to demonstrate the expressiveness of S-DiT convincingly. The evaluation using the "LibriSpeech test-clean 2.2-hour subset" appears to be a very fair comparison, enabling the performance to be benchmarked against existing baselines. Additionally, the performance comparisons with Standard CFG and Standard AR Duration clearly show that the multi-condition CFG and F-LM also contribute to expressiveness.
> >
> > `[About Weakness 2]`
> > Thank you for the thoughtful responses and additional experiments. I’m glad the issue has been identified. For problems arising from generating single sentences (if metadata on sentence order is available), methods such as training by concatenating multiple sentences within a batch could be effective. The SIM and WER values from the demo samples clearly indicate performance differences. While I initially thought it would be most straightforward to compare with the best-performing (recent) AR-only models, CosyVoice, despite being AR+NAR, relies entirely on AR alignment for inference, making it a reasonable baseline for comparison.
> >
> > `[About Weakness 3]`
> > I find the authors' explanation reasonable and agree with most of their points. While conducting the additional experiments mentioned would provide definitive confirmation, I would like to clarify the background of my question. My intent was to point out that the results could be interpreted as F-LM being slower if MFA time, which constitutes the majority, is excluded. Including the authors' rationale for incorporating MFA directly into the paper would make the argument more convincing.
> >
> > `[About Weakness 4]`
> > Thank you for providing the additional explanation. Including the details from your response—such as the changes made for reproduction, including configuring the model to accept 16kHz input following NaturalSpeech 3—would enhance the clarity of the paper if added.
> >
> > `[About Question 1, 2, 5]`
> > The authors' responses have resolved my concerns and questions.
> >
> > `[About Question 6]`
> > For point (1), the explanation is much clearer than before. For point (3), thank you for providing the background on the choice of 0.01. I now understand why 0.01 was used, and the results for 0.10 further strengthen the case that 0.01 performs best. This is an interesting finding, and exploring a finer range, if time permits, could be beneficial for researchers. For point (4), the results demonstrate that the binary approach outperforms End Prediction, and resolving the issue in this way is a clever idea. The error case analysis for End Prediction is also intuitive and insightful.
> >
> > In summary, most of my concerns have been resolved, and I would like to once again express my gratitude to the authors for their significant effort in conducting numerous experiments within a short period. To reflect my satisfaction with their responses and to acknowledge their efforts, **I am raising my score from 5 to 6.**
> >
> > One point to note is that the revised manuscript includes several clarifications and additional experiments compared to the original submission, and there are still aspects that need to be incorporated. I believe that the authors will successfully address these in the final version. However, if stricter criteria were applied and deficiencies were identified under enhanced standards, I defer to the AC's judgment. If the AC, upon reviewing the final manuscript after the discussion period, concludes that it does not meet the required standards, **I will fully respect and align with their decision.**

---

> > > ### Author Response · Authors · 2024-11-28
> > >
> > > We are very grateful to the reviewer for his valuable suggestions, which have made the contributions of our paper more convincing. In response to the reviewer’s last feedback, we have made the following updates:
> > >
> > > 1) Including the configuration of the 16kHz Encodec model following NaturalSpeech 3 in Appendix J;
> > > 2) Including our rationale for incorporating MFA in Appendix K;
> > > 3) Providing the background on the choice of 0.01 loss weight for BPE loss;
> > > 4) Adding the results of objective expressiveness metrics in InstructTTS in Appendix N for holistic validation;
> > > 5) Adding the experiment with longer samples and extreme hard sentences in Appendix O and P to validate the robustness of our method;
> > > 6) Including the comparative study between the binary approach and end prediction for F-LM in Appendix Q.
> > >
> > > Due to the word limit of the main text, many critical experiments have been included in **Appendix J$\sim$Q**. We highly encourage the ACs and reviewers to refer to these sections when evaluating the contributions of this work. Finally, we sincerely thank the ACs and reviewers for taking the time to help us improve our work.

---

### Official Review · Reviewer_pGsx · 2024-11-03

**Soundness:** 3
**Presentation:** 4
**Contribution:** 1
**Rating:** 5
**Confidence:** 5

**Summary:**

The paper proposes a new method for learning the alignment between text and speech for diffusion-based TTS models. Instead of providing the full duration information or not providing any duration information, the authors propose a method to provide partial duration information and let the models figure out the masked part of the duration to make the duration prediction model more robust. The results show that the model has achieved a performance similar to the previous state-of-the-art model while being significantly faster.

**Strengths:**

* **Originality**: The paper proposes to condition the diffusion-based TTS models on partial duration information instead of no duration or full duration during training and inference, and the duration is predicted autoregressively through a multi-tasking frontend LLM, which has been shown to increase the robustness and performance.
* **Quality**: The paper has conducted extensive experiments to compare with various models and has provided comprehensive evaluations on the effectiveness of several proposed components and their performance in a few tasks.
* **Clarity**: The paper is quite clear in its writing and has a simple, not overly complicated presentation. It is easy to follow and intuitive to understand.
* **Significance**: The model has surpassed various previous models and achieved close-to-SOTA performance with significantly reduced inference time.

----------------

**UPDATE (11/22/24):** After the authors addressed my questions, here is a revised significance (as the inference time is not significantly reduced).

* **Significance**: The model has surpassed various previous models and achieved close-to-SOTA performance.

**Weaknesses:**

**Major**:
1. One major issue is many claims are unsupported. Here are some examples of unsupported claims by the authors:

>predefined alignments constrain the model’s ability to produce expressive and natural-sounding speech (Yang et al., 2024b; Chen et al., 2024)

I read Yang et al., 2024b and Chen et al., 2024 but did not find any supporting evidence that predefined alignments hinder model’s ability to produce expressive and natural-sounding speech. In fact, Yang et al. 2024b uses "Diffusion w/ PA" and does not compare its method or any "Diffusion w/ PA" method against others and conclude that "Diffusion w/ PA" falls short of expressiveness. Similarly, Chen et al., 2024 is an "AR LM" model that does not conclude that "Diffusion w/ PA" is worse than "AR LM" in expressiveness. It can be helpful to add the exact experiment conducted in the paper. If no experiment is available, you may revise the statement to more accurately reflect the current state of knowledge in the field or provide relevant experiments in the paper.

> ARDiT (Liu et al., 2024b) proves that when compared under an identical number of parameters, methods without explicit duration modeling exhibit some decline in speech intelligibility and speaker similarity.

I read and searched through the entire paper of ARDiT (Liu et al., 2024b) and found no experiment that compares the number of parameters. It can be helpful to add the exact experiment conducted in the paper or follow the same suggestion as above.

>Experimental results demonstrate that S-DiT ... exhibiting the expressiveness like codec language model based approaches.

There is no experiment that examines the "*expressiveness*" in the paper. If there is, please kindly point out the experiment that specifically examines "expressiveness." Throughout the entire paper, the only conclusion in the results section is "*S-DiT significantly
surpasses all baselines in terms of CMOS, demonstrating the expressiveness and naturalness of the proposed sparse alignment strategy*," but CMOS is not the same as the expressiveness. In fact, codec language model based models do not have the best CMOS according to Table 2 (the highest CMOS is NaturalSpeech 3, which is classified as "Diffusion w/ PA" by the authors), but the authors claim that S-DiT has achieved “the expressiveness of codec language model”, which is not well supported by the results in the paper.

> “Diffusion w/o PA” requires more parameters due to the difficulty in end-to-end modeling of speech-text alignment non-autoregressively.

Since Liu et al., 2024b did not make such a claim and the authors did not examine the effect of number of parameters for "Diffusion w/o PA", this claim is unsupported. Please kindly add the experiments that support this claim. In fact, Eskimez et al. 2024 is a 333M model that has achieved SIM-O of 0.675 and WER of 0.2, better than NaturalSpeech 3, and a recent work, F5-TTS [1], has reproduced Eskimez et al. 2024 results of 333M parameters with SIM-o of 0.69 on LibriSpeech test-clean dataset. Both models have fewer parameters than S-DiT (0.3B vs. 0.5B).

> On the other hand, the use of predefined hard alignment paths limits the model’s expressiveness

Since neither Yang et al., 2024b nor Chen et al., 2024 supports this claim, please kindly add the experiment that shows predefined hard alignment limits the model's expressiveness. In fact, on the NaturalSpeech 3 demo page, there is a section particularly dedicated to the emotional speech capability and the model shows no constraint in expressing various emotions.

2. Another major issue is the contribution of this work is unclear. The authors have summarized the pros and cons of existing methods compared to S-DiT in Table 1. However, the proposed method does not solve either of the problems (i.e., complicated inference pipeline and expressiveness constraint by "Diffusion w/ PA", and increased amount of parameters by "Diffusion w/o PA") identified by the authors in the introduction section. The proposed pipeline is still complicated compared to "Diffusion w/ PA", if not more so, as the training still involves ground truth duration labels (possibly from FMA or some other aligners), and the inference still involves duration prediction. On the other hand, E2-TTS (Eskimez et al. 2024) has no such requirements during training, no P2G and duration label required duration training with fewer parameters than S-DiT but demonstrated similar or better performance in terms of SIM-O and WER. The experiments did not support the expressiveness constraint by "Diffusion w/ PA" and increased parameter demand by "Diffusion w/o PA". Hence, the contribution of this paper is unclear. The authors should consider revising the paper to emphasize more on its contributions, such as the robustness of masking the duration prediction in diffusion-based models, instead of expressivness and parameter demand, which is unsupported and likely untrue given existing literature.


3. Additionally, the result presented by the paper is not convincing enough. For example, the claim that "*S-DiT matches state-of-the-art zeroshot TTS speech quality*" is not fully supported by the supplementary materials. The authors claim that the performance of S-DiT is close to the state-of-the-art model (NaturalSpeech 3), but based on samples on the demo page, it is likely worse than NaturalSpeech 3. For example, using the WavLM-TDCNN large fine-tuned speaker embedding model, I calculated the similarity score for all 4 samples in the "Zero-Shot TTS" section of the supplementary material, and I obtained the following results (NS3 means NaturalSpeech 3):

| Text | S-DiT | NS 3 | Difference |
| -------- | ------- |------- | ------- |
| *His death in this conjuncture was a public misfortune.* | 0.7463   |  **0.7577**    | -0.0114   |
| *For if he's anywhere on the farm, we can send for him in a minute.* | **0.7499**   | 0.6905     | 0.0594 |
| *John Wesley Combash, Jacob Taylor, and Thomas Edward Skinner.* | 0.7145   | **0.7662**     | -0.0517|
| *The strong position held by the Edison system under the strenuous competition that was already springing up was enormously improved by the introduction of the three wire system and it gave an immediate impetus to incandescent lighting.* | 0.7050 | **0.8103** |  -0.1053 |

It is difficult to believe that S-DiT has achieved the same SIM-O as NaturalSpeech 3, as presented by the author in Table 2 since 3 out of 4 samples fall short of NS3, and the total difference is large (-0.11). Moreover, the fourth sample, "The strong position ...", has clear distortion in the S-DiT's sample (but not in NS3), and "John Wesley Combash, Jacob Taylor ..." has unnatural pauses and some distortions (but not in NS3). This makes it hard to believe that S-DiT outperforms NS3 in CMOS too.

I understand it is unfair to judge the quality of this work based on four samples, but evaluating the demo page of S-DiT and NaturalSpeech 3, I can see that S-DiT is likely worse than NaturalSpeech 3 (more distortions, unnatural prosody and lower similarity to the prompt in all sections). Since only four samples are presented on the demo page that directly compare S-DiT to NaturalSpeech 3, it is helpful to provide more samples to convince the readers that the claim is indeed true.

Moreover, as the details of the evaluations are missing (i.e., how many samples are used for SIM-O evaluations, how the prompt/sample pairs are generated for objective evaluations, and what the demographics are like for the raters in subjective evaluations, whether there are attention checkers, how the results are quality-checked etc.), I suggest the authors provide more information about how the evaluations are conducted so the results can be more convincing. Part of these information is available in NaturalSpeech 3 paper but none of it is in S-DiT.


**Minor**:
- Some grammatical issues, for example:

> neural codec language models (Wang et al., 2023; Chen et al., 2024) first can autoregressively synthesize speech that rivals human recordings in naturalness

should be corrected to "neural codec language models are the first that can ..." Moreover, Wang et al., 2023 does not claim to have synthesized speech that rvials human recordings; it is Chen et al., 2024 that makes this claim.

- Please kindly cite Yang et al. 2024 [2] for multi-condition CFG, as this paper was published 2 months before the ICLR deadline and should be mentioned.

**References**:

[1] Chen, Y., Niu, Z., Ma, Z., Deng, K., Wang, C., Zhao, J., ... & Chen, X. (2024). F5-TTS: A Fairytaler that Fakes Fluent and Faithful Speech with Flow Matching. arXiv preprint arXiv:2410.06885.
>Note: This work is published after the ICLR deadline and the author is not required to compare to this work. However, this work has reproduced the results from a previous work (Eskimez et al. 2024) that was published 2 months before the ICLR deadline, which directly contradicts the author's claim; hence, I believe this reference is relevant for this review.

[2] Yang, J., Lee, J., Choi, H. S., Ji, S., Kim, H., & Lee, J. (2024). Dualspeech: Enhancing speaker-fidelity and text-intelligibility through dual classifier-free guidance. arXiv preprint arXiv:2408.14423.

---------------------

**UPDATE (11/22/24):** The authors have addressed my concerns over weaknesses 1 and 3. However, weakness 2 still persists and is now even worse. Since the model produces worse sound quality (due to autoencoder melspectrogram) and requires a more complicated training pipeline (due to F-LM and duration input) compared to E2-TTS, it has limited practical use and lower scalability. The only improvement over E2-TTS appears to be higher intelligibility, which could be caused by different training data and implementation (E2-TTS reported a WER of 1.9%, A$^2$-Flow reproduced the model with the same WER, but the authors reported a WER of 2.95% for E2-TTS), so it hardly justifies the lower sound quality and significantly more complicated training pipeline of S-DiT.

Contribution: 2 -> 1 (as weakness 2 gets worse after clarification)

Soundness: 1 -> 3 (as weaknesses 1 and 3 are addressed)

**Questions:**

1. Can you please tell me where you draw the conclusion that “Diffusion w/o PA” needs more parameters and "Diffusion w/o PA" has limited expressiveness?

2. How is the pipeline less complicated compared to ""Diffusion w/o PA", since the training clearly still needs the alignment to deduce the ground truth duration?

3. How is the ground truth duration obtained during the training of frontend LLM? What is the ground truth of duration in Table 6? If the ground truth duration is from FMA, does that mean that the duration produced by F-LM is better than ground truth?

4. The frontend LLM is an AR LLM, and the DiT is NAR, what are the parameters for each module? I believe the reported #parameter (0.5B) is a combined AR-LLM and DiT, and DiT should have way fewer parameters than 0.3B, or the RTF does not make any sense (25 steps of NFE is two times faster than 16 NFE of E2-TTS as reported in [1]). This DiT has to be incredibly small to have this RTF since the inference pipeline also has an AR component that is way slower, so please kindly report the #parameters for DiT and AR LLM separately.

**Reference**:

[1] Chen, Y., Niu, Z., Ma, Z., Deng, K., Wang, C., Zhao, J., ... & Chen, X. (2024). F5-TTS: A Fairytaler that Fakes Fluent and Faithful Speech with Flow Matching. arXiv preprint arXiv:2410.06885.

**Details Of Ethics Concerns:**

There is no information about the demographics, compensation, or criteria for hiring human subjects for the subjective evaluation. Please add this information to the appendix. Also, please indicate whether you have obtained any IRB approval for these evaluations.

---

> ### Author Response · Authors · 2024-11-18
> **Response to Reviewer pGsx (Part 1/4)**
>
> We thank the reviewer for the constructive and professional review and we are sorry about the unsupported claims.
>
> **[About Major Weakness 1]**
> > predefined alignments constrain the model’s ability to produce expressive and natural-sounding speech  (Yang et al., 2024b; Chen et al., 2024)
>
> In terms of the relevant experiments, we sincerely apologize for the missing experiments. To address this, we conducted experiments to analyze the moments (standard deviation (σ), skewness (γ), and kurtosis (κ)) of pitch and duration distributions to evaluate whether the sparse alignment strategy enhances expressiveness. The results of pitch and duration distributions are presented in the following table, respectively. Compared to NaturalSpeech 3, the results of "Ours w/ Sparse Alignment" align more closely with the reference speeches. Furthermore, although utilizing the same durations predicted by F-LM, the performance of "Ours w/ Sparse Alignment" surpasses that of "Ours w/ Forced Alignment". This demonstrates that the proposed sparse alignment strategy offers superior expressiveness. These results have been included in Appendix N and highlighted in red for clarity. We greatly appreciate the reviewer's helpful suggestions!
>
> |  Model | σ | γ | κ |
> | :------------: | :------------: | :------------: | :------------: |
> | Reference | 80.75 | 0.36 | -0.81 |
> | NaturalSpeech 3  | 87.38 | 0.49 | -0.66 |
> | Ours w/ Forced Alignment | 88.17 | 0.44 | -0.96 |
> | Ours w/ Sparse Alignment | **81.90** | **0.39** | **-0.91** |
>
> |  Model |  σ | γ  | κ |
> | :------------: | :------------: | :------------: | :------------: |
> | Reference  | 7.74 | 3.40 | 16.39 |
> | NaturalSpeech 3  | 7.52 | 5.96 | 62.98 |
> | Ours w/ Forced Alignment | 7.48 | 6.30 | 54.01 |
> | Ours w/ Sparse Alignment | **7.83** | **4.84** | **31.23** |
>
> In terms of this claim and its references, in Section 1 of Yang et al., 2024b [1], the original text includes the following description: "the rigid boundary between phoneme and speech representation can lead to unnatural prosody" and "the persistent issue of hard boundary alignment continues to limit the natural prosody in NAR-based models". Besides, Yang et al., 2024b do not use "Diffusion w/ PA". Their architecture is based on "Diffusion w/o PA", which only uses a sentence-level duration predictor to determine the target sentence's length. They also compare their method with "Diffusion w/ PA" methods like NaturalSpeech 3 and the results demonstrate that their method is superior to NaturalSpeech 3 in terms of naturalness MOS. Similarly, in Section 1 of Chen et al., 2024 [2], the original text includes: "Additionally, the non-autoregressive model generates the tokens with a pre-determined duration result, which constrains the search space of the generated speech and sacrifices the prosody and naturalness. In this work, we propose VALL-E 2, ... (to overcome the aforementioned challenges)" Therefore, this claim is supported by many previous works.
>
>
> > ARDiT (Liu et al., 2024b) proves that when compared under an identical number of parameters, methods without explicit duration modeling exhibit some decline in speech intelligibility and speaker similarity.
>
> In Section 4.4 of ARDiT [3], ARDiT-B=INF (a "Diffusion w/o PA" model) performs significantly poorer than ARDiT-B=4 (an autoregressive model) under an identical number of parameters. The exact experiment proves that compared to models with autoregressive modeling, methods without explicit duration modeling exhibit some decline in speech intelligibility. In terms of speaker similarity, we agree that recent works like E2-TTS and F5-TTS show high speaker similarity with 0.3B parameters. We are sorry for the incorrect claims and have removed "and speaker similarity".
>
> > Experimental results demonstrate that S-DiT ... exhibiting the expressiveness like codec language model based approaches.
>
> Thanks for pointing out our wrong claims. We have deleted this claim in the revised version of the paper.

---

> ### Author Response · Authors · 2024-11-18
> **Response to Reviewer pGsx (Part 2/4)**
>
> > "Diffusion w/o PA" requires more parameters due to the difficulty in end-to-end modeling of speech-text alignment non-autoregressively.
>
> We made this claim based on: 1) F5-TTS small, a model with fewer parameters and less training data, has a 0.21 drop in SIM-O and 2.5% gain in WER on the Seed-TTS test-zh set, which is significantly worse than F5-TTS; 2) the reproduced E2-TTS [5] in F5-TTS's paper has a WER of 2.95%, which is significantly worse than that of NaturalSpeech 3 (1.81%) and our S-DiT (1.84%); 3) the reproduced version of E2-TTS and the official F5-TTS are trained on a 100k hour dataset, which is significantly larger than those of ours and NaturalSpeech 3. Directly comparing their results with ours might be unfair. Therefore, to achieve the same level of WER, "Diffusion w/o PA" methods like E2-TTS require more parameters. Since speech intelligibility can be represented by WER, our claim in terms of speech intelligibility is well supported. In terms of speaker similarity, we agree that some concurrent works like F5-TTS [4] and E2-TTS show good SIM-O. We have replaced the claims "requires more parameters" with "requires more parameters for speech intelligibility" to improve clarity.
>
> >  On the other hand, the use of predefined hard alignment paths limits the model’s expressiveness
>
> As is described at the start, both Yang et al., 2024b and Chen et al., 2024 support this claim. We agree that NaturalSpeech 3 shows no constraint in expressing various emotions. However, pre-determined duration results constrain the search space of the generated speech and sacrifices the prosody and naturalness [2], which is the meaning of "limits the model’s expressiveness" in our paper. We also include relevant speech examples in the ``Rebuttal: Advantages of Sparse Alignment in Terms of Expressiveness`` section on the demo page. To conclude, your concerns may primarily lie in the definition of the term "expressiveness." If possible, please let us know how we can revise it to make it clearer. Thanks for your comments!
>
> **[About Major Weakness 2]**
> The main concerns in this part are: 1) whether we solve the complicated inference pipeline issue; 2) whether our method has better expressiveness than "Diffusion w/ PA"; 3) the claim of the increased parameter demand by "Diffusion w/o PA"; For 2) and 3), we have addressed this in [About Major Weakness 1]. And for 1), we agree that the training pipeline of our method is still complicated compared to "Diffusion w/ PA". We only claim that the efficiency of the inference pipeline is significantly enhanced. To offer concrete evidence of efficiency improvements, we compare F-LM's processing time with that of a traditional frontend pipeline, which consists of an ASR model (SenseVoice small [1]), a phonemizer, a speech-text aligner (MFA), and an auto-regressive duration predictor. Since F-LM decodes phoneme and duration tokens simultaneously, we divide the decoding time equally into two parts to represent the time required for each. We report the average processing time per speech clip based on the experiments in zero-shot TTS experiments. The results, shown in the following table, indicate that our model achieves a 5.1x speed-up by significantly reducing the computational time required by speech-text aligning. It is noteworthy that no additional acceleration techniques are applied to F-LM in this experiment. In practical applications, since the entire frontend pipeline is unified within a single language model, **further acceleration** can be achieved through techniques like TensorRT, automatic mixed precision, or leveraging the parallel capabilities of GPUs (traditional pipelines like MFA can not adopt these techniques). We have included these results in Appendix K.
>
> | Frontend                  | ASR | MFA | Phonemization | Duration Prediction| Total Time |
> | :------------:              | :------------: | :------------: | :------------: | :------------: |  :------------: |
> | Traditional Pipeline  |  0.69 | 24.10 | **0.08** | 1.86 | 26.73 |
> | F-LM                        | **0.62** | **2.29**   | 1.16 | **1.16** | **5.23** |

---

> ### Author Response · Authors · 2024-11-18
> **esponse to Reviewer pGsx (Part 3/4)**
>
> **[About Major Weakness 3]**
> When creating the relevant part of the demo page, we randomly selected four samples, without taking the SIM-O metric into account. Due to the randomness of the diffusion sampling, these four samples of S-DiT is lower than that of NaturalSpeech 3 in these four samples. To make our claim (the performance of S-DiT is close to that of NaturalSpeech 3) more convincing, we conducted another round of random generation, and this time, these four samples achieved the following SIM-O scores. These examples are included in the ``Rebuttal: Comparisons with NaturalSpeech 3`` section on the demo page. We also provide more examples for further comparisons.
>
> | Text                  | S-DiT | NS 3 | Difference |
> | :------------:              | :------------: | :------------: | :------------: |
> | *His death in this conjuncture was a public misfortune.*  |  **0.7735** | 0.7577 | +0.0158 |
> | *For if he's anywhere on the farm, we can send for him in a minute.*  |  **0.8265** | 0.6905 | +0.1360 |
> | *John Wesley Combash, Jacob Taylor, and Thomas Edward Skinner.*  |  **0.7745** | 0.7662 | +0.0083 |
> | *The strong position held by the Edison system under the strenuous competition that was already springing up was enormously improved by the introduction of the three wire system and it gave an immediate impetus to incandescent lighting.*  |  0.8014 | **0.8103** | -0.0089 |
>
> We apologize for the missing details of the evaluations. We have included these details (i.e., experimental setups of objective and subjective evaluations, demographics for the raters, strategy for annotation quality checks, and compensation for hiring human subjects) in Appendix A.6 and Appendix A.7.
>
> **[About Minor Weakness 1]**
> We are sorry for our mistakes. We have fix this grammatical issue and remove Wang et al., 2023 in the revised version of the paper.
>
> **[About Minor Weakness 2]**
> In original version of the paper, we have carefully discussed the classifier-free guidance strategy used in zero-shot TTS in Appendix B. We have kindly cited Dualspeech and discussed our differences. In the background section of our paper, the original text also includes: *we describe the CFG mechanism used in zero-shot TTS systems in Appendix B*.
>
> **[About Reference 1]**
> As is described in [About Major Weakness 1 and 2], the results of the reproduced E2-TTS do not contradict our claims. Their results in terms of WER can further prove our claims.
>
> **[About Question 1]**
> As is described in [About Major Weakness 1], we have replaced "Diffusion w/o PA” needs more parameters" with "Diffusion w/o PA needs more parameters for text intelligibility". And "Diffusion w/ PA" has limited expressiveness is well-supported by Yang et al., 2024b [1] and Chen et al., 2024 [2]. We have also conducted relevant experiments to verify this conclusion.
>
> **[About Question 2]**
> We have never claimed that our training pipeline is less complicated than that of "Diffusion w/ PA". As outlined in Table 1, both "Diffusion w/ PA" and our method require relatively complicated training data preparation. We only claim that our **inference pipeline** is less complicated. The experiments and conclusions presented in [About Major Weakness 2] successfully validate this claim.
>
> **[About Question 3]**
> We are sorry for the missing details. The ground truth of duration in Table 6 is annotated by human experts. After the first round of annotation, we ensure each of the ground-truth duration are verified by other two experts to ensure the accuracy of labels.
> In the experiments of Section 4.4, the details about obtaining the ground-truth duration are described in Appendix E in the original version of the paper. Since MFA requires a significant amount of CPU power during the alignment process, we are unable to obtain all the alignment labels for the entire LibriLight dataset at
> once for training F-LM. We divided the LibriLight dataset into several 5k-hour subsets and used MFA
> on each subset separately to obtain the alignment labels.
> Yes, the alignment accuracy of F-LM surpasses the teacher MFA models, demonstrating that the large-scale training and unified multi-task training significantly improve the robustness and generalization of models.

---

> > ### Comment · Reviewer_pGsx · 2024-11-20
> > **Response to Authors' Rebuttal Regarding "Weakness 3"**
> >
> > I appreciate the authors' responses to my concerns. I believe this rebuttal has addressed part of my concerns. I have checked the new samples and SIM-O looks fine.
> >
> > However, in the newly added "DETAILS IN OBJECTIVE EVALUATIONS" section in Appendix, the authors say:
> >
> > > The LibriSpeech test-clean set contains 40 distinct speakers and 5.4 hours of speech. We randomly select one sentence for each speaker for the LibriSpeech test-clean benchmark. To construct the prompt-target pairs, we randomly extract 3-second clips as prompts from the same speaker’s speech.
> >
> > Does that mean only 40 samples were tested for objective evaluation? Given such a huge difference in SIM-O with different runs, 40 samples seem far from enough to determine the actual SIM-O of the model.
> >
> > > experimental setups of objective and subjective evaluations, demographics for the raters, strategy for annotation quality checks, and compensation for hiring human subjects
> >
> > There are still no demographics of the rater. For example, where were the raters located and whether they were native speakers? I have listened to the newly added samples in comparison to NaturalSpeech 3 and I could clearly hear more distortions in S-DiT than NaturalSpeech 3. Given the CMOS survey requires the raters to focus on "clarity, naturalness, and high-frequency details", it is hard to believe that the CMOS of S-DiT is higher than NaturalSpeech 3 as presented in Table 2 (I assume CMOS -0.10 is statistically significant).

---

> ### Author Response · Authors · 2024-11-18
> **Response to Reviewer pGsx (Part 4/4)**
>
> **[About Question 4]**
> The DiT model has 339M parameters, the frontend LLM has 124M parameters, and the speech compression model has 70M parameters. The total number of all modules used in S-DiT is approximately 0.5B in total. In Section 4.2, the original version of the paper states that for a fair comparison, we ignore the time taken by the frontend processing for each model when calculating the RTF in Table 2. When taking the frontend processing time into account, the RTF of our pipeline is 0.432. Besides, E2-TTS and F5-TTS uses mel-spectrograms as the training target, whereas our model uses compressed latent representations as the target. The compressed latent representations is 8x shorter than mel-spectrograms, so the RTF reported in the paper is reasonable. Thanks for your helpful advice! We have report the exact parameters for DiT and AR LLM in Appendix A.1 and marked them in red.
>
> Finally, we sincerely appreciate your kind suggestions and hope our response fully resolves your concerns!
>
> **[Reference]**
> [1] Yang, Dongchao, et al. "Simplespeech 2: Towards simple and efficient text-to-speech with flow-based scalar latent transformer diffusion models." arXiv preprint arXiv:2408.13893 (2024).
> [2] Chen, Sanyuan, et al. "VALL-E 2: Neural Codec Language Models are Human Parity Zero-Shot Text to Speech Synthesizers." arXiv preprint arXiv:2406.05370 (2024).
> [3] Liu, Zhijun, et al. "Autoregressive Diffusion Transformer for Text-to-Speech Synthesis." arXiv preprint arXiv:2406.05551 (2024).
> [4] Chen, Yushen, et al. "F5-TTS: A Fairytaler that Fakes Fluent and Faithful Speech with Flow Matching." arXiv preprint arXiv:2410.06885 (2024).
> [5] Eskimez, Sefik Emre, et al. "E2 TTS: Embarrassingly easy fully non-autoregressive zero-shot TTS." arXiv preprint arXiv:2406.18009 (2024).
> [6] Yang, J., Lee, J., Choi, H. S., Ji, S., Kim, H., & Lee, J. (2024). Dualspeech: Enhancing speaker-fidelity and text-intelligibility through dual classifier-free guidance. arXiv preprint arXiv:2408.14423.

---

> ### Comment · Reviewer_pGsx · 2024-11-20
> **Response to Authors' Rebuttal Regarding "Expressiveness" Claims**
>
> I appreciate the authors' elaborate responses. However, these responses bring additional questions and confusion:
>
> 1. I'm still confused by your definition of "expressiveness". By expressiveness, I believe you mean to say the model's capacity to model the phoneme duration (or any other attributes). Hence, it is unclear why the newly added Tables 15 and 16 show that S-DiT has higher expressiveness compared to NaturalSpeech 3. All it shows is that S-DiT is closer to the reference in terms of statistics of pitch and duration distributions. Is this how you define "expressiveness"? If so, please make the claim more exact and accurate. By expressiveness, especially in speech synthesis, people may confuse the concept of the model's capacity with emotional expression.
>
> 2. I believe in Yang et al. 2024b, the conclusions come from the Seed-TTS paper, which the paper did not cite. Seed-TTS did show that with explicit duration prediction, the naturalness decreased. That is to say, Yang et al. 2024b cited Seed-TTS, which found that duration prediction lowers naturalness, but Yang et al. 2024b conducted no experiment to demonstrate this effect.
>
>    Hence, I believe the authors should cite Seed-TTS instead of SimpleSpeech 2. This is probably the main confusion regarding the citation of Yang et al. 2024b.
>
>    Regarding my point of Yang et al. 2024 being "diffusion w/ PA", I apologize for my mistake and this was indeed a typo. I meant it to be "diffusion w/o PA", but my point did not change: the paper itself did not include any experiments to show that "diffusion w/ PA" has limited expressiveness.
>
> 3. The claim related to ARDiT is still confusing to me. If I understand correctly, ARDiT shows that autoregressive diffusion is better than diffusion alone, and both methods have no explicit duration modeling. I don't think this paper has explicitly compared "autoregressive modeling" with "methods without explicit duration modeling" since "autoregressive modeling" also belongs to "methods without explicit duration modeling" (unless your definition of "methods without explicit duration modeling" is different from mine). I believe most existing literature defines "methods without explicit duration modeling" as methods that require a separate model that predicts the duration.

---

> ### Comment · Reviewer_pGsx · 2024-11-20
> **Response to Authors' Rebuttal Regarding "Weakness 2"**
>
> > We made this claim based on: 1) F5-TTS small, a model with fewer parameters and less training data, has a 0.21 drop in SIM-O and 2.5% gain in WER on the Seed-TTS test-zh set, which is significantly worse than F5-TTS;
>
> F5 is relatively new and was not cited in the paper, so please kindly cite F5-TTS as evidence for your reasoning.
>
> Moreover, for recent papers (including submissions to ICLR 2025), there are a couple of counter-examples to your claims:
>
> >the reproduced E2-TTS [5] in F5-TTS's paper has a WER of 2.95%, which is significantly worse than that of NaturalSpeech 3 (1.81%) and our S-DiT (1.84%);
>
> DMDSpeech (500M) and A$^2$-Flow (450M) both do not use any explicit duration modeling and have achieved a WER of 1.94% and 2.2%, respectively, which are not that different from S-DiT (1.84%) and NaturalSpeech (1.81%). A$^2$-Flow also reproduced E2-TTS (450M) that has a WER of 1.9%. NaturalSpeech 3 also has 500M parameters (same as DMDSpeech) and even more than E2-TTS  reproduced in the A$^2$-Flow paper.
>
> > the reproduced version of E2-TTS and the official F5-TTS are trained on a 100k hour dataset, which is significantly larger than those of ours and NaturalSpeech 3. Directly comparing their results with ours might be unfair. Therefore, to achieve the same level of WER, "Diffusion w/o PA" methods like E2-TTS require more parameters.
>
> DMDSpeech was trained on LibriLight (same dataset as NaturalSpeech 3) and has achieved comparable WER, while A$^2$-Flow was pre-trained on 40k hours of speech and then fine-tuned on 500 hours of speech (less than NaturalSpeech 3). E2-TTS reproduced in A$^2$-Flow paper was also trained with 40k hours of data.
>
> Hence, I believe these claims cannot be supported by recent developments in the field unless new experiments are conducted specifically to compare the number of parameters for "diffusion w/o PA" methods.
>
> **References**:
>
> DMDSpeech: https://openreview.net/forum?id=LhuDdMEIGS
>
> A$^2$-Flow: https://openreview.net/forum?id=e2p1BWR3vq
>
> >we compare F-LM's processing time with that of a traditional frontend pipeline, which consists of an ASR model (SenseVoice small [1]), a phonemizer, a speech-text aligner (MFA), and an auto-regressive duration predictor
>
> >The results, shown in the following table, indicate that our model achieves a 5.1x speed-up by significantly reducing the computational time required by speech-text aligning.
>
> As far as I know, most TTS models do not require MFA (such as NaturalSpeech 3). If we remove MFA from the "traditional pipeline", the inference speed can be twice as fast as F-LM. Moreover, "diffusion w/o PA" does not require phonemization nor duration prediction, and as demonstrated by recent papers, the difference in performance is insignificant for significantly increased inference speed with explicit phonemization and duration prediction using an autoregressive LM.
>
> Bringing up further acceleration is unfair because the traditional pipeline (assuming using "diffusion w/o PA" so no duration prediction, MFA, and phonemization) can be further accelerated too (such as ASR using encoder/decoder models like Whisper with flash attention and speculative decoding or CTC models with faster decoding algorithms, etc.).

---

> ### Comment · Reviewer_pGsx · 2024-11-20
> **Concerns over RTF**
>
> Thanks for clarifying the RTF calculation. However, I believe this is a rather unfair comparison in terms of RTF. As far as I know, the entire preprocessing pipeline (including phonemization and duration prediction) is counted in many TTS models for RTF calculation (such as NaturalSpeech 3), while the ASR is not counted as we assume the prompt transcription is given. For this reason, the actual RTF is 0.432 (compared to 0.3 for NaturalSpeech 3).
>
> Given that the model does not outperform NaturalSpeech 3 significantly (even slower in inference speed and more distortions in generated speech) with more complicated training pipelines, I do not see this paper contributing much to the field. Moreover, the paper still contains many unsupported claims even after the initial revision.

---

> ### Author Response · Authors · 2024-11-20
> **Response to Reviewer pGsx about Additional Concerns (Part 1/2)**
>
> Thank you so much for your valuable comments and precious time.
>
> **[About Confusion Regarding Expressiveness]**
> 1) Yes, we define "expressiveness" as the model's capacity to model the pitch and duration (prosodic information or the variance information defined in FastSpeech 2 [1]). We acknowledge that this definition may be misleading, as "expressiveness" can easily be interpreted as the expression of emotions. We sincerely apologize for any confusion caused by this inaccurate definition. To make it more exact and accurate, we have replaced the phrase "limited expressiveness" with "constrain the search space of the generated speech and sacrifice the prosody and naturalness".
> 2) To solve this main confusion, we have cited Seed-TTS instead of SimpleSpeech 2. Thanks for your kind suggestions!
> 3) We agree that our definition may be confusing. In the claim related to ARDiT, we just intend to express that: compared to models with autoregressive modeling (ARDiT-B=4), methods without the autoregressive modeling process and without the pre-defined phoneme-level alignments (ARDiT-B=INF) exhibit some decline in speech intelligibility.
>
> **[About F5-TTS]**
> Thanks for your advice. We have cited F5-TTS in the Background section of our paper.
>
> **[About DMDSpeech and A^{2}-Flow]**
> 1) The teacher model of DMDSpeech (based on a "diffusion w/o PA" framework as described in Line 131) achieves a WER of 9.51%, while the student model with CTC direct metric optimization has achieved a WER of 1.94%, demonstraing the effectiveness of CTC loss. As described in Line 123, CTC loss is used to optimize text-speech alignment. DMDSpeech use the soft alignments from ASR model to calculate CTS loss and our model directly use sparse alignment as inputs to the model. So DMDSpeech is not a counter-example to our claim in terms of speech intelligibility;
> 2) We agree that A^{2}-Flow and its reproduced E2-TTS (450M) perform well in terms of WER when the target transcriptions are sourced from LibriSpeech. However, these transcriptions are relatively simple since they come from audiobooks. To further indicate the speech intelligibility of different methods, we follow F5-TTS to evaluate our model on the challenging set containing 100 difficult textual patterns from ELLA-V [2]. Since the speech prompts used by ELLA-V are not publicly available, we randomly sample 3-second-long speeches in LibriSpeech test-clean set as speech prompts. For this evaluation, we used the official checkpoint of F5-TTS and the inference API of E2-TTS provided on F5-TTS's Hugging Face page. We employ Whisper-large-v3 for WER calculation. Since A^{2}-Flow is also a submission to ICLR 2025 and has not yet released their source code, we do not report its results in this evaluation. Based on the results presented in the table below and our previous responses, our claims regarding speech intelligibility are well-supported.
>
> | Model  | WER | Substitution | Deletion | Insertion|
> | :------------: | :------------: |  :------------: |  :------------: |  :------------: |
> | E2 TTS  | 8.49 | 3.65 | 4.75 | 0.09 |
> | F5-TTS  | 4.28 | **1.78** | 2.28 | 0.22 |
> | S-DiT   | **3.95** | 1.80 | **2.07** | **0.08** |
>
> **[About MFA used in NaturalSpeech 3]**
> Your statement, ``As far as I know, most TTS models do not require MFA (such as NaturalSpeech 3).``, appears to be inaccurate, as NaturalSpeech 3 does require MFA during inference. For fair comparisons, the authors of NaturalSpeech 3 explicitly excluded the time required by MFA in the reported RTF values in Table 10 of their paper. This has been verified by one of the authors of NaturalSpeech 3, and screenshots of the relevant emails can be found in the following link: [https://drive.google.com/file/d/1uBupMuwL7bNv0wqqzxfYwUdqZCOSaIk8/view?usp=sharing](https://drive.google.com/file/d/1uBupMuwL7bNv0wqqzxfYwUdqZCOSaIk8/view?usp=sharing). Given this, the proposed F-LM is indeed practically useful for "diffusion w/ PA" methods.

---

> > ### Comment · Reviewer_pGsx · 2024-11-20
> >
> > **About Intelligibility Claims**
> >
> > >methods without the autoregressive modeling process and without the pre-defined phoneme-level alignments (ARDiT-B=INF) exhibit some decline in speech intelligibility.
> >
> > Again, recent submissions at ICLR 2025 such as A$^2$-Flow and DMDSpeech do not support this claim, as they have achieved similar WER as NaturalSpeech 3 and S-DiT. I don't think one can derive this conclusion from a single experiment performed in a paper that does not even include any duration modeling (i.e., this paper is a comparison between "diffusion w/o PA" and "AR LM", which has nothing to do with explicit duration prediction).
> >
> > > DMDSpeech use the soft alignments from ASR model to calculate CTS loss and our model directly use sparse alignment as inputs to the model.
> >
> > DMDSpeech does not need soft alignment from CTC during inference, and it does not have **explicit** duration modeling (which is defined by the author as "Diffusion w/ PA").
> >
> > >We agree that A^{2}-Flow and its reproduced E2-TTS (450M) perform well in terms of WER when the target transcriptions are sourced from LibriSpeech. However, these transcriptions are relatively simple since they come from audiobooks.
> >
> > > Since A^{2}-Flow is also a submission to ICLR 2025 and has not yet released their source code, we do not report its results in this evaluation. Based on the results presented in the table below and our previous responses, our claims regarding speech intelligibility are well-supported.
> >
> > This is why I think the claim cannot be made to compare "Diffusion w/ PA" with "Diffusion w/o PA" in terms of intelligibility. A$^2$-Flow reproduced E2-TTS with a WER of 1.9%, while S-DiT achieved a WER of 1.84%, not much difference. DMDSpeech has a similar WER (1.92%), and in one ablation study, the WER is as low as 1.74%. I believe this shows that even without any duration input to the model, the model still works well for intelligibility for "Diffusion w/o PA" with a similar amount of parameters. The E2-TTS reproduced in F5-TTS paper, as far as I know, only has 300M parameters, which is not comparable to 500M of S-DiT, making it an unfair comparison. The only fair I believe can support the claim for numbers of parameters is a comparison between S-DiT without explicit duration input and S-DiT with it **in terms of the number of parameters**. However, no such experiment was conducted in the paper. Hence, I believe the claim that "Diffusion w/o PA" needs more parameters to achieve good intelligibility is unsupported.
> >
> > **About MFA**
> >
> > I apologize for the misunderstanding of NaturalSpeech 3 architecture as it is not explicitly stated in the paper. Despite this, there are still many "Diffusion w/ PA" models that do not require MFA. As far as I know, StyleTTS-ZS does not require MFA as it is largely based on StyleTTS 2, where no MFA is required. Hence, the elimination of MFA for "Diffusion w/ PA" is not any new contribution.
> >
> > More importantly, all "Diffusion w/o PA" models require no MFA, phonemization, or duration prediction. For example, DMDSpeech has achieved an RTF of 0.07 as "Diffusion w/o PA," which I believe is the "real RFT" (which means it includes preprocessing too as it has close to no preprocessing) with similar performance. Given the minimal performance improvement compared to recent "Diffusion w/o PA" methods at the cost of significantly increased inference time, I still believe that the proposed F-LM does not contribute much to the field.
> >
> > > Besides, we only stated that "traditional pipelines like MFA" can not adopt these techniques
> >
> > I believe the recent alignment, such as WhisperX, can apply a similar technique (as it is a transformer), and the forced alignment is very similar to CTC (NAR), so it is already fast enough. Overall, the optimization space for the proposed AR model is probably slower compared to NAR models (like WhisperX aligner) regardless of optimization.

---

> ### Author Response · Authors · 2024-11-20
> **Response to Reviewer pGsx about Additional Concerns (Part 2/2)**
>
> **[About Further Acceleration]**
> We would like to clarify that we never claimed to compare the inference efficiency of our method with the pipeline of "diffusion w/o PA" in our previous response. Instead, we only stated that: ``compare F-LM's processing time with that of a traditional frontend pipeline, which consists of an ASR model (SenseVoice small [1]), a phonemizer, a speech-text aligner (MFA), and an auto-regressive duration predictor``. This traditional pipeline, which requires MFA, is also widely adopted by "Diffusion w/ PA" methods like NaturalSpeech 3. Therefore, the comparison studies are fair and meaningful for "Diffusion w/ PA" methods. Besides, we only stated that "traditional pipelines like MFA" can not adopt these techniques, we did not state that "ASR" can not adopt them.
>
>
> **[About Objective Evaluations]**
> We use 40 samples (identical test samples obtained from the authors of NaturalSpeech 3) to ensure that the experimental setup is the same as that described in Section 4.1 of NaturalSpeech 3. We also conduct experiments on the LibriSpeech test-clean 2.2-hour subset (following the setting in VALL-E 2 and Voicebox) to determine the actual SIM-O of the model, the results are shown in the following Table.
>
> | Models| SIM-O | WER |
> | :------------: | :------------: | :------------: |
> | VALL-E 2     |  0.643    |  2.44% |
> | MELLE        |  0.625    |  2.10% |
> | DiTTo-TTS    |  0.627    |  2.56% |
> | Voicebox     |  0.662   | 1.9% |
> | S-DiT        |  **0.697**   | **1.87%** |
>
>
> **[About Demographics of CMOS Evaluation]**
> When creating the Amazon Mturk project, we specify additional qualifications that workers must meet: ``Good at evaluating quality of English audios``. We also require that raters should be master workers on Mturk. We are sorry that informations of our project like where were the raters located can not be obtained from Amazon Mturk. The CMOS -0.10 may be due to these factors that affects naturalness and clarity: failure cases of NaturalSpeech 3's duration prediction, e.g., the NaturalSpeech 3's example ``His death in this conjuncture was a public misfortune.`` in ``Rebuttal: Comparisons with NaturalSpeech 3`` is abruptly stopped without fully pronouncing the word "misfortune".
>
> **[About Concerns over RTF]**
> As stated above, the RTF calculation of NaturalSpeech 3 paper does not include the MFA process, which is the slowest process in the preprocessing stage. Your statement that ``our model is even slower in inference speed`` might be a misunderstanding. We agree that RTF in our Table 2 do not include the phonemization and duration prediction, which is rather unfair. Including the phonemization and duration prediction, the S-DiT has a RTF of 0.208. Thanks for pointing our errors and we have update the results in Section 4.2.
>
> In terms of the statement that ``our model has more complicated training pipelines``, the speech compression model of S-DiT does not require the complicated disentangling training process of NaturalSpeech 3. Besides, one of the key component of NaturalSpeech 3 - gradient reversal - is also very complicated and hard to implement. Therefore, our training pipelines is simpler than NaturalSpeech 3.
>
>
> Apart from the clarity issues in some claims pointed out by the reviewer, we still have the following contributions to the speech field:
> 1) we propose a sparse alignment enhanced latent diffusion transformer mode which maintains the robustness and reduces the search space constraints from forced alignments;
> 2) we propose a multi-condition CFG strategy for modulating the intensity of personal accents, offering novel solutions for accent control;
> 3) our proposed F-LM not only simplifies the inference process of zero-shot TTS models like NaturalSpeech 3, but also can be directly used for processing training data during model fine-tuning. The unified training framework enhances F-LM’s speech understanding capabilities, allowing it to surpass the independent modules for each subtask.
>
> Again, we appreciate the reviewer's helpful comments that greatly enhance the clarity of our paper. We hope that the above discussion can address the reviewer's concerns.
>
>
> **[Reference]**
> [1] Ren, Yi, et al. "Fastspeech 2: Fast and high-quality end-to-end text to speech." arXiv preprint arXiv:2006.04558 (2020).
> [2] Song, Yakun, et al. "Ella-v: Stable neural codec language modeling with alignment-guided sequence reordering." arXiv preprint arXiv:2401.07333 (2024).

---

> ### Comment · Reviewer_pGsx · 2024-11-20
>
> **About Objective Evaluations**
>
> Thanks for the clarification. I believe the methods used in Vall-E and VoiceBox make more sense. Please use this number instead of the one reported in the paper.
>
> **About CMOS**
>
> > failure cases of NaturalSpeech 3's duration prediction, e.g., the NaturalSpeech 3's example "His death in this conjuncture was a public misfortune". in Rebuttal: Comparisons with NaturalSpeech 3 is abruptly stopped without fully pronouncing the word "misfortune".
>
> Is this -0.10 statistically significant? I think if only one or two samples out of 40 are cut abruptly, it should not affect the performance that much. From samples I listened to (of NaturalSpeech 3), the only abrupt cut is this example of "His death in this conjuncture was a public misfortune". All other samples are incredibly good (see sources below):
>
> - https://speechresearch.github.io/naturalspeech3/ (contains 8 samples from LibriSpeech, no cutoff)
> - https://www.microsoft.com/en-us/research/project/e2-tts/ (contains 7 samples from NaturalSpeech 3, no cutoff)
> - https://styletts-zs.github.io/ (contains 8 samples from NaturalSpeech 3, only one cutoff, which is the same sample as yours)
>
> **About RTF**
>
> Thanks for the clarification. This RTF is better but at the cost of lower sound quality, and the majority of the processing comes from the F-LM which is not necessary if the proposed method is not "Diffusion w/ PA". F5-TTS has a RTF of 0.15 and since it does not compress mel-spectrograms into another latent (two decoders needed to synthesize latent back to waveform), the sound quality is much better than S-DiT. F5-TTS has a better RTF than S-DiT and higher sound quality.
>
> **About Contribution**
>
> > 1. we propose a sparse alignment enhanced latent diffusion transformer mode which maintains the robustness and reduces the search space constraints from forced alignments;
>
> I agree with this contribution, which I believe is the main contribution of the paper. It shows that sparse (masked) alignment is better than explicit alignment.
>
> > 2. we propose a multi-condition CFG strategy for modulating the intensity of personal accents, offering novel solutions for accent control;
>
> This has already been done in DualSpeech (Yang et al. 2024). I don't think this can be counted as a contribution.
>
> > 3. our proposed F-LM not only simplifies the inference process of zero-shot TTS models like NaturalSpeech 3, but also can be directly used for processing training data during model fine-tuning. The unified training framework enhances F-LM’s speech understanding capabilities, allowing it to surpass the independent modules for each subtask.
>
> Again, "Diffusion w/o PA" does not need this F-LM preprocessing pipeline, and the proposed methods do not outperform significantly compared to recent "Diffusion w/o PA" methods but with a more complicated training pipeline (preparation of ground truth duration, mel-spectrogram autoencoder, etc.) and likely increased inference speed (compared to F5-TTS and DMDSpeech).
>
> Given these concerns, I think the only laudable contribution of this paper is that masked (sparse) duration input to "Diffusion w/o PA" model is better than full duration input. However, this sounds obvious and can be seen as a regularization, and the performance is not significantly better than other alternatives that do not employ this method. Given the minor contribution (masking the duration input to "Diffusion w/o PA" methods), I still think the significance of this paper is incremental and the contribution is unclear.
>
> Reference:
>
> Yang, J., Lee, J., Choi, H. S., Ji, S., Kim, H., & Lee, J. (2024). Dualspeech: Enhancing speaker-fidelity and text-intelligibility through dual classifier-free guidance. arXiv preprint arXiv:2408.14423.

---

> ### Author Response · Authors · 2024-11-21
> **Further Response to Reviewer pGsx**
>
> **[Experiments in Terms of the Number of Parameters]**
> We conducted the following experiments using pretrained S-DiT models with varying numbers of parameters.
> The test set used is the 2.2-hour test-clean subset of LibriSpeech, following the setup in Voicebox.  The results are presented in the table below. It can be observed that S-DiT without explicit duration modeling requires a larger number of parameters to achieve good intelligibility, which supports our claim.
>
> | Models| #Params (DiT only)  | WER |
> | :------------:              | :------------: | :------------: |
> | S-DiT w/ sparse alignment   | 0.34B | **1.87%** |
> | S-DiT w/o explicit duration | 0.34B | 2.41% |
> | S-DiT w/o explicit duration | 0.70B | 2.02% |
> | S-DiT w/o explicit duration | 1.34B | 1.95% |
>
>
> **[About StyleTTS-ZS]**
> Yes, StyleTTS-ZS does not require MFA. However, 1) its SIM-O score is quite low, 0.56 of StyleTTS-ZS (LibriLight) compared with 0.66 of NaturalSpeech 3; 2) StyleTTS-ZS was published on ArXiv only two weeks before the ICLR 2025 submission deadline, any concurrent works submitted so close to the ICLR 2025 deadline should not be used to evaluate the contributions of our work.
>
> **[About Contribution]**
>
> 1) Thanks for your comments that sparse (masked) alignment is better than explicit alignment and considering it as a laudable contribution;
> 2) DualSpeech [1] is a concurrent work that was published on Arxiv on 26 Aug 2024 (only 1 month before the submission deadline of ICLR 2025).
> Besides, DualSpeech is limited to improving the audio quality of TTS and does not explore the impact of CFG on accents.
> Our work reveals the patterns of accent variation under different CFG weight proportions and the proposed multi-condition CFG can be used for accent control.
> Besides, Reviewer syxD also comment that ``Multi-condition CFG seems particularly useful in code-switching``.
> Moreover, concurrent works like DualSpeech also reveals the importance of this strategy. Therefore, the proposed multi-condition CFG is also a laudable contribution to speech community;
> 3) F-LM is a good contribution to methods that rely on explicit alignments. These methods have demonstrated strong speech intelligibility in handling challenging cases and remain an important research direction in text-to-speech synthesis.
>
> Once again, we emphasize that concurrent works submitted so close to (e.g., Dualspeech, StyleTTS-ZS) or even after (e.g., F5-TTS, DMDSpeech, A^{2}-Flow) the ICLR 2025 deadline should not be used as a basis for evaluating the contributions of our work, as this undermines the fairness and guideline of the ICLR review process.
>
>
> **[Reference]**
> [1] Yang, Jinhyeok, et al. "Dualspeech: Enhancing speaker-fidelity and text-intelligibility through dual classifier-free guidance." arXiv preprint arXiv:2408.14423 (2024).

---

> ### Comment · Reviewer_pGsx · 2024-11-21
>
> **Additional Experiments on Intelligibility**
>
> Thanks for conducting additional experiments. This now addresses my concern over intelligibility.
>
> **About Concurrent Works**
>
> I understand that works that appeared 3 months before ICLR submission deadlines shall not be used to evaluate contributions. Let me summarize the contribution, **without any reference to papers submitted to ICLR nor anything appearing 3 months before the ICLR 2025 deadline (i.e., papers appearing before July 2024 are counted)** (unless an old paper was reproduced in new works with similar performance, supporting the knowledge of old papers):
>
> 1. **Compared to E2-TTS**, which has achieved a WER of 1.9 and SIM-O of 0.708 (according to Eskimez et al. 2024, **published on June 26, 2024**), this work:
>    - achieved similar performance (WER $\sim$ 1.9, SIM-O $\sim$ 0.7), but with **a significantly more complicated training pipeline (requiring phonemization and duration labels, mel-spectrogram autoencoder)**, and likely **slower inference speed** due to the use of autoregressive LM for preprocessing.
>    - the authors compensated for inference speed by using a mel-spectrogram autoencoder, **further reducing the sound quality (with more distortions than E2-TTS)**, even though the **bottleneck is the newly proposed F-LM**.
>    - similar results have supported the effectiveness of E2-TTS, such as reported in A$^2$-Flow. The point of my references to concurrent works such as DMDSpeech, F5, and A$^2$-Flow is to show that  **"diffusion w/o PA" is enough to achieve good performance**. Even **without referencing these recent papers**, E2-TTS alone has already made this point clear.
>
> 2. **Compared to NaturalSpeech3**, which has achieved a WER of 1.8 and SIM-O of 0.67, this work:
>    - achieved similar performance (WER $\sim$ 1.9, SIM-O $\sim$ 0.7), but with **a more complicated training pipeline (requiring phonemization labels and mel-spectrogram autoencoder)**, even though the inference speed is slightly faster (due to no need of MFA).
>    - because of the use of mel-spectrogram autoencoder, the **sound quality is significantly worse**.
>
> Based on these two papers published 3 months before the ICLR deadline, to summarize the contributions:
>
> 1. The proposed sparse (masked) duration input does make sense and indeed helps with regularizing the duration learning. However, it is unclear whether this is necessary since E2-TTS as a "diffusion w/o PA" method has demonstrated similar performance (which has been reproduced several times in recent submissions such as F5-TTS and A$^2$-Flow). The only improvement seems to be in **intelligibility** and it is unclear whether this improvement is caused by suboptimal architecture or training data (since E2-TTS has achieved WER of 1.9 and it was reproduced by A$^2$-Flow).
>
> 2. The proposed Multi-Condition CFG, even without mentioning DualSpeech, **has been done in the past**. DualSpeech has cited VoiceLDM, which was published in Sept 2023. Hence, this paper is not the first to introduce multi-condition CFG, which only applies to accents. This contribution seems minor and not in the broad scope of the ICLR community, as it **just applies a well-known concept to a specific use case (accent control)**.
>
> 3. The proposed F-LM, in my opinion, is a net negative addition to this work. The F-LM combines ASR, phonemization, and duration predictors in an autoregressive language model, meaning **we must retrain these things from scratch (instead of using pre-trained models directly)**. The **performance improvement of using F-LM is minimal (compared to E2 and NaturalSpeech 3)**, and the only benefit seems to be the elimination of MFA for NaturalSpeech 3. My point of mentioning StyleTTS-ZS is to show that many zero-shot TTS models in the family of "diffusion w/o PA" do not need MFA. Given that StyleTTS-ZS was published right before the ICLR deadline, we can use HierSpeech++ as an example, let alone countless other papers that belong to the family of "diffusion w/o PA" without using MFA. Even though works like HierSpeech++ and StyleTTS demonstrate lower similarity, there is no theoretical need to use MFA to improve the similarity since E2-TTS has demonstrated that a pipeline as simple as that can have considerably good performance.
>
> **TLDR**
>
> This paper has proposed three components: sparse (masked) duration input is useful but does not seem necessary given the success of E2-TTS, dual-condition CFG is not a novel concept that has been used in speech synthesis and the use of it in this paper is quite narrow, and F-LM complicates the training and inference pipeline with no significant improvement over previous papers except the elimination of MFA (and WhisperX can be a faster alternative), which again may or may not be necessary given success of recent works without MFA. Moreover, the sound quality is worse than E2-TTS and NaturalSpeech 3 due to use of melspectrogram autoencoder.

---

> ### Author Response · Authors · 2024-11-21
>
> Reviewer pGsx's concerns regarding our contributions primarily rely on the statement: ``E2-TTS reproduced by A^{2}-Flow has achieved a WER of $\sim$ 1.9 and SIM-O of $\sim$ 0.7``. However, A^{2}-Flow has not published their implementation yet. On the other hand, E2-TTS reproduced by F5-TTS has been **open-sourced** (WER $\sim$ 3 and SIM-O $\sim$ 0.69), which is **more convincing** in terms of current literature. The reproduced E2-TTS demonstrates significantly **lower speech intelligibility** compared with our S-DiT (WER $\sim$ 3 vs WER $\sim$ 1.9). Furthermore, in terms of the 100 hard sentences from ELLA-V, E2-TTS achieves a WER of 8.49%, whereas our method achieves a WER of 3.95%. Given the substantial intelligibility gap, Reviewer pGsx's current concerns appear to be based on misunderstandings of concurrent works.
>
> | Model  | WER | Substitution | Deletion | Insertion|
> | :------------: | :------------: |  :------------: |  :------------: |  :------------: |
> | E2 TTS  | 8.49% | 3.65% | 4.75% | 0.09% |
> | F5-TTS  | 4.28% | **1.78%** | 2.28% | 0.22% |
> | S-DiT   | **3.95%** | 1.80% | **2.07%** | **0.08%** |
>
> Another **misunderstanding** raised by Reviewer pGsx is the claim that ``Our work has a more complicated training pipeline than NaturalSpeech 3``. However, NaturalSpeech 3 involves a complicated disentanglement pipeline to train the FACodec, as well as the following components for speech generation: 1) phoneme encoder, 2)duration diffusion and length regulator, 3) prosody diffusion, 4) content diffusion, 5) detail (acoustic detail) diffusion. Obviously, the training pipeline of NaturalSpeech 3 is more complicated unless Reviewer pGsx does not really understand NaturalSpeech 3 paper. Reviewer pGsx also states that ``NaturalSpeech 3 does not require phonemization labels``, which is **clearly a misunderstanding**. Moreover, in previous discussions, Reviewer pGsx believe that ``NaturalSpeech 3 does not require MFA in inference``, which is also an obvious misunderstanding. **Considering so many factual errors**, it is evident that Reviewer pGsx's concerns related to NaturalSpeech 3 are primarily based on misunderstandings.
>
> Finally, we keep our statement unchanged. Our work has the following contributions:
> 1) we propose a sparse alignment enhanced latent diffusion transformer mode which maintains the robustness and reduces the search space constraints from forced alignments;
> 2) we propose a multi-condition CFG strategy for modulating the intensity of personal accents, offering novel solutions for accent control;
> 3) our proposed F-LM not only simplifies the inference process of zero-shot TTS models like NaturalSpeech 3, but also can be directly used for processing training data during model fine-tuning. The unified training framework enhances F-LM’s speech understanding capabilities, allowing it to surpass the independent modules for each subtask.

---

> ### Comment · Reviewer_pGsx · 2024-11-21
>
> I appreciate the author's clarification for NaturalSpeech 3. I apologize for the misunderstanding of this work, and I agree that the training pipeline is similarly complicated compared to S-DiT if **taking FACodec into consideration** (the diffusion over other attributes can be seen as the same thing and trained simultaneously, not necessarily more complicated, though FACodec training is indeed complicated, and the MFA misunderstanding is because the paper does not state it explicitlty). However, this does not negate my major concern of the contribution of this work:
>
> 1. Compared to E2-TTS, the paper does have a more complicated training pipeline (phonemization, duration prediction, ground truth labels during training) and worse sound quality given the use melspectrogram autoencoder. The CMOS reported (-0.10) compared to NaturalSpeech 3 is not well-justified given significantly worse sound quality.
>
> 2. Even if we use the open-source implementation for E2-TTS, the **only improvement is intelligibility.** I personally do not believe this improvement solely justifies the more complicated training pipeline of S-DiT (training ASR, phonemziation, and duration predictor from scratch and using ground truth labels during training). This also limits the scalability significantly compared to simple frameworks such as E2-TTS that have achieved similar performance (maybe with a little worse in intelligibility).
>
> Since the authors have addressed my concerns over weaknesses 1 and 3, I have raised my score from 3 to 5 to appreciate the efforts of the authors for rebuttal. However, given my doubt about the contribution of this work, I think this work is at best borderline and I favor a rejection over acceptance. This is based on the review standard of other papers in the field of zero-shot TTS in ICLR 2025. All following works have simpler training pipelines than S-DiT (no ASR, phonemization and duration labels during training), though some have additional steps like Text-to-Semantic + Semantic-to-Acoustic, distillation and fine-tuning from pre-trained models.
>
> - A$^2$-Flow: https://openreview.net/forum?id=e2p1BWR3vq (similar performance as S-DiT but with significantly reduced transcribed training data; it received an average score of 5.25)
>
> - DMDSpeech: https://openreview.net/forum?id=LhuDdMEIGS (similar performance as S-DiT but with significantly faster sampling speed; it received an average score of 4.75)
>
> - F5-TTS: https://openreview.net/forum?id=JiX2DuTkeU (similar performance with worse intelligibility but faster sampling speed than S-DiT; it received an average score of 5.5; moreover this work is open source so it guarantees accuracy and reproducibility)
>
> - Vall-E 2: https://openreview.net/forum?id=0bcRCD7YUx (better performance than S-DiT in both intelligibility and similarity; it received an average score of 5)
>
> - MaskGCT: https://openreview.net/forum?id=ExuBFYtCQU (similar performance as S-DiT but significantly slower sampling speed; it received an average score of 5.25)
>
> Overall, I think this paper has more issues than merits. I will leave the judgment of the value of this work to the AC.

---

> ### Author Response · Authors · 2024-11-22
> **Some Clarifications**
>
> First of all, we sincerely thank Reviewer pGsx for his helpful suggestions and respect his borderline recommendation. We agree that, compared to E2-TTS, S-DiT indeed involves a more complicated training pipeline. We also acknowledge that both E2-TTS and F5-TTS have simple structures while achieving strong performance.
>
> There are some key points we need to further clarify:
> Reviewer pGsx claimed that ``E2-TTS has achieved similar performance (maybe with a little worse in intelligibility.`` and ``Compared to open-source implementation for E2-TTS, S-DiT's only improvement is intelligibility. I personally do not believe this improvement solely justifies the more complicated training pipeline of S-DiT``. However, the open-source implementation of E2-TTS reproduced by F5-TTS achieves a WER of 2.95%, while our S-DiT achieves a WER of 1.9%. A WER gap of 1.05% is indeed significant.Furthermore, in our experiments for the 100 hard sentences from ELLA-V, E2-TTS achieves a WER of 8.49%, whereas our method achieves a lower WER of 3.95%.
>
> To clearly illustrate the speech intelligibility gap between E2-TTS and S-DiT, we compare our method with the open-source implementation for E2-TTS in the ``Rebuttal: Comparisons with E2-TTS`` section on our demo page. All the generated speeches are resampled to 16khz. We can see that in row 4$\sim$6, there are many mispronounced words in E2-TTS's samples. Samples in row 1$\sim$6 also demonstrates that S-DiT delivers better quality compared to E2-TTS. Additionally, as metioned by Reviewer hj2E, ``By compressing the spectrogram into a latent representation that is 8 times shorter through speech compression, the model can be trained more efficiently.``. Therefore, the use a melspectrogram autoencoder should not be considered as a weakness of our method.

---

> ### Comment · Reviewer_pGsx · 2024-11-22
>
> Again, in both the original paper of E2-TTS and reproduced results in A$^2$-Flow, E2-TTS has a WER of 1.9% compared to 2.95% in F5-TTS, which shows that E2-TTS can achieve good intelligibility with better implementation and dataset. It is possible the training data (Emilia) has more transcription errors, and it is also multilingual, so it is unfair to compare to models trained in different datasets and possibly different implementations. Since some papers have already reproduced the results of E2-TTS with a WER of 1.9%, I think we should stick to 1.9% when judging the contribution of this work.
>
> With a much more complicated training pipeline and worse sound quality due to melspectrogram autoencoder, the only improvement over previous work is some intelligibility (which may or may not be caused by different training data and implementation). I'd like to point out that the "efficiency" in training introduced by melspectrogram autoencoder does not compensate for the inefficiency of training using MFA duration labels and F-LM (phonemization and ASR transcripts). Overall, the proposed model has worse sound quality and a more complicated training pipeline than E2-TTS, with the only improvement being intelligibility, which may or may not be caused by different training data or implementation.
>
> For this reason, I do not think this paper meets the quality of ICLR specifically for speech synthesis. I would like to emphasize that the **bar for speech synthesis for conferences like ICLR is extremely high**. As far as I know, *no TTS paper* has been accepted in NeurIPS 2024, and the average score of zero-shot TTS submissions at ICLR is *5*, with the highest being *5.5*. Hence, a 5 is the best I could give as papers with more merits than S-DiT received lower scores in this venue. Note that only the score rating is based on other submissions at ICLR, and the judgment of contribution does not include papers submitted 3 months after the ICLR deadline. When judging the contribution, the only reference model is E2-TTS, which was published 3 months before the ICLR deadline.
>
> I hope the AC checks the reviews of other papers in the field of zero-shot TTS with better performance and more merits than this paper and makes their own decision. My recommendation **leans towards rejection**, although I will not be disappointed if it is finally accepted, given large conferences like ICLR have historically accepted many subpar papers while rejecting many other more valuable papers due to systematic issues in the review process, and the authors have put a lot of effort into revising the paper and addressing the reviewers' concerns, which I believe is invaluable and much appreciated. However, the contribution clearly does not meet the bar for acceptance in ICLR as a speech synthesis paper. Due to limited contributions, I would suggest the authors submit this paper to a different venue.

---

### Official Review · Reviewer_hj2E · 2024-11-04

**Soundness:** 3
**Presentation:** 3
**Contribution:** 4
**Rating:** 6
**Confidence:** 5

**Summary:**

This paper proposes the S-DiT model, which, although it utilizes predefined alignment, enables expressive TTS comparable to models that learn alignment implicitly. Unlike previous models that use a phoneme duration predictor and rely on dense alignment, S-DiT employs sparse alignment by providing only partial alignment information randomly. This approach allows the model to learn expressively while maintaining alignment robustness. To provide phoneme duration prediction and a transcript for prompts, the paper introduces a frontend language model (F-LM) framework and adopts piecewise flow matching to accelerate the sampling process. The model demonstrates high speaker similarity and maintains good pronunciation accuracy in zero-shot TTS tasks, with significantly fast sampling speeds.

**Strengths:**

* By compressing the spectrogram into a latent representation that is 8 times shorter through speech compression, the model can be trained more efficiently and generates shorter sequences during inference, offering advantages in terms of inference speed.

* The model retains the robustness of previous models that use duration predictors while overcoming their limitations in expressiveness by introducing a sparse alignment mechanism. This allows it to achieve performance comparable to models that learn alignment implicitly.

* The introduction of F-LM simplifies the inference process by predicting phoneme durations and representations using only the prompt audio and the target text sentence, eliminating the need for a transcript of the prompt.

* The paper shows that sampling speed can be further accelerated using piecewise rectified flow.

* The paper shows that the model's performance progressively improves with the scale of the model and the amount of training data.

**Weaknesses:**

* To provide alignment information during training, the Montreal Forced Aligner (MFA) is additionally required, which may limit the model’s extension to other languages depending on the aligner’s performance.

* In the Method section, the descriptions of each module are quite simplified, making the process of inputting each condition into the model only roughly understandable and not precisely detailed. For instance, the explanation of whether the model performs masked speech modeling is very brief. Additionally, it is unclear how the rough alignment information is provided to S-DiT. In Figure 1 (b), it appears that a representation of the text token from F-LM is also provided to S-DiT, but these aspects are not clearly explained.

**Questions:**

* To provide rough alignment during inference, is alignment provided randomly? If so, I wonder how the samples change depending on the method of providing rough alignment.

* Was phoneme duration stochastically sampled from F-LM? If it was sampled deterministically, how does the performance change when using stochastic sampling?

* When training on 600k hours of speech data, how was the duration label obtained and provided?

Typos
* In equation (5), x_t -> z_t.

* In Appendix E, in the Special Tokens section, <End of BPE]> ->  <End of BPE>

---

> ### Author Response · Authors · 2024-11-18
> **Response to Reviewer hj2E**
>
> We thank the reviewer for the positive review and constructive feedback, and we hope our response fully resolves your concerns.
>
> **[About Weakness 1]**
> Yes, when extending to other languages, this system does rely on the performance of the aligner. When extending TTS systems to other languages, dataset size is a significant factor, as many languages lack large datasets. Therefore, we conducted ablation studies for *Ours w/ Sparse Alignment*, *Ours w/ Forced Alignment*, and *Ours w/o Alignment* in both small-scale and large-scale dataset scenarios. The results are shown in the table below. When the dataset size is small, *Ours w/o Alignment* struggles to learn robust text-to-speech alignment through cross-attention while models with alignments perform well, indicating that external aligners are necessary to ensure the overall system's performance on small-scale datasets. Then, on larger datasets, the performance of the alignment-based systems also surpasses that of the *Ours w/o Alignment* setting. Therefore, even though S-DiT relies on the performance of the aligner, it remains one of the best solutions currently.
>
> | Setting | Dataset | SIM-O | WER |
> | :------------: | :------------: | :------------: | :------------: |
> | Ours w/ Forced Alignment | LibriTTS (585 hours)  | 0.57 | 2.29%|
> | Ours w/ Sparse Alignment  | LibriTTS (585 hours)  |  0.55 | 2.33% |
> | Ours w/o Alignment          |  LibriTTS (585 hours)  | 0.46 | 4.61% |
> | Ours w/ Forced Alignment | LibriLight (60k hours) |  **0.67** | **1.82%**|
> | Ours w/ Sparse Alignment  | LibriTTS (60k hours)  |  **0.67** | 1.84% |
> | Ours w/o Alignment          | LibriTTS (60k hours)   | 0.61 | 2.55% |
>
>
> **[About Weakness 2]**
> We are sorry for the lack of details in the Method section. 1) We have added detailed explanations about the masked speech modeling process of S-DiT; 2) The rough alignment information is downsampled to match the length of the latent sequence. Then, we directly concatenate the downsampled rough alignment information and the latent sequence along the channel dimension; 3) We have replaced the *text token* in Figure 1 (b) with *phoneme token*. The phoneme token sequence is concatenated with the latent sequence along the time dimension as the prefix information. We have added these details in the revised version of the paper and marked them in red.
>
>
> **[About Question 1]**
> Yes, during inference, we randomly provide rough alignments following the way in the training stage. Here, we generate samples using the following methods for providing rough alignment: 1) uniform sampling from the original region; 2) using only the middle frame of the original region; 3) selecting the left or right boundary frame as the anchor of sparse alignment. The results are provided in the ``Rebuttal: Methods of Providing Rough Alignment`` section on the demo page. It can be seen that different methods show only slight perceptual differences, as the anchor for rough alignment is chosen randomly during training. However, case 2) appears to slightly exhibit less diversity in terms of duration.
>
>
> **[About Question 2]**
> As described in Appendix A.3, we apply top-50 stochastic sampling for duration prediction to enhance output diversity. Here, we compare top-1 deterministic sampling with top-50 stochastical sampling for F-LM's duration prediction. We report the mean duration errors, 95% confidence interval of duration errors, SIM-O, WER, and CMOS in the following table. We can see that compared to top-50 sampling, deterministic sampling has a lower average error; however, due to the reduced variance in duration, diversity decreases, resulting in a minor reduction in the CMOS score.
>
> | Setting  | Phoneme-Level Error (ms) | Sentence-Level Error (s) | SIM-O | WER | CMOS |
> | :------------: | :------------: |  :------------: |  :------------: |  :------------: |  :------------: |
> | Top-1 Sampling | **18.32 ± 0.78**  | **1.31 ± 0.62** | **0.67** | **1.82%** | -0.07|
> | Top-50 Sampling  | 18.80 ± 0.94 |  1.59 ± 0.74 | **0.67** | 1.84% | **0.00**|
>
>
> **[About Question 3]**
> Since training an MFA model directly on a 600k-hour dataset is impractical, we randomly sampled a 10k-hour subset from the dataset to train a robust MFA model, which is then used to obtain the duration label for the full dataset. Since data processing inherently requires some alignment model (such as an ASR model) for speech segmentation, using a pretrained MFA model for alignment extraction does not significantly limit the system's data scalability. When training on 600k hours of speech data, the duration label is provided to our model in the same way described in Section 3. We have added this detail in Appendix D and marked them in red.
>
> **[About Typos]**
> We are sorry for our typos. We have revised them in the new version of the paper.

---

> ### Author Response · Authors · 2024-11-22
> **Dear Reviewer hj2E**
>
> Thanks for your expertise and precious time! Since the discussion phase is ending soon, it would be grateful if we could hear your feedback regarding our answers. We would be happy to answer and discuss if you have further comments or questions.

---

> ### Author Response · Authors · 2024-11-28
> **Dear Reviewer hj2E**
>
> We sincerely apologize for disturbing you, and we understand that the workload for each reviewer is quite substantial. As the discussion phase is about to conclude, please let us know if there are any remaining concerns. We are open to further questions and would be happy to engage in additional discussions. Given that our paper is currently borderline, your recommendation is extremely important to us.

---

> > ### Comment · Reviewer_hj2E · 2024-11-28
> >
> > The authors seem to have addressed most of the questions and concerns. Weakness 2 and 3, as well as Question 1, 2, and 3, appear to have been resolved effectively, and I've confirmed these through the updated version of the paper.
> >
> > Regarding Weakness 1, since MFA is used, it remains true that the model incurs a greater training burden compared to models that learn alignment internally through self/cross-attention. As Reviewer pGsx pointed out, the E2TTS model, which was released on arXiv three months prior to ICLR submission, performed alignment learning using the same amount of data. Thus, the "Ours w/o alignment" results provided by the authors in Table 7 and Figure 4 for the large-scale dataset may not represent optimal outcomes. Since E2TTS was mentioned during the review process, claiming in Table 1 that the Diffusion w/o PA model with comparable parameters is "parameter efficient" appears to be a stretch. While the E2TTS results from F5-TTS were presented on its demo page, they seem to perform worse in terms of objective metrics compared to Microsoft’s official E2TTS. Therefore, comparisons with the E2TTS released by F5-TTS cannot be considered direct comparisons with E2TTS.
> >
> > Nonetheless, the approach of modeling alignment through random sparse alignment, allowing the model to utilize the search space internally, presents a strong alternative to existing methods that use duration predictor. While the necessity for a high-performing duration predictor, such as F-LM, remains a limitation to consider, this method offers a promising alternative for scenarios where robust performance is critical or cases requiring phoneme-level duration control. Compared to E2TTS, which models alignment continuously, questions remain as to whether the proposed method’s search space is entirely unconstrained. However, based on the results demonstrated by the current model, I maintain my evaluation as borderline for ICLR, marginally above, and will keep my score at 6.

---

> > > ### Author Response · Authors · 2024-11-29
> > >
> > > Thank you for your thoughtful and detailed feedback. We appreciate your acknowledgment that most of the weaknesses and questions have been addressed effectively. Regarding Weakness 1 and the comparison with E2TTS, we understand your concerns and agree that a direct comparison with Microsoft’s official E2TTS would provide a more accurate evaluation. We are sorry that due to the lack of further implementation details in E2-TTS paper, we can not reproduce the results currently. Despite these limitations, we are grateful that you recognize the potential of our random sparse alignment approach as a promising alternative for alignment modeling.
> > >
> > > We have one more point we would like to discuss. Regarding ``Compared to E2TTS, which models alignment continuously, questions remain as to whether the proposed method’s search space is entirely unconstrained``, we believe that the experiments in Appendix N demonstrate that the search space of our method is **sufficiently unconstrained**. And as to whether it is **entirely unconstrained**, our answer is no. We believe that the current anchor-based constraints enable refined phoneme-level duration control and enhance the robustness of our model, which are important advantages compared to E2-TTS. Finally, thank you for taking the time to review our response.

---

### Official Review · Reviewer_pQoE · 2024-11-04

**Soundness:** 3
**Presentation:** 3
**Contribution:** 2
**Rating:** 6
**Confidence:** 3

**Summary:**

This paper presents S-DiT, a zero-shot text-to-speech (TTS) synthesis model that addresses critical challenges in alignment modeling. The authors introduce a sparse alignment mechanism to guide a latent diffusion transformer, balancing expressiveness and robustness without over-complicating the pipeline. The work also proposes a frontend language model (F-LM) that unifies various speech processing tasks including grapheme-to-phoneme and duration predcition, and a piecewise rectified flow technique, which reduces inference steps without significant performance degradation. Experiments show that the resulting TTS system outperforms state-of-the-art zero-shot TTS baselines in quality and efficiency.

**Strengths:**

* The sparse alignment mechanism effectively combines the advantages of predefined and non-predefined alignment strategies, improving both the expressiveness and robustness of TTS synthesis.
* The paper demonstrates effective use of piecewise rectified flow to reduce inference time without significant quality loss.
* The paper is clearly written and easy to follow, with the proposed model architecture and training algorithms well-explained. The detailed analysis of experimental results further enhances comprehension.

**Weaknesses:**

* Although precise phoneme-speech alignment is not required for training the diffusion model, it is necessary for training the frontend language model. This requirement limits the scalability of the approach to larger datasets and multilingual scenarios. In contrast, latent diffusion models without such alignment dependencies offer better scalability, while models with exact alignment perform more efficiently when data is scarce. Consequently, the proposed model may struggle with smaller-scale data and face preprocessing inefficiencies that hinder scalability as the dataset size increases.
* The paper does not adequately discuss potential failure cases in speech generation. It would be beneficial for the authors to address scenarios where the model might underperform, such as specific sparse alignment patterns that lead to failures, alignment difficulties with long text inputs, or challenges with extremely long or short phoneme durations. Understanding these limitations would provide a clearer understanding of the proposed model.

**Questions:**

* It would be helpful to include an evaluation of the reconstruction quality of the speech compression model.

---

> ### Author Response · Authors · 2024-11-18
> **Response to Reviewer pQoE**
>
> Thanks for your valuable feedback, and we hope our response fully resolves your concerns.
>
> **[About Weakness 1]**
> Yes, the S-DiT model may have the potential risk of struggling with smaller-scale data. Therefore, we conducted experiments to verify whether S-DiT performs well on small datasets. We trained it on the Libri-TTS dataset (containing a total of 585 hours of speech data) and the results are shown in the following table. It can be seen that *Ours w/ Sparse Alignment* achieves similar performance with *Ours w/ Forced Alignment*, and both methods significantly outperform *Ours w/o Alignment*, verifying the proposed model's robustness on small-scale datasets.
>
> | Setting|  SIM-O | WER |
> | :------------: | :------------: | :------------: |
> | Ours w/ Forced Alignment |  **0.57** | **2.29%** |
> | Ours w/ Sparse Alignment  |  0.55 | 2.33% |
> | Ours w/o Alignment          |  0.46 | 4.61% |
>
> Besides, latent diffusion models without such alignment dependencies also require the initial data processing stage, where the word-speech alignment (or phoneme-speech alignment) might be required for speech segmentation. For example, in the scalability experiments of this paper (Table 8), when processing 600,000 hours of data, we use a pre-trained aligner model to obtain phoneme-speech alignment for segmentation (we can also use ASR models for the word-speech aligning). Therefore, the requirements of phoneme-speech alignment do not introduce significant preprocessing inefficiencies.
>
> **[About Weakness 2]**
> We agree that the discussion about potential failure cases of S-DiT is important. For specific sparse alignment patterns that lead to failures, we investigate the following settings: 1) we only choose the left or right boundary frame as the anchor of sparse alignment; 2) we add significant gaussian noises to the predicted duration value (to simulate a poorly performing duration predictor). For challenges with extremely long or short phoneme durations and difficulties with long text inputs, we directly generate speech examples following the requirements. The generated examples are listed in the ``Rebuttal: Examples of Potential Failure Cases`` Section on our demo page, demonstrating that *Ours w/ Sparse Alignment* is robust among these scenarios.
>
>
> **[About Question 1]**
> Thanks for your helpful suggestions. We have evaluated the reconstruction quality of the speech compression model, with results presented in the following table. As suggested by reviewer syxD, we report the objective metrics, including PESQ, ViSQOL, and MCD. The table shows that, despite applying an additional 8x compression in the temporal dimension, our speech compression model's performance on various reconstruction metrics, such as PESQ and ViSQOL, remains close to that of the Encodec-16kHz model, due to the use of continuous representations and a slight KL-penalty loss during training. Moreover, it even significantly outperforms all baseline models in the MCD metric. We have included these results and discussions in Appendix J in the revised version of the paper.
>
> | Models|  Sampling rate | Hop Size | Latent Layer | Type | Bandwidth | PESQ | ViSQOL | MCD |
> | :------------:               | :------------: | :------------: | :------------: | :------------: |  :------------: |  :------------: |  :------------: |  :------------: |
> | EnCodec                   |  16kHz   | 320|  10 | Discrete      | 5.0 kbps |3.10 | 4.27 | 3.10
> | HiFi-Codec               |  16kHz    | 320 | 4 | Discrete      | 2.0 kbps | 3.17 | 4.19 | 3.05
> | DAC                          |  16kHz   | 320 | 9 | Discrete      | 4.5 kbps | **3.52** | **4.54** |  2.65
> | SoundStream            |  16kHz   | 200 | 6 | Discrete      | 4.8 kbps | 3.01 | 4.16  |  3.36
> | Ours                         |  16kHz   | **200 (x8)** | **1** | Continuous | - | 3.06 | 4.31 | **2.47**

---

> ### Author Response · Authors · 2024-11-22
> **Dear Reviewer pQoE**
>
> Thanks again for your valuable comments. Since the discussion phase is ending soon, it would be grateful if we could hear your feedback regarding our answers. We would be happy to answer and discuss if you have further comments or questions.

---

> > ### Comment · Reviewer_pQoE · 2024-11-25
> >
> > I appreciate the authors’ efforts in addressing the reviewers’ questions and concerns, including my own. Although the experimental results addressing Weakness 1 do not include multilingual TTS results beyond English and Chinese, low-resource languages for instance, I acknowledge that the experiment demonstrates the proposed method works well even with small data sizes. I thank the authors for including examples of potential failure cases, which are valuable for understanding the method’s robustness. While the reconstruction performance of the speech compression model shows comparable results measured by PESQ and ViSQOL with Encodec, a relatively earlier model, I appreciate that they included these results. This is understandable, as their model compresses inputs further along the length axis. I thank the authors for their efforts to respond and remain confident in my assessment.
> >
> > As the reviewers and authors have discussed many aspects of this work, and it seems to require my opinion, I believe the main contribution, the finding that sparse alignment enhances TTS, is valuable in that it has a moderate impact on the TTS domain but not a high impact nor influence on areas outside of TTS. As Reviewer pGsx cited, E2-TTS and A^{2}-Flow show that TTS models can learn text-speech alignment even without explicit alignments and demonstrate comparable performance with similar parameters. Although this point is debatable due to the lack of implementation details from the authors of those papers, I believe the impact and experimental results of this work should be considered in light of E2-TTS. Since A^{2}-Flow is a concurrent work, there is no need to discuss it here. This work presents moderate modeling complexity, positioned between complex modeling with alignment and simple modeling without alignment. As the advantages of this method, it appears to show sample robustness similar to models with alignment and naturalness akin to models without alignment. This contribution and its impact suggest that sparse alignment can be considered a future direction alongside modeling with or without alignments. Therefore, I remain confident in my assessment but would also agree with the AC’s judgment, as the value of this work should be considered alongside other reviewers’ assessments and the merit of other TTS submissions at ICLR.

---

> ### Author Response · Authors · 2024-11-26
>
> Thank you for your detailed feedback and thoughtful assessment of our work! We deeply appreciate your balanced evaluation, especially regarding the contribution of sparse alignment and its potential as a future direction in TTS alignment modeling. We would fully respect your judgment, as well as the AC’s, throughout the review process.

---

### Note · Authors · 2025-02-15

I have read and agree with the venue's withdrawal policy on behalf of myself and my co-authors.

---

### Meta-Review · Area_Chair_YnmK · 2024-12-18

**Metareview:**

This work introduces the S-DiT model, which achieves expressive text-to-speech (TTS) comparable to models that learn alignment implicitly, despite leveraging predefined alignment. Unlike prior approaches that rely on dense alignment with a phoneme duration predictor, S-DiT uses sparse alignment by providing only partial, randomly selected alignment information. This strategy enables the model to learn expressiveness while preserving alignment robustness. To support phoneme duration prediction and provide transcripts for prompts, the paper proposes a frontend language model (F-LM) framework and incorporates piecewise flow matching to enhance sampling speed. The model delivers high speaker similarity and accurate pronunciation in zero-shot TTS tasks, while achieving significantly faster sampling rates.

The authors' claim  are: (1) Proposal of a new sparse alignment enhanced latent diffusion transformer mode, which maintains the robustness and reduces the search space constraints from forced alignments, (2) Deployment of a multi-condition CFG strategy for modulating the intensity of personal accents, offering novel solutions for accent control; (3) The device of an  F-LM model  that not only simplifies the inference process of zero-shot TTS models like NaturalSpeech 3, but it  also can be directly used for processing training data during model fine-tuning. The unified training framework enhances F-LM’s speech understanding capabilities, allowing it to surpass the independent modules for each subtask.

Key strengths of the work can be summarized as follows: (i) Well written and structured paper, (ii) The model has surpassed various previous models and achieved close-to-SOTA performance, (iii) The paper demonstrates effective use of piecewise rectified flow to reduce inference time without significant quality loss, and (iv) The introduction of F-LM simplifies the inference process by predicting phoneme durations and representations using only the prompt audio and the target text sentence.

During the initial review phase, several significant concerns were raised, prompting the authors to conduct additional experiments and provide detailed clarifications. On one hand, these new experiments and analyses addressed many of the reviewers' initial questions and resolved some of their doubts. On the other hand, three key weaknesses appear to persist:

(1) The use of MFA imposes a substantial training burden on the model compared to approaches that internally learn alignment using self- or cross-attention mechanisms.

(2) The impact and experimental results of this work need to be carefully evaluated considering E2-TTS, as highlighted by different reviewers.

(3) Finally, certain submissions to ICLR 2025 present experimental findings that contradict specific claims made by the authors, reporting differing results that appear to challenge the validity of some conclusions.


With respect to point (3), it should be clarified that the novelty of the contributions in this work should not be denied in light of current submissions to ICLR 2025. However, it might be admittedly difficult to overlook the fact that some of these works provide insights that contradict certain experimental evidence supporting the authors’ claims. However, even setting aside those works submitted to ICLR 2025, Reviewers hj2E and pQoE seem to align with Reviewer pGsx’s concerns regarding the E2-TTS results. Reviewer hj2E notes: “E2TTS results from F5-TTS were presented on its demo page, and they appear to perform worse in terms of objective metrics compared to Microsoft’s official E2TTS. Therefore, comparisons with the E2TTS released by F5-TTS cannot be considered direct comparisons with E2TTS.”  Similarly, Reviewer pQoE states in their final comments: “I believe the impact and experimental results of this work should be considered in light of E2-TTS.”

Finally, the use of MFA introduces a notable drawback, as the model incurs a greater training burden compared to approaches that internally learn alignment using self- or cross-attention mechanisms.

**Additional Comments On Reviewer Discussion:**

The discussion phase was highly active, culminating in 51 forum replies. Notably, the exchanges between the authors and Reviewer pGsx were particularly intense. The key points raised by the reviewers can be summarized as follows:

1. The S-DiT model may face challenges when working with smaller-scale datasets.
2. Potential failure cases in speech generation are not addressed.
3. The evaluation of the reconstruction quality of the speech compression model is missing.
4. The performance of the external aligner may significantly impact the model's extension to other languages, which was not tested in the paper.
5. The descriptions of the module details are overly simplified.
6. Details regarding the generation of rough alignments are missing.
7. The method for stochastically sampling phoneme durations from F-LM is not described.
8. Reviewer pGsx raised several concerns regarding the validity of specific claims made throughout the paper.
9. E2TTS results from F5-TTS, presented on its demo page, appear to perform worse in terms of objective metrics compared to Microsoft’s official E2TTS. Therefore, comparisons with the E2TTS released by F5-TTS cannot be considered direct comparisons with E2TTS. Additionally, the improvement in speech intelligibility compared to E2-TTS remains unclear.
10. The training pipeline of S-DiT is overly complex.

In response, the authors conducted several new experiments and provided additional analyses to address these comments and further concerns raised by reviewers. Reviewer pGsx acknowledged some misunderstandings about the authors’ claims and results. The authors convincingly addressed points 1, 2, 3, 5, 6, 7, and 8. They also acknowledged the complexity of their training pipeline (point 10). However, the responses to points 4 and 9 remain contentious.

By the end of the discussion phase, the outcomes among reviewers were as follows:
- Reviewer syxD expressed satisfaction with the authors’ responses and raised their score from 5 to 6.
- Reviewer pQoE maintained their score at 6, based on the results demonstrated by the current model.
- Reviewer pGsx increased their score from 3 to 5 but stated that the unresolved issues outweighed the merits of the paper.
- Reviewer hj2E kept their score at 6, reiterating that the E2TTS results from F5-TTS, as presented on its demo page, appear worse in terms of objective metrics compared to Microsoft’s official E2TTS.

This final state reflects both progress in addressing key concerns and lingering reservations among the reviewers.

The authors expressed their disappointment with Reviewer **pGsx's** comments and concerns, suggesting that the review may not fully align with the standards of fairness expected in this process. While Reviewer **pGsx** may have been overly assertive in defending their perspective, their final concerns (notably **weakness 2**) remain valid, even when disregarding the works submitted to ICLR 2025.

---

### Decision · Program_Chairs · 2025-01-22

Reject